# CUT THE CRAP: AN ECONOMICAL COMMUNICATION PIPELINE FOR LLM-BASED MULTI-AGENT SYSTEMS

**Guibin Zhang**[1†], **Yanwei Yue**[1†], **Zhixun Li**[2†], **Sukwon Yun**[3], **Guancheng Wan**[4],
**Kun Wang**[5*], **Dawei Cheng**[1,6], **Jeffrey Xu Yu**[2], **Tianlong Chen**[3]
[1]Tongji University  [2]The Chinese University of Hong Kong
[3]University of North Carolina at Chapel Hill  [4]Wuhan University
[5]Nanyang Technological University  [6]Shanghai AI Laboratory

## ABSTRACT

Recent advancements in large language model (LLM)-powered agents have shown that collective intelligence can significantly outperform individual capabilities, largely attributed to the meticulously designed inter-agent communication topologies. Though impressive in performance, existing multi-agent pipelines inherently introduce substantial token overhead, as well as increased economic costs, which pose challenges for their large-scale deployments. In response to this challenge, we propose an economical, simple, and robust multi-agent communication framework, termed `AgentPrune`, which can seamlessly integrate into mainstream multi-agent systems and prunes redundant or even malicious communication messages. Technically, `AgentPrune` is the first to identify and formally define the *communication redundancy* issue present in current LLM-based multi-agent pipelines, and efficiently performs one-shot pruning on the spatial-temporal message-passing graph, yielding a token-economic and high-performing communication topology. Extensive experiments across six benchmarks demonstrate that `AgentPrune` **(I)** achieves comparable results as state-of-the-art topologies at merely \$5.6 cost compared to their \$43.7, **(II)** integrates seamlessly into existing multi-agent frameworks with $28.1\% \sim 72.8\% \downarrow$ token reduction, and **(III)** successfully defend against two types of agent-based adversarial attacks with $3.5\% \sim 10.8\% \uparrow$ performance boost. The source code is available at https://github.com/yanweiyue/AgentPrune.

## 1 INTRODUCTION

Large Language Model (LLM) based agents (Richards & et al., 2023; Nakajima, 2023; Reworkd, 2023) have demonstrated strong performance across a diverse range of tasks, including reasoning (Yao et al., 2023b), code generation (Shinn et al., 2023), and even more complex applications like video gaming (Wang et al., 2023) and autopilot systems (Jin et al., 2023). Recent endeavors have shown that combining implicitly or explicitly different LLM-based agents into a team can outperform a single agent in handling complex tasks (Du et al., 2023b; Liang et al., 2023; Wang et al., 2023c; Jiang et al., 2023; Shinn et al., 2023; Zheng et al., 2023; Wu et al., 2023), which supports the presence of human-esque collaborative intelligence in multi-agent systems (Zhang et al., 2023b). In practice, previous research has explored approaches in which instances of LLMs, referred to as agents (Wang et al., 2024b; Xi et al., 2023;

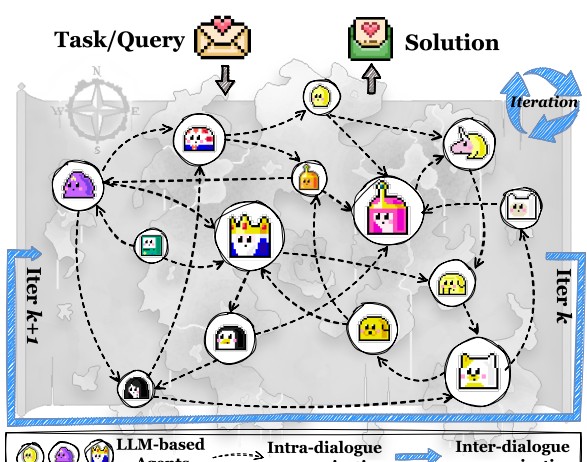

**Figure 1:** The workflow of existing LLM-based (problem-solving) multi-agent systems. The agent-agent communication occurs at both intra- and inter-dialogue stages.

---

*Kun Wang is the corresponding author, † denotes equal contributions.

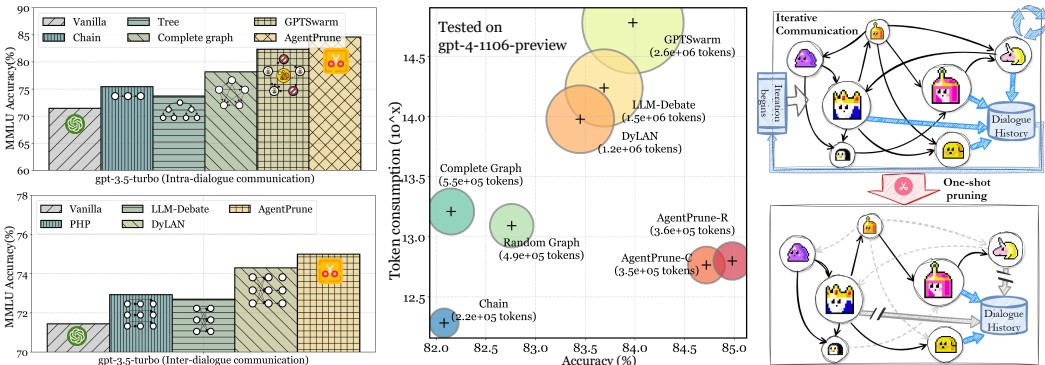

**Figure 2:** (*Left*) The accuracy comparison on MMLU (Hendrycks et al., 2021) among (1) a single `gpt-3.5-turbo`, (2) three `gpt-3.5-turbo` agents equipped with intra-dialogue communication structures like chain and tree (Qian et al., 2024), complete graph, and GPTSwarm (Zhuge et al., 2024), and (3) those equipped with inter-dialogue communication structures like PHP (Zheng et al., 2023), LLM-Debate (Du et al., 2023b), DyLAN (Liu et al., 2023b) and our `AgentPrune`. (*Middle*) The prompt token consumption comparison on MMLU between different methods. (*Right*) The overview of our proposed `AgentPrune`.

Gao et al., 2023; Cheng et al., 2024; Ma et al., 2024), collaborate synergistically (*e.g.*, through debate or reflection) to complete tasks (Du et al., 2023a; Pezeshkpour et al., 2024; Guo et al., 2024; Du et al., 2024; Han et al., 2024) via diverse communication topologies (*e.g.*, chain (Wei et al., 2022), tree (Yao et al., 2023a), complete graph (Qian et al., 2024), random graph (Qian et al., 2024), optimizable graph (Zhuge et al., 2024), LLM-based network (Hao et al., 2023; Liu et al., 2023b)). The exceptional performance of these cooperative agents significantly benefits from their interactive communication and collaboration, specifically how agents *transmit, exchange,* and *assimilate* information (Chan et al., 2023; Wang et al., 2023c; Liu et al., 2023b).

Taking a closer look into the communication mechanisms in existing multi-agent systems, they typically involve two key types (as shown in Figure 1): ❶ **Intra-dialogue communication**: For a given query/task, multiple agents interact–whether by cooperating (Du et al., 2023a; Wu et al., 2023; Hong et al., 2024), teaching (Zhang et al., 2024d), or competing (Zhao et al., 2023; Fu et al., 2023)– to produce a solution within a dialogue round; ❷ **Inter-dialogue communication**: In a specific manner—whether by summarizing (Chan et al., 2023; Shen et al., 2024), replicating (Yin et al., 2023; Du et al., 2023b), or filtering (Liu et al., 2023b)—the content of the current dialogue is passed to the next round of interaction as a reference, initiating a new cycle of collaborative efforts.

To better illustrate the power of agent communication, Figure 2 (*Left*) compares the performance of a single `gpt-3.5-turbo` with three agents equipped with different inter/intra-dialogue communication structures. The results demonstrate that even the simplest communication framework significantly leads to a notable accuracy improvement, which vividly showcases the social intelligence and collaborative capabilities of LLMs (Mei et al., 2024). However, the success of multi-agents comes at the cost of significantly increased token consumption, imposing substantial economic burdens (Wang et al., 2024a), which are detrimental to the widespread application of multi-agent systems, as deployment on edge devices does not accommodate excessively costly inference (Liu et al., 2023a). A piece of empirical evidence is in Figure 2 (*Middle*), where various communication methods result in a $2 \sim 11.8\times$ increase in token consumption compared to the simple chain structure, severely undermining the **token economy** of existing multi-agent systems.

In the light of this limitation, we *for the first time* identify a significant phenomenon of *Communication Redundancy* (Meyer et al., 2021) within existing LLM-based multi-agent (LLM-MA) communication topologies, where a substantial portion of message passing does not contribute meaningfully to the collaborative intelligence. With this finding, we introduce *an economical and versatile communication pruning framework for LLM-powered multi-agent systems*, dubbed `AgentPrune`, which can be smoothly incorporated within various existing LLM-MA systems, offering comparable reasoning and planning performance as well as significantly lower token consumption. Practically, `AgentPrune` treats the entire LLM-MA framework as a spatial-temporal communication graph, in which each agent, along with its unique properties (*e.g.*, profile (Li et al., 2023a), external API tools (Zhuang et al., 2023), or knowledge base (Chen et al., 2024a)), is packaged as a node, communication between agents within the same dialogue forms spatial edges, and communication across dialogues forms temporal edges. By training a low-rank-principle-guided graph mask, `AgentPrune`

efficiently identifies the important graph connectivities (*i.e.*, message passing through edges). This comes with a one-shot pruning to derive a sparse yet informative communication graph (in Figure 2 (*Right*)), which is then fixed as the communication topology for subsequent token-economic and efficient reasoning. Our contributions can be summarized as follows:

❶ **System Discovery.** We present a spatial-temporal graph paradigm to describe the communication topology of contemporary LLM-MA frameworks, and further identify and define the *Communication Redundancy* issue in current systems, wherein a significant portion of spatial and temporal edges, *i.e.*, communication, does not contribute to collaborative intelligence.

❷ **Pratical Solution.** We propose `AgentPrune`, an economical, simple, and robust multi-agent communication pruning pipeline. By leveraging a trainable communication graph mask, `AgentPrune` identifies key message exchanges and prunes non-essential components in a one-shot manner, resulting in a sparse, token-economical, and highly informative communication graph. Notably, `AgentPrune` employs a low-rank principle to guide the graph mask training, successfully robustifying LLM-MA systems against various agent-targeted adversarial attacks.

❸ **Experimental Validation.** Extensive experiments on six benchmarks show that `AgentPrune` is: **(1) high-performing**, achieving comparable performance on MMLU at \$5.6 cost, to that of state-of-the-art communication topologies at \$43.7; **(2) token-economical**, integrating seamlessly into popular multi-agent frameworks including AutoGen and GPTSwarm, reducing their token cost by $28.1\% \sim 72.8\% \downarrow$; and **(3) adversarially robust**, successfully defending against two types of agent adversarial attacks, with a $3.5\% \sim 10.8\% \uparrow$ performance improvement.

## 2 LLM-MA AS SPATIAL-TEMPORAL GRAPHS

**Notations** We describe the whole multi-agent system as a graph $\mathcal{G} = (\mathcal{V}, \mathcal{E})$, with $\mathcal{V} = \{v_1, v_2, \cdots, v_{|\mathcal{V}|}\}$ being the node set and $\mathcal{E}$ being the edge set. Each node $v_i \in \mathcal{V}$ represents an agent, which can be further interpreted as follows:

$$v_i = \{\texttt{Base}_i, \texttt{Role}_i, \texttt{State}_i, \texttt{Plugins}_i\}, \quad \texttt{Plugins}_i = \{\texttt{F}_j, \texttt{C}_j\}_{j=1}^P, \quad (1)$$

where an agent $v_i$ consists of: (1) $\texttt{Base}_i$, the language model instance used by $v_i$; (2) $\texttt{Role}_i$, the pre-defined role or responsibility of the agent; (3) $\texttt{State}_i$, the state of the agent, encapsulating the accumulated knowledge and experience from previous interactions; (4) $\texttt{Plugins}_i$, a set of $P$ external plugins available to agent $v_i$, where each plugin is defined by its functionalities $\texttt{F}_j$ (*e.g.*, web search, python compiler) and configurations $\texttt{C}_j$.

For the edges, we divide them into two subsets: **intra-dialogue (spatial) edges** $\mathcal{E}^{\mathcal{S}} \subseteq \mathcal{V}^{(t)} \times \mathcal{V}^{(t)}$ and **inter-dialogue (temporal) edges** $\mathcal{E}^{\mathcal{T}} \subseteq \mathcal{V}^{(t-1)} \times \mathcal{V}^{(t)}$. For each spatial edge $e_{ij}^{\mathcal{S}} = (\mathbf{M}_{ij}, \mathbf{O}_{ij})$, it represents the information flow from agent $v_i$ to agent $v_j$ *within the same utterance*, composed of the message content $\mathbf{M}_{ij}$, as well as the associated operation $\mathbf{O}_{ij}$ (*e.g.*, task assignments, requests). For each temporal edge $e_{ij}^{\mathcal{T}}$, it denotes the message passing *between two utterance rounds*, *i.e.*, whether the output from agent $v_i$ in the $(t-1)$-th round should be passed on to agent $v_j$ in the $t$-th round. Further, we define the temporal/spatial (in-)neighbors for each agent as follows:

$$\mathcal{N}^{\mathcal{T}}(v_i) = \{v_j \mid (j, i) \in \mathcal{E}^{\mathcal{T}}\}, \quad \mathcal{N}^{\mathcal{S}}(v_i) = \{v_j \mid (j, i) \in \mathcal{E}^{\mathcal{S}}\}. \quad (2)$$

**Multi-agent Communication** We provide a graph-based description of the reasoning process in task-oriented multi-agent systems. Given a query/task $q$, it is sequentially fed to each agent, which produces its output. To maintain an orderly sequence of agent interactions, we utilize topological ordering (Bondy et al., 1976) to ensure that each node is processed only after all its dependencies have been addressed. This necessitates that the spatial communication graph $\mathcal{G}^{\mathcal{S}} = (\mathcal{V}, \mathcal{E}^{\mathcal{S}})$ be structured as a directed acyclic graph (DAG). Formally, for $\mathcal{G}^{\mathcal{S}}$, the following condition holds:

$$\forall (v_i, v_j), \; \mathbb{I}(v_i) < \mathbb{I}(e_{ij}) < \mathbb{I}(v_j), \quad (3)$$

where $\mathbb{I}(x)$ denotes the execution order of $x$. For each agent $v_i^{(t)}$ at round $t$, it produces its rationale or answers, uniformly denoted as $\mathbf{M}_i$, as follows:

$$\mathbf{M}_i^{(t)} \sim \mathcal{P}_\theta \left( \mathbf{M}_i \mid q, \texttt{Role}_i^{(t)}, \texttt{State}_i^{(t)}, \overbrace{\cup_{v_j \in \mathcal{N}^{\mathcal{T}}(v_i)} \mathbf{M}_j}^{\text{temporal}}, \overbrace{\cup_{v_j \in \mathcal{N}^{\mathcal{S}}(v_i)} \mathbf{M}_{ji}}^{\text{spatial}} \right), \quad (4)$$

where agent $v_i$ responds based on the query $q$, its current role and state, temporal and spatial messages, and certain prompting instruction $\mathcal{P}_\theta$. Typically, after $K$ rounds of dialogue, a summarizer agent or an answer aggregation mechanism (*e.g.*, voting) is employed to produce the final solution $a^{(K)}$ for the given query $q$. We conclude the general pipeline in Algorithm 1.

---

**Algorithm 1:** Execution pipeline of LLM-MA systems from spatial-temporal graph perspective

---

**Input:** Query $q$, Communication graph $\mathcal{G} = \{\mathcal{G}^{\mathcal{S}}, \mathcal{G}^{\mathcal{T}}\}$, Maximum number of iterations $N$

1   **for** iteration $t \leftarrow 1$ **to** $N$ **do**
2     **if** `MeetEndCondition()` **then**
3       break // `Extra stopping criteria like agent consensus`
4     **end**
5     **for** $v_i$ in `TopologicalSort(V)` **do**
6       $\boldsymbol{m}^{\mathcal{T}} \leftarrow \{\mathbf{M}_j \mid v_j \in \mathcal{N}^{\mathcal{T}}(v_i)\}$ // `Messages from temporal in-neighbors`
7       $\boldsymbol{m}^{\mathcal{S}} \leftarrow \{\mathbf{M}_{ij} \mid v_j \in \mathcal{N}^{\mathcal{S}}(v_i)\}$ // `Messages from spatial in-neighbors`
8       $\mathbf{M}_i^{(t)} \sim \mathcal{P}_\theta(\mathbf{M}_i \mid q, \texttt{Role}_i^{(t)}, \texttt{State}_i^{(t)}, \boldsymbol{m}^{\mathcal{T}}, \boldsymbol{m}^{\mathcal{S}})$ // `Generate rationale or answer`
9     **end**
10    $a^{(t)} \leftarrow \texttt{AggregateSolution}(\mathbf{M}_1^{(t)}, \mathbf{M}_2^{(t)}, \cdots, \mathbf{M}_{|\mathcal{V}|}^{(t)})$ // `Depending on the specific system, possible implementations of` `AggregateSolution` `include (but are not limited to) majority voting or using the output of a summarizer agent.`
11 **end**
12 **return** $a^{(t)}$ as the final solution

---

**Problem Formulation**   In this section, we explore and define the *communication redundancy* issue within existing multi-agent communication pipelines. Specifically, we examine two representative communication topologies: (1) for *spatial communication*, the fully-connected mesh graph from MacNet (Qian et al., 2024), which exemplifies a densely structured intra-utterance communication, and (2) for *temporal communication*, the LLM-Debate (Du et al., 2023b), where at the

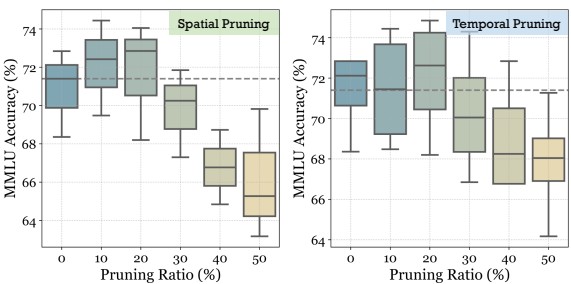

**Figure 3:** The performance of mesh graph and LLM-Debate structure under different random pruning ratios on MMLU.

start of each dialogue round, an agent receives all responses from the previous round as input. Using four `gpt-3.5-turbo` as agents, we assess system performance on MMLU after randomly pruning a certain proportion of connections. As illustrated in Figure 3, when randomly removing $10\% \sim 30\%$ of the communication connectivity, the performance actually gains up to $2.83\%$ improvement. This suggests that, in both spatial and temporal information flow, a substantial portion of messages does not contribute to the task-solving process, which we formally define as follows:

**Definition 1** (Communication Redundancy). *For any LLM-based multi-agent communication graph $\mathcal{G} = (\mathcal{V}, \mathcal{E}^{\mathcal{S}} \cup \mathcal{E}^{\mathcal{T}})$, the following condition holds:*

$$\exists\, \mathcal{G}^{sub} = (\mathcal{V}, \mathcal{E}' \cup \mathcal{E}'') \subseteq \mathcal{G}, \text{ where } \mathcal{E}' \subseteq \mathcal{E}^{\mathcal{S}},\ \mathcal{E}'' \subseteq \mathcal{E}^{\mathcal{T}},\ \text{s.t. }\ \phi(\mathcal{G}^{sub}) \geq \phi(\mathcal{G}), \tag{5}$$

*where $\phi(\cdot)$ represents a utility function that measures the solution quality achieved by the system. The redundant components in the communication topology, denoted as $(\mathcal{E}^{st} \setminus \mathcal{E}') \cup (\mathcal{E}^{tp} \setminus \mathcal{E}'')$, are referred to as the communication redundancy in LLM-MA systems.*

We further outline the objective of this study as follows:

$$\arg\max_{\mathcal{E}', \mathcal{E}''} \mathcal{G} \setminus \mathcal{G}^{\text{sub}}, \text{ s.t. } |\phi(\mathcal{G}^{\text{sub}}) - \phi(\mathcal{G})| \leq \epsilon, \tag{6}$$

where $\epsilon$ represents the allowable threshold for performance variation. Equation (6) aims to minimize communication redundancy with performance guarantee.

## 3   METHODOLOGY

Figure 4 illustrates how our method is applied within an LLM-MA system. Specifically, given an input query, **AgentPrune** first performs *spatial pruning* by eliminating redundant spatial messages within a dialogue round, followed by *temporal pruning* to discard unnecessary dialogue history. In the following sections, we will first explain how **AgentPrune** facilitates efficient multi-round communication based on an optimizable spatial-temporal communication graph (▷ Sections 3.1 and 3.2), leverages one-shot pruning to derive a sparse interaction topology (▷ Section 3.3), and finally, detail the optimization paradigm for the entire framework (▷ Section 3.4).

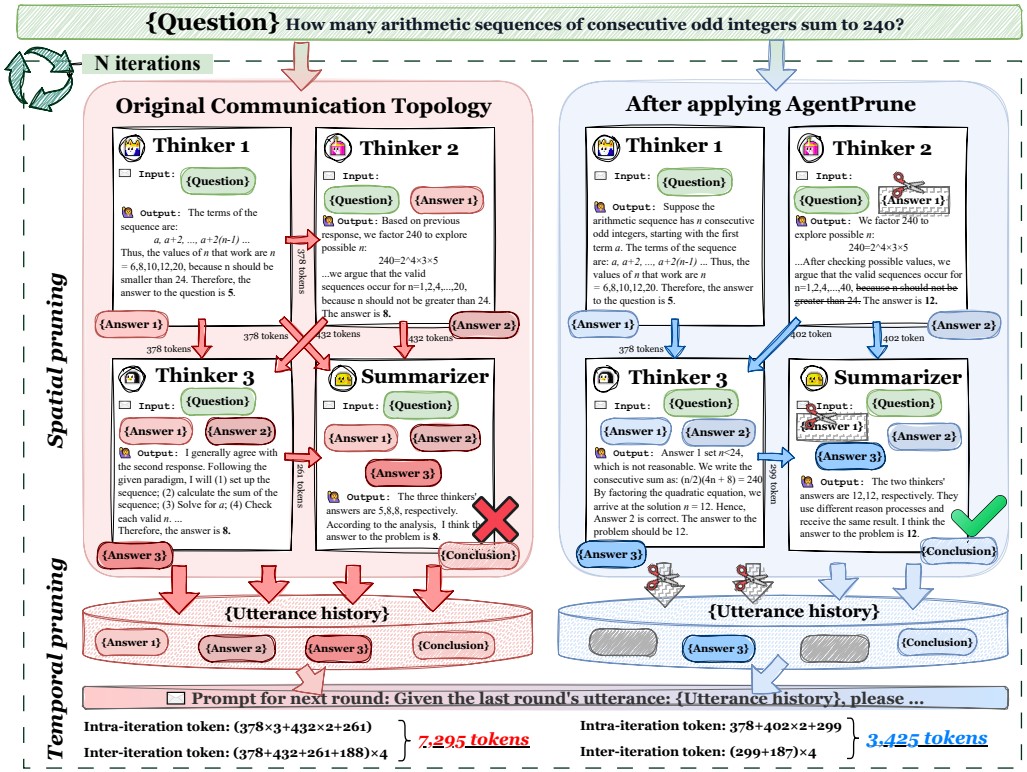

**Figure 4:** The overview of our proposed `AgentPrune`.

## 3.1 SPATIAL-TEMPORAL GRAPH COMMUNICATION

Given an arbitrary LLM-MA system and its corresponding spatial-temporal communication graph $\mathcal{G}$, the task of `AgentPrune` is to discover its sparse yet equally high-performing counterpart $\mathcal{G}^{\text{sub}}$. The objective is essentially a graph sparsification problem (Spielman & Srivastava, 2008; Chen et al., 2023b), whose goal is to identify the essential graph connections and discard the less critical ones. To achieve this, following classical practices in graph sparsification (Chen et al., 2021b; Wang et al., 2023a; Zhang et al., 2024a), we relax the original binary communication graph $\mathcal{G}$ by transforming its edge elements from binary values to continuous variables, denoted as $\tilde{\mathcal{G}}$. We have:

$$\mathbf{A}(\mathcal{G}) = \{\mathbf{A}^{\mathcal{S}}, \mathbf{A}^{\mathcal{T}}\}, \ \mathbf{A}(\tilde{\mathcal{G}}) = \mathbf{A}(\{\tilde{\mathcal{G}}^{\mathcal{S}}, \tilde{\mathcal{G}}^{\mathcal{T}}\}) = \{\mathbf{A}^{\mathcal{S}} \odot \mathbf{S}^{\mathcal{S}}, \mathbf{A}^{\mathcal{T}} \odot \mathbf{S}^{\mathcal{T}}\}, \tag{7}$$

where $\mathbf{A}(\mathcal{G})$ obtains the adjacency matrix of input graph $\mathcal{G}$, and $\mathbf{A}^{\mathcal{S}}, \mathbf{A}^{\mathcal{T}} \in \{0,1\}^{|\mathcal{V}| \times |\mathcal{V}|}$ represent the spatial and temporal adjacency matrices, respectively. Specifically, $\mathbf{A}^{x}[i,j] = 1$ indicates that $e_{ij} \in \mathcal{E}^{x}$, and 0 otherwise. It is important to note that both $\mathbf{A}^{\mathcal{S}}$ and $\mathbf{A}^{\mathcal{T}}$ are predefined by the LLM-MA system. $\mathbf{S}^{\mathcal{S}}, \mathbf{S}^{\mathcal{T}} \in \mathbb{R}^{|\mathcal{V}| \times |\mathcal{V}|}$ are differentiable graph masks. As mentioned in Section 2, we require the interaction topology to be a DAG to ensure that agent input/output (I/O) can be processed sequentially. Therefore, we leverage a `DAGSampling` function to transform the original $\tilde{\mathcal{G}}^{\mathcal{S}}$ into a DAG: $\hat{\mathcal{G}}^{\mathcal{S}} \leftarrow \texttt{DAGSampling}(\tilde{\mathcal{G}}^{\mathcal{S}})$, whose procedure is described in Algorithm 2.

## 3.2 OPTIMIZING SPATIAL-TEMPORAL CONNECTIVITY

With $\hat{\mathcal{G}}^{\mathcal{S}}$ and $\tilde{\mathcal{G}}^{\mathcal{T}}$ in hand, we aim to optimize them toward both high-performance and token efficiency. To this end, we introduce two optimization objectives for $\hat{\mathcal{G}}^{\mathcal{S}}$ and $\tilde{\mathcal{G}}^{\mathcal{T}}$: ❶ *distribution approximation*, ensuring accurate estimation of their underlying probability distributions, and ❷ *low-rank sparsity*, which promotes a more efficient and sparse structure (Li et al., 2024). The first objective ensures that the magnitudes of the graph masks correctly reflect the importance of different communication channels, facilitating subsequent redundancy pruning, and the second ensures that the learned connectivity remains sparse and robust. Formally, we define the following objective:

$$\underset{\mathbf{S}^{\mathcal{S}}, \mathbf{S}^{\mathcal{T}} \in \mathbb{S}}{\arg\max} \ \overbrace{\mathbb{E}_{\hat{\mathcal{G}}^{\mathcal{S}}, \mathcal{G}^{\mathcal{T}} \sim \mathbb{G}} \left[ \phi \left( \{\hat{\mathcal{G}}^{\mathcal{S}}, \tilde{\mathcal{G}}^{\mathcal{T}}\} \right) \right]}^{\text{distribution approximation}} - \overbrace{\sum_{\mathcal{X} \in \{\mathcal{S}, \mathcal{T}\}} \text{rank}(\mathbf{S}^{\mathcal{X}})}^{\text{low-rank sparsity}}, \ \text{s. t.} \sum_{\mathcal{X} \in \{\mathcal{S}, \mathcal{T}\}} ||\mathbf{A}^{\mathcal{X}} - \mathbf{S}^{\mathcal{X}}||_{F} \leq \delta, \tag{8}$$

where $\mathbb{S}$ and $\mathbb{G}$ represent the viable parameter space, $\phi(\cdot)$ serves as the utility evaluator for the input multi-agent framework, $\text{rank}(\cdot)$ calculates the rank of matrix, and $\delta$ is the noise level. Next, we will provide a detailed explanation of the implementation of these two optimization objectives.

**Distribution Approximation**    The first term in Equation (8) encourages $\{\mathbf{S}^{\mathcal{S}}, \mathbf{S}^{\mathcal{T}}\}$ towards the maximization of the system's utility. However, since $\phi(\cdot)$ often depends on external APIs (Li et al., 2023b) or compilers (Chen et al., 2021a) for evaluation, it is generally non-differentiable. Therefore, we employ policy gradient (Williams, 1992) to make Equation (8) tractable:

$$\nabla_{\mathbf{S}} \, \mathbb{E}_{\hat{\mathcal{G}}^{\mathcal{S}}, \tilde{\mathcal{G}}^{\mathcal{T}} \sim \mathbb{G}} \left[ \phi\left(\{\hat{\mathcal{G}}^{\mathcal{S}}, \tilde{\mathcal{G}}^{\mathcal{T}}\}\right) \right] \approx \frac{1}{M} \sum_{k=1}^{M} \phi\left(\{\hat{\mathcal{G}}_k^{\mathcal{S}}, \tilde{\mathcal{G}}_k^{\mathcal{T}}\}\right) \nabla_{\mathbf{S}} \log\left(p_{\mathbf{S}}(\{\hat{\mathcal{G}}_k^{\mathcal{S}}, \tilde{\mathcal{G}}_k^{\mathcal{T}}\})\right), \quad (9)$$

$$p_{\mathbf{S}}\left(\{\hat{\mathcal{G}}_k^{\mathcal{S}}, \tilde{\mathcal{G}}_k^{\mathcal{T}}\}\right) = \left(\prod \mathbb{1}_{e_{ij} \in \mathcal{E}^{\mathcal{S}}} \mathbf{S}^{\mathcal{S}}[i, j]\right) \cdot \left(\prod \mathbb{1}_{e_{ij} \in \mathcal{E}^{\mathcal{T}}} \mathbf{S}^{\mathcal{T}}[i, j]\right) \quad (10)$$

where $\mathbf{S} = \{\mathbf{S}^{\mathcal{S}}, \mathbf{S}^{\mathcal{T}}\}$, $\{\hat{\mathcal{G}}^{\mathcal{S}}, \tilde{\mathcal{G}}^{\mathcal{T}}\}_{k=1}^{M}$ are independently sampled from $\{\hat{\mathcal{G}}^{\mathcal{S}}, \tilde{\mathcal{G}}^{\mathcal{T}}\}$, $p_{\mathbf{S}}(\{\hat{\mathcal{G}}_k^{\mathcal{S}}, \tilde{\mathcal{G}}_k^{\mathcal{T}}\})$ calculates the probability of the sampled structure, and $\mathbb{1}(\cdot)$ is an indicator function.

**Low-rank Sparsity**    The second term in Equation (8) promotes the graph masks $\{\mathbf{S}^{\mathcal{S}}, \mathbf{S}^{\mathcal{T}}\}$ to be low-rank, which not only filters out informative agent communications but also aids in removing redundant, noisy, and even malicious messages, which has been demonstrated in recent studies, showing that low-rank graphs are more robust to network attacks (Entezari et al., 2020; Ennadir et al., 2024). We will empirically validate `AgentPrune`'s ability to enhance multi-agent robustness in Section 4.4. However, directly optimizing the rank minimization is NP-hard, so we replace the rank function with the nuclear norm as an alternative, reformulating this term as follows:

$$\underset{\mathbf{S}^{\mathcal{S}}, \mathbf{S}^{\mathcal{T}} \in \mathbb{S}}{\arg\min} \sum_{\mathcal{X} \in \{\mathcal{S}, \mathcal{T}\}} ||\mathbf{S}^{\mathcal{X}}||_*, \text{ s.t. } ||\mathbf{A}^{\mathcal{X}} - \mathbf{S}^{\mathcal{X}}||_F \leq \delta, \quad (11)$$

where $||\mathbf{S}||_* = \sum_i \sigma_i$, and $\sigma_i$ represents the $i$-th singular value of $\mathbf{S}$. Guided by Equation (8), we iteratively optimize the spatial-temporal connectivity in conjunction with the multi-agent conversation over $K'$ rounds, where $K' \ll K$.

## 3.3 ONE-SHOT PRUNING

We dynamically optimize $\{\mathbf{S}^{\mathcal{S}}, \mathbf{S}^{\mathcal{T}}\}$ for only $K'$ iterations, rather than the full $K$ iterations, because prior work on Early-bird (EB) and Graph EB has demonstrated that limited training can also construct high-quality benchmarks reflecting the topology distribution (Achille et al., 2018; You et al., 2019; Zhang et al., 2024b), which also aligns with `AgentPrune`'s token-saving initiation. To eliminate redundancy in the current communication structure, we perform one-shot magnitude pruning on the optimized graph masks $\mathbf{S}$ (either $\mathbf{S}^{\mathcal{S}}$ or $\mathbf{S}^{\mathcal{T}}$):

$$\mathbf{B} = \mathbb{1}\left(\mathbf{A} \neq 0 \wedge \text{TopK}\left(\mathbf{S}, |\mathbf{A}| \times (1 - p\%)\right)\right), \quad (12)$$

where $\text{TopK}(S, x\%)$ return the largest $x\%$ elements in matrix $S$, and $p\%$ is the pruning ratio. By applying the binary masks to the original topology, we obtain sparse, compact, and communication-minimizing connectivity $\mathcal{G}^{\text{sub}}$, where $\mathbf{A}(\mathcal{G}^{\text{sub}}) = \{\mathbf{A}^{\mathcal{S}} \odot \mathbf{B}^{\mathcal{S}}, \mathbf{A}^{\mathcal{T}} \odot \mathbf{B}^{\mathcal{T}}\}$. In the subsequent $(K - K')$ rounds, the entire framework's message passing pipeline is strictly constrained by $\mathcal{G}^{\text{sub}}$, and agents are continuously optimized to refine the solution for query $q$.

## 3.4 APPLICATION AND ANALYSIS

**Algorithm Pipeline**    As a plug-and-play module, `AgentPrune` can be harmoniously embedded in mainstream multi-agent frameworks to facilitate token-efficient communication, provided that the number of agents exceeds three and the communication structure is moderately organized (*e.g.*, chain or direct-output structures are too simple to be applicable). When combined with `AgentPrune`, multiple agents first undergo $K'$ rounds of interactions alongside trainable graph masks, which are then one-shot pruned to yield the sparse $\mathcal{G}^{\text{sub}}$, leveraged for the subsequent $(K - K')$ rounds of optimization. We summarize all the notations used in Appendix B and the comprehensive algorithmic workflow in Appendix C.

**Multi-Query Training**    For complex tasks like repository-level code generation (Qian et al., 2023; Liu et al., 2024), multi-turn dialogues are often inevitable. However, for simpler tasks that involve a

**Table 1:** Performance comparison with three types of baselines, including single-agent execution, spatial communication and temporal communication. The best results are highlighted in bold, and the runner-ups are underlined. All methods, except for the single-agent category, utilize **five** `gpt-4`-based agents.

| Method | Spa. | Tem. | MMLU | GSM8K | MultiArith | SVAMP | AQuA | HumanEval | Avg. |
|---|---|---|---|---|---|---|---|---|---|
| Vanilla | ✗ | ✗ | 82.14 | 85.40 | 93.15 | 87.18 | 70.34 | 71.68 | 81.65 |
| CoT | ✗ | ✗ | $82.65_{\uparrow0.51}$ | $87.17_{\uparrow1.77}$ | $94.79_{\uparrow1.64}$ | $88.32_{\uparrow1.14}$ | $73.91_{\uparrow3.57}$ | $75.52_{\uparrow3.84}$ | 83.73 |
| ComplexCoT | ✗ | ✗ | $83.78_{\uparrow1.64}$ | $87.62_{\uparrow2.22}$ | $95.86_{\uparrow2.71}$ | $90.17_{\uparrow2.99}$ | $77.58_{\uparrow7.24}$ | $74.94_{\uparrow3.26}$ | 84.99 |
| SC (CoT) | ✗ | ✗ | $82.66_{\uparrow0.52}$ | $87.93_{\uparrow2.53}$ | $96.88_{\uparrow3.73}$ | $88.69_{\uparrow1.51}$ | $75.08_{\uparrow4.74}$ | $77.30_{\uparrow5.62}$ | 84.67 |
| SC (ComplexCoT) | ✗ | ✗ | $83.65_{\uparrow1.51}$ | $86.14_{\downarrow0.74}$ | $96.94_{\uparrow3.79}$ | $89.72_{\uparrow2.54}$ | $77.69_{\uparrow7.35}$ | $77.94_{\uparrow6.26}$ | 85.35 |
| Chain | ✓ | ✗ | $82.35_{\uparrow0.21}$ | $85.57_{\uparrow0.17}$ | $94.38_{\uparrow1.23}$ | $83.41_{\downarrow3.77}$ | $70.94_{\uparrow0.60}$ | $80.88_{\uparrow9.20}$ | 82.92 |
| Star | ✓ | ✗ | $80.79_{\downarrow1.35}$ | $85.55_{\uparrow0.15}$ | $93.79_{\downarrow0.64}$ | $88.09_{\uparrow0.91}$ | $68.57_{\downarrow1.77}$ | $75.65_{\downarrow3.97}$ | 82.07 |
| Tree | ✓ | ✗ | $81.89_{\downarrow0.25}$ | $84.56_{\downarrow0.84}$ | $94.60_{\uparrow1.45}$ | $89.25_{\uparrow2.07}$ | $72.84_{\uparrow2.50}$ | $77.38_{\uparrow5.70}$ | 83.42 |
| Complete Graph | ✓ | ✗ | $83.15_{\uparrow1.01}$ | $86.49_{\uparrow1.09}$ | $97.20_{\uparrow4.05}$ | $89.48_{\uparrow2.30}$ | $79.21_{\uparrow8.87}$ | $83.75_{\uparrow12.07}$ | 86.55 |
| Layered Graph | ✓ | ✗ | $78.41_{\downarrow3.73}$ | $85.34_{\downarrow0.06}$ | $95.04_{\uparrow1.89}$ | $88.61_{\uparrow1.43}$ | $73.18_{\uparrow2.84}$ | $80.38_{\uparrow8.70}$ | 83.49 |
| Random Graph | ✓ | ✗ | $83.76_{\uparrow1.62}$ | $86.14_{\uparrow0.74}$ | $95.46_{\uparrow2.31}$ | $85.41_{\downarrow1.77}$ | $74.07_{\uparrow3.73}$ | $82.66_{\uparrow10.98}$ | 84.58 |
| LLM-Blender | ✓ | ✗ | $81.22_{\downarrow0.92}$ | $89.17_{\uparrow3.77}$ | $94.27_{\uparrow1.12}$ | $88.77_{\uparrow1.59}$ | $77.05_{\uparrow6.71}$ | - | 86.10 |
| GPTSwarm | ✓ | ✗ | $\underline{83.98}_{\uparrow1.84}$ | $89.74_{\uparrow4.34}$ | $\mathbf{97.84}_{\uparrow4.69}$ | $86.42_{\downarrow0.76}$ | $78.16_{\uparrow7.82}$ | $88.49_{\uparrow16.81}$ | 86.77 |
| LLM-Debate | ✗ | ✓ | $83.69_{\uparrow1.55}$ | $90.23_{\uparrow4.83}$ | $96.27_{\uparrow3.12}$ | $90.56_{\uparrow3.38}$ | $77.52_{\uparrow7.18}$ | $83.79_{\uparrow12.11}$ | 87.01 |
| PHP | ✗ | ✓ | $83.45_{\uparrow1.31}$ | $92.45_{\uparrow7.05}$ | $96.41_{\uparrow3.26}$ | $90.62_{\uparrow3.44}$ | $76.25_{\uparrow5.91}$ | $82.96_{\uparrow11.28}$ | 87.02 |
| DyLAN | ✗ | ✓ | $80.16_{\downarrow1.98}$ | $88.16_{\uparrow2.76}$ | $94.27_{\uparrow1.12}$ | $87.40_{\uparrow0.22}$ | $74.16_{\uparrow3.82}$ | $\underline{89.70}_{\uparrow18.02}$ | 84.48 |
| `AgentPrune-C` | ✓ | ✓ | $\mathbf{84.72}_{\uparrow2.58}$ | $\underline{95.62}_{\uparrow10.22}$ | $\underline{97.25}_{\uparrow4.10}$ | $\mathbf{91.85}_{\uparrow4.67}$ | $\mathbf{79.47}_{\uparrow9.13}$ | $89.38_{\uparrow15.70}$ | $\mathbf{89.72}$ |
| `AgentPrune-L` | ✓ | ✓ | $83.50_{\uparrow1.36}$ | $93.78_{\uparrow8.38}$ | $96.39_{\uparrow3.24}$ | $89.58_{\uparrow2.40}$ | $78.44_{\uparrow8.10}$ | $88.61_{\uparrow16.93}$ | 88.38 |
| `AgentPrune-R` | ✓ | ✓ | $83.94_{\uparrow1.80}$ | $\mathbf{95.83}_{\uparrow10.43}$ | $96.30_{\uparrow3.15}$ | $\underline{91.68}_{\uparrow4.50}$ | $\underline{78.60}_{\uparrow8.26}$ | $\mathbf{90.30}_{\uparrow18.62}$ | $\underline{89.44}$ |

large number of queries, such as multiple choice answering (Agashe et al., 2023; Qian et al., 2024), typically only one or two dialogue rounds are needed, according to previous practices (Yin et al., 2023). Under such circumstances, optimizing the connectivity for each query independently can be unnecessarily costly. Therefore, we give a *multi-query training paradigm* for `AgentPrune`, which optimizes and prunes the spatial-temporal topology using merely $Q'(Q' \ll Q)$ queries, given a dataset composed of $Q$ queries. See details in Appendix D.

**Cost Analysis** In this section, we quantify the difference in token consumption between `AgentPrune` and the vanilla pipeline. Given a communication graph $\mathcal{G}$ and $K$ dialogue rounds, assuming that the average token count per spatial/temporal/query message is $c_{\mathcal{S}}, c_{\mathcal{T}}, c_q$, respectively, then the total token consumption of the vanilla system is $C_{\mathcal{G}} = K\left[c_{\mathcal{S}}|\mathcal{E}^{\mathcal{S}}| + c_{\mathcal{T}}|\mathcal{E}^{\mathcal{T}}| + C_q|\mathcal{V}|\right]$. The token consumption after applying `AgentPrune` is divided into two stages. The first stage involves $MK'\left[c_{\mathcal{S}}|\mathcal{E}^{\mathcal{S}}| + c_{\mathcal{T}}|\mathcal{E}^{\mathcal{T}}| + C_q|\mathcal{V}|\right]$, while in the second stage, after the topology is fixed, the consumption becomes $(K - K')\left[(1 - p\%) \cdot \left(c_{\mathcal{S}}|\mathcal{E}^{\mathcal{S}}| + c_{\mathcal{T}}|\mathcal{E}^{\mathcal{T}}|\right) + C_q|\mathcal{V}|\right]$. Therefore, the total token savings $\Delta$ achieved by `AgentPrune` can be expressed as:

$$\Delta = \left((1 + p\%)\,K - (M + p\%)\,K'\right)\left(c_{\mathcal{S}}|\mathcal{E}^{\mathcal{S}}| + c_{\mathcal{T}}|\mathcal{E}^{\mathcal{T}}|\right) + (1 - M)K'C_q|\mathcal{V}|. \quad (13)$$

We present the cost analysis of `AgentPrune` in multi-query training in Appendix E. We will empirically evaluate the substantial token savings gained by `AgentPrune` in Section 4.2.

# 4 EXPERIMENTS

In this section, we conduct extensive experiments to answer the following research questions: (**RQ1**) How does `AgentPrune` perform with respect to task completion and token efficiency? (**RQ2**) Can `AgentPrune` reduce the economical cost of existing multi-agent systems without compromising performance? (**RQ3**) Is `AgentPrune` effective in defending against adversarial attacks on agents? (**RQ4**) How sensitive is `AgentPrune` to its key components or parameters?

## 4.1 EXPERIMENTAL SETUP

**Tasks and Benchmarks** In our experiments, we test the performance of `AgentPrune` on three types of reasoning tasks and the corresponding logically challenging benchmarks: (**1**) **General Reasoning**: We opt for MMLU (Hendrycks et al., 2021) dataset; (**2**) **Mathematical Reasoning**: We select GSM8K (Cobbe et al., 2021), MultiArith (Roy & Roth, 2016), SVAMP (Patel et al., 2021) and AQuA (Ling et al., 2017) to verify the mathematical reasoning capacity; (**3**) **Code Generation**: We use the HumanEval (Chen et al., 2021a) to test the function-level code generation ability.

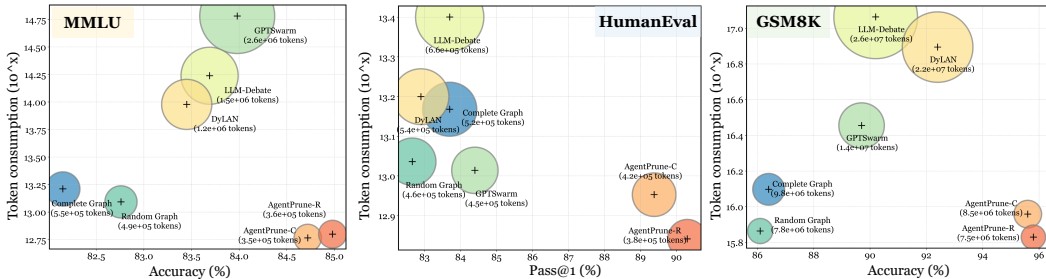

**Figure 5: Visualization of performance and prompt token consumption.** This scatter plot illustrates the performance metrics and **prompt token consumption** of different multi-agent communication topologies across MMLU, HumanEval, and GSM8K. The diameter of each point is proportional to its y-axis value.

**Baselines** We compare `AgentPrune` with three series of multi-agent communication paradigms, namely: **(1) Single agent execution methods**, including Chain-of-Thought prompting (CoT; Wei et al. (2022)), (2) Complexity-based prompting (ComplexCoT; Fu et al. (2022)), and (3) Self-Consistency (SC; Wang et al. (2023b)); **(2) Spatial communication methods**, including chain, tree, star, complete graph, layered graph and random graph[1] from MacNet (Qian et al., 2024), LLM-Blender (Jiang et al., 2023), and GPTSwarm (Zhuge et al., 2024); **(3) Temporal communication methods**, including PHP (Zheng et al., 2023), LLM-Debate (Du et al., 2023b), DyLAN (Liu et al., 2023b). Detailed introductions and implementations of the baselines are in Appendix G.1.

**Implementation Details** We accessed the GPT models via the OpenAI API, and mainly tested on `gpt-3.5-turbo-0301` (`gpt-3.5`) and `gpt-4-1106-preview` (`gpt-4`). We set the temperature at 1 during the generation. We set the dialogue round $K = 2$ for mathematical and general reasoning tasks, and $K = 4$ for code generation tasks. For multi-query settings, we vary $Q' \in \{5, 10, 20\}$ and fix $M = 10$. We generate different agent profiles using `gpt-4`. The pruning ratio is chosen among $\{50\%, 30\%\}$. More experimental details are in Appendix G.2.

## 4.2 Performance & Cost Comparison (RQ1)

To evaluate whether `AgentPrune` achieves a dual benefit of token savings and task completion, we integrate it with three predefined spatial communication topologies: the complete graph, layered graph, and random graph, denoted as `AgentPrune`-C, `AgentPrune`-L, and `AgentPrune`-R, respectively. For the temporal communication topology, we consistently employ the fully connected LLM-Debate-style structure. Tables 1 and 2 presents a performance comparison of various communication paradigms within *five* `gpt-4`-based multi-agent systems, and Figures 5 and 17 to 19 visualizes the performance and token cost of different methods. Our observations (Obs.) are as follows: **Obs.❶ Not**

**Table 2: Performance on the HumanEval Benckmark** with more baselines. The best and runner-up results are **bolded** and underlined, respectively.

| Method | Pass@1 | Δ |
|---|---|---|
| Vanilla | 71.68 | - |
| AutoGen [2023] | 85.41 | ↑11.97 |
| Reflexion [2023] | **91.40** | **↑19.72** |
| CodeT+Parsel [2023a] | 85.10 | ↑13.42 |
| MetaGPT [2023] | 85.90 | ↑14.22 |
| ANPL [2024] | 86.60 | ↑14.92 |
| `AgentPrune`-C | 89.38 | ↑17.70 |
| `AgentPrune`-R | 90.30 | ↑18.62 |

**all multi-agent topologies consistently deliver collective intelligence.** As illustrated in Table 1, certain topologies, such as star/tree structures, fail to consistently improve performance for multi-agent systems, even resulting in performance drops of $0.17\% \sim 3.97\%$. In contrast, single-agent prompting methods like CoT or ComplexCoT demonstrate much more stable and significant improvements. **Obs.❷ The high performance of existing multi-agent systems comes at a substantial economical cost.** From Table 1, we observe that the top-performing baselines, GPTSwarm and DyLAN, achieve *pass@1* improvements of $16.81\%$ and $18.02\%$ on HumanEval, respectively; however, this is accompanied by extremely high economic costs. As shown in Figure 5, the prompt token consumption of GPTSwarm and DyLAN is $2.4 \sim 5.3\times$ that of the random graph structure. **Obs.❸ `AgentPrune` achieves a double win in economic savings and utility.** Among the three variants, `AgentPrune`-R delivers consistently impressive performance, achieving $90.3\%$ on HumanEval and $95.8\%$ on GSM8K. Importantly, this performance does not come at a high token cost: on both HumanEval and GSM8K, the token consumption of `AgentPrune` is less than $40\%$ that of DyLAN. Overall, `AgentPrune` excels in both task completion and token efficiency.

---
[1]Detailed explanations of these topologies are placed in Appendix F.

**Table 3: Performance and cost comparison before/after combining `AgentPrune`.** We evaluated the performance and economical cost of `AgentPrune` in conjunction with two classic multi-agent systems, under a **five** `gpt-4`-based setting. "# Prompt tokens" refers to the total number of tokens input, while "# Completion tokens" accounts for the total number of tokens output by the API.

| Dataset | Method | Performance | # Prompt Tokens | # Completion Tokens | Cost (USD) |
|---|---|---|---|---|---|
| MMLU | AutoGen | 82.13 | $486,034$ | $89,224$ | \$7.537 |
| | +`AgentPrune` | 82.78($\uparrow$ 0.65) | $349,583(71.9\%)$ | $86,582$ | \$6.093($80.8\%$) |
| HumanEval | AutoGen | 85.41 | $492,273$ | $130,196$ | \$8.828 |
| | +`AgentPrune` | 86.65($\uparrow$ 1.24) | $315,105(64.0\%)$ | $139,714$ | \$7.342($83.1\%$) |
| GSM8K | AutoGen | 90.06 | $4,327,740$ | $998,042$ | \$73.21 |
| | +`AgentPrune` | 92.85($\uparrow$ 2.79) | $3,791,251(59.9\%)$ | $1,156,884$ | \$59.60($81.4\%$) |
| MMLU | GPTSwarm | 83.98 | $3,055,230$ | $569,124$ | \$47.60 |
| | +`AgentPrune` | 83.05($\downarrow$ 0.93) | $990,312(32.4\%)$ | $439,551$ | \$23.05($48.4\%$) |
| HumanEval | GPTSwarm | 88.49 | $2,736,136$ | $1,004,616$ | \$57.49 |
| | +`AgentPrune` | 88.96($\uparrow$ 0.47) | $745,617(27.2\%)$ | $745,926$ | \$29.80($51.8\%$) |
| GSM8K | GPTSwarm | 89.74 | $14,005,945$ | $3,156,916$ | \$234.76 |
| | +`AgentPrune` | 90.58($\uparrow$ 0.84) | $3,526,035(39.4\%)$ | $730,552$ | \$57.17($24.3\%$) |

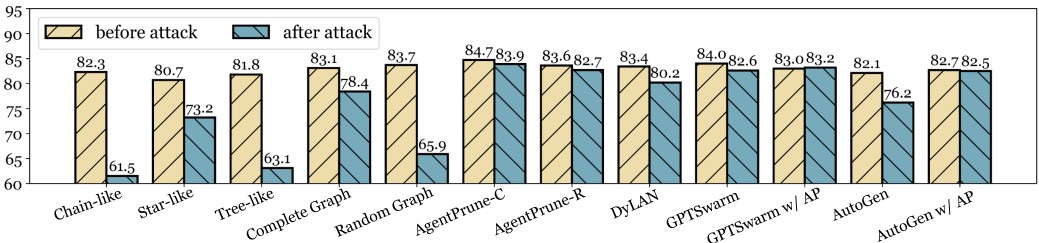

**Figure 6: Performance under adversarial attack.** We compare the accuracy (%) of various multi-agent frameworks before and after *prompt attacks* on MMLU. "w/ AP" indicates the integration with `AgentPrune`.

## 4.3 PLUG-IN INTO EXISTING FRAMEWORKS (RQ2)

As a plug-in, `AgentPrune` can be seamlessly combined with mainstream multi-agent pipelines, effectively reducing the economic costs associated with LLM token throughput while maintaining the original performance levels. To validate our argument, we combined `AgentPrune` with two representative LLM-MA frameworks, AutoGen and GPTSwarm. With the results presented in Table 3 and Table 5, we offer the following two key observations: **Obs.❹ Scaling multi-agent collaboration is costly.** Comparing Table 3 and Table 5, we observe that for the GPTSwarm on the GSM8K dataset, optimizing a three-agent system incurs a cost of \$97.23, while the expense for a five-agent system skyrockets to \$234.76, with the total token count reaching $1.7e+7$. AutoGen, on the other hand, has relatively lower costs because it does not involve the iterative optimization of the communication topology as extensively as GPTSwarm (Zhuge et al., 2024). Nevertheless, it still requires \$73.21 on the GSM8K benchmark, which comprises up to 8.5K data entries. **Obs.❺ `AgentPrune` is an economically friendly assistant.** When applied to HumanEval+AutoGen, `AgentPrune` achieves a 36% reduction in prompt tokens and saves \$1.486. In tasks with larger datasets, the economic savings become even more pronounced: on GSM8K+GPTSwarm, `AgentPrune` reduces 60.6% of the prompt token consumptions and saves a cost of up to \$177.58, with even a performance increase of 0.84%. Overall, `AgentPrune` serves as a token-efficient plug-in, effectively fostering the development of larger and more cost-effective multi-agent systems.

## 4.4 ROBUSTNESS VERIFICATION (RQ3)

`AgentPrune` can not only eliminate *unnecessary* communications but also remove *malicious* messages. To validate this, we design two types of adversarial attacks for the multi-agent frameworks: the **agent prompt attack** and **agent replacement attack**. The former attacks the role prompts of the agents, while the latter attacks the LLM's generation process, with detailed implementation elaborated in Appendix G.4. We observe that: **Obs.❻ Existing LLM-MA frameworks often lack adversarial robustness.** Despite variations in performance, most frameworks experience significant declines when subjected to both types of attacks. As shown in Figures 6 and 20, the chain-like structure suffers a performance drop of up to 20.8% due to its oversimplistic topology. AutoGen and DyLAN similarly experience accuracy declines ranging from 3.2% to 6.2%. **Obs.❼ `AgentPrune` significantly enhances multi-agent robustness.** Figure 6 demonstrates that combining `AgentPrune`

with a complete graph not only improves performance ($83.1\% \rightarrow 84.7\%$), but also increases robustness under agent prompt attacks ($78.4\% \rightarrow 83.9\%$). Additionally, `AgentPrune` successfully boosts the robustness of DyLAN and AutoGen by up to $6.3\%$. The impact of `AgentPrune` on GPTSwarm is relatively marginal, due to its inherent defenses against adversarial agents. Overall, `AgentPrune` serves as an easy-to-use enhancer for multi-agent robustness.

### 4.5 EXPERIMENTAL ANALYSIS

**Ablation Study**    We ablate agent profiling and low-rank regularization in `AgentPrune`, with details presented in Table 7 and Appendix H.4.1. Our key finding is that (1) the utility of agent profiling varies across different datasets, demonstrating a more pronounced effect on general reasoning and code generation tasks, while being relatively less significant in math reasoning; (2) low-rank sparsity consistently facilitates the optimization of the communication topology.

**Sensitivity Analysis *and* Case Study**    We present the parameter sensitivity analysis concerning three hyper-parameters, $|\mathcal{V}|$, $Q'$ and $p\%$ in Appendix H.4.2, and provide extensive visualizations on `AgentPrune`'s pruning process and optimized communication structure in Appendix I.

## 5    RELATED WORK

**LLM-agent Collaboration**    Collaboration between multiple LLM-based agents has emerged as a promising approach to enhance the capabilities of individual LLMs (Du et al., 2023b; Liang et al., 2023; Wang et al., 2023c; Zhang et al., 2024c; Niu et al., 2025). As stated in Section 1, current multi-agent communication methods can be categorized into two types: ❶ **Intra-dialogue (spatial) communication** focuses on how different agents exchange messages within a single dialogue round. Common structures include (1) *Direct output*, where functioning agents do not communicate with each other, adopted by systems like LATM (Zhang et al., 2023a), LLM-Debate (Du et al., 2023b); (2) *Chain*, employed by ChatDev (Qian et al., 2023), MetaGPT (Hong et al., 2023) and L2MAC (Holt et al., 2024); (3) *Tree*, where an administrative agent (usually referred to as commander, manager, *etc.*) controls subordinate agents, adopted by AutoGen (Wu et al., 2023), SecurityBot (Yan et al., 2024), and MiniGrid (Zhou et al., 2023); and (4) *Graph,* employed by LLM-Blender (Jiang et al., 2023), ChatEval (Chan et al., 2023), MacNet (Qian et al., 2024) and GPTSwarm (Zhuge et al., 2024); ❷ **Inter-dialogue (temporal) communication** focuses on how information is passed between different rounds of utterances. Common topologies include (1) *Full transmission*, where every agent receives the utterances of all agents from the previous round, as used by LLM-Debate (Du et al., 2023b); (2) *Partial transmission*, where some responses are filtered through scoring or rating mechanisms, adopted by PHP (Zheng et al., 2023) and DyLAN (Liu et al., 2023b); (3) *Summarization*, where dialogue history is compressed and summarized for the next round of communication, as seen in Reflexion (Shinn et al., 2023), ICL-AIF (Fu et al., 2023), AgentVerse (Chen et al., 2023a), CoMM (Chen et al., 2024b), Corex (Sun et al., 2023), and MAD (Liang et al., 2023).

**Agents as Graphs**    Learning to facilitate communication via learning graph connectivity is a long-standing and viable approach to enhance multi-agent cooperation (Pesce & Montana, 2023; Hu et al., 2024). In the pre-LLM era, numerous efforts explored optimal communication graph structures for reinforcement learning-based multi-agents with graph diffusion (Pesce & Montana, 2023), weighted GNN (Liu et al., 2022), or transformers (Hu et al., 2024). In the emerging wave of LLM-powered agents, attempts that leverage graphs for modeling agent-agent interaction also exist: ChatEval (Chan et al., 2023) and AutoGen (Wu et al., 2023) implicitly adopt graph structures to describe "simultaneous talk", and STOP (Zelikman et al., 2023b) and DSPy (Khattab et al., 2023) optimize both the prompts and the inference structure together. MacNet (Qian et al., 2024) and GPTSwarm (Zhuge et al., 2024) model agent communication via directed acyclic graphs (DAG). However, none of these approaches simultaneously optimize both intra- and inter-dialogue communication structures, and they often result in even increased token consumption.

## 6    CONCLUSION

This paper makes the first attempt towards a high-performance and token-efficient LLM-powered multi-agent system. We propose an economical, simple, and robust multi-agent communication pipeline, termed `AgentPrune`, which can be harmoniously embedded into mainstream multi-agent frameworks while effectively pruning the *communication redundancy* that we have identified and defined. `AgentPrune` achieves performance comparable to, or even superior to, the original systems with significantly smaller token throughput and economic costs. We believe that `AgentPrune` can facilitate the advancement toward larger-scale collective intelligence.

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

## A    DAG SAMPLING FUNCTION

---

**Algorithm 2:** Sample DAG from spatial communication graph

---

**Input:** Spatial communication graph $\mathcal{G}^{\mathcal{S}} = \{\mathcal{V}, \mathcal{E}^{\mathcal{S}}\}$
**Output:** A directed acyclic graph $\hat{\mathcal{G}}^{\mathcal{S}}$
$\hat{\mathcal{G}} \leftarrow \mathcal{G}^{\mathcal{S}}$ // Create a copy of $\mathcal{G}^{\mathcal{S}}$
**while** not is_acyclic $(\hat{\mathcal{G}})$ **do**
    /* Use DFS to locate cycle                                              */
    cycle $\leftarrow$ find_cycle $(\hat{\mathcal{G}})$
    e $\leftarrow$ random_choice (cycle) // Randomly select an edge from the cycle
    $\hat{\mathcal{G}} \leftarrow \hat{\mathcal{G}}$.remove_edge (e) // Remove the selected edge from $G'$
**return** $\hat{\mathcal{G}}^{\mathcal{S}} \leftarrow \hat{\mathcal{G}}$

---

## B    NOTATIONS

We conclude the commonly used notations in Table 4 for reference.

**Table 4:** The notations that are commonly used throughout the manuscript.

| Notation | Definition |
|---|---|
| $\mathcal{G} = (\mathcal{V}, \mathcal{E}) = \{\mathcal{G}^{\mathcal{S}}, \mathcal{G}^{\mathcal{T}}\}$ | the spatial-temporal communication graph |
| $\mathcal{V} = \{v_1, v_2, \cdots, v_{|\mathcal{V}|}\}$ | the set of nodes (agents) |
| $\mathcal{E} = \mathcal{E}^{\mathcal{S}} \cup \mathcal{E}^{\mathcal{T}}$ | the overall edge set |
| $\mathcal{E}^{\mathcal{S}} \subseteq \mathcal{V}^{(t)} \times \mathcal{V}^{(t)}$ | the spatial edge set |
| $\mathcal{E}^{\mathcal{T}} \subseteq \mathcal{V}^{(t-1)} \times \mathcal{V}^{(t)}$ | the temporal edge set |
| $\texttt{Base}_i$ | the LLM base utilized by agent $v_i$ |
| $\texttt{Role}_i$ | the predefined responsibilities or roles of agent $v_i$ |
| $\texttt{State}_i$ | the state of agent $v_i$ |
| $\texttt{Plugins}_i = \{\texttt{F}_j, \texttt{C}_j\}_{j=1}^{P}$ | the plugins available to agent $v_i$ |
| $e_{ij}^{\mathcal{S}} = (\mathbf{M}_{ij}, \mathbf{O}_{ij})$ | the spatial edge from $v_i$ to $v_j$ |
| $e_{ij}^{\mathcal{T}}$ | the temporal edge from $v_i$ to $v_j$ |
| $\mathcal{N}^{\mathcal{T}}(v_i) = \{v_j \mid (j, i) \in \mathcal{E}^{\mathcal{T}}\}$ | the temporal (in-)neighbors of $v_i$ |
| $\mathcal{N}^{\mathcal{S}}(v_i) = \{v_j \mid (j, i) \in \mathcal{E}^{\mathcal{S}}\}$ | the spatial (in-)neighbors of $v_i$ |
| $\mathbf{M}_i^{(t)}$ | the rationale or answers provided by $v_i$ at the $t$-th epoch |
| $\mathcal{G}^{\text{sub}} = (\mathcal{V}, \mathcal{E}' \cup \mathcal{E}'')$ | the sparsified communication topology |
| $\mathbf{S}^{\mathcal{S}}, \mathbf{S}^{\mathcal{T}} \in \mathbb{R}^{|\mathcal{V}| \times |\mathcal{V}|}$ | the spatial and temporal graph masks |
| $\mathbf{A}^{\mathcal{S}} \in \{0, 1\}^{|\mathcal{V}| \times |\mathcal{V}|}$ | the predefined spatial communication topology |
| $\mathbf{A}^{\mathcal{T}} \in \{0, 1\}^{|\mathcal{V}| \times |\mathcal{V}|}$ | the predefined temporal communication topology |
| $\tilde{\mathcal{G}}^{\mathcal{S}}$ | the parameterized spatial graph |
| $\hat{\mathcal{G}}^{\mathcal{S}}$ | the parameterized spatial graph after DAG sampling |
| $\tilde{\mathcal{G}}^{\mathcal{T}}$ | the parameterized temporal graph |
| $\phi(\cdot)$ | the utility evaluation function |
| $\mathbf{B}^{\mathcal{S}} \in \{0, 1\}^{|\mathcal{V}| \times |\mathcal{V}|}$ | the obtained binary spatial mask |
| $\mathbf{B}^{\mathcal{T}} \in \{0, 1\}^{|\mathcal{V}| \times |\mathcal{V}|}$ | the obtained binary temporal mask |
| $K$ | the total number of dialogue rounds |
| $K'$ | the dialogue round after which pruning takes place |
| $Q$ | the total number of queries |
| $Q'$ | the number of queries after which pruning takes place |

## C  ALGORITHM WORKFLOW

We conclude the overall algorithm workflow of `AgentPrune` in algorithm 3.

## D  MULTI-QUERY TRAINING OF `AgentPrune`

For complex tasks such as repository-level code generation (Qian et al., 2023), multi-turn dialogues ($K > 5$) are often essential. In such cases, utilizing $K' \in \{1, 2\}$ rounds to optimize the topology and subsequently continue the dialogue for $K - K'$ rounds is reasonable. However, for simpler tasks that involve numerous queries, such as multiple-choice answering (Agashe et al., 2023) or basic mathematical problems (Cobbe et al., 2021), previous studies (Yin et al., 2023; Qian et al., 2024) suggest that typically only 1 to 2 dialogue rounds are needed. In this context, prior dialogue-level optimization is no longer applicable. To better adapt `AgentPrune` to such circumstances, we propose a query-level optimization paradigm for `AgentPrune`.

Given a benchmark consisting of $Q$ queries, any LLM-MA framework processes these $Q$ queries sequentially to provide solutions one by one. We utilize the initial $Q'(Q' << Q)$ queries as a "training phase," collaboratively optimizing the spatio-temporal communication topology while leveraging multiple agents for reasoning and evaluation. Following this, we perform one-shot pruning as described in Equation (12). The fixed topology $\mathcal{G}^{\text{sub}}$ is then employed for the reasoning and evaluation of the remaining $(Q - Q')$ queries. We also refer to this approach as **query-level optimization**, in contrast to the **dialogue-level optimization** discussed in the main text. The distinction between the two lies in their focus: the latter concentrates on resolving a single query by utilizing several initial

---

**Algorithm 3:** Execution pipeline of LLM-MA systems combined with `AgentPrune`.

---

**Input:** Query $q$, Communication graph $\mathcal{G} = \{\mathcal{G}^{\mathcal{S}}, \mathcal{G}^{\mathcal{T}}\}$, Maximum rounds of iterations $K$,
    Rounds for optimization $K'$, Initial masks $\mathbf{S}^{\mathcal{S}}, \mathbf{S}^{\mathcal{T}}$

1 $\mathbf{A}(\{\tilde{\mathcal{G}}^{\mathcal{S}}, \tilde{\mathcal{G}}^{\mathcal{T}}\}) \leftarrow \{\mathbf{A}^{\mathcal{S}} \odot \mathbf{S}^{\mathcal{S}}, \mathbf{A}^{\mathcal{T}} \odot \mathbf{S}^{\mathcal{T}}\}$
   /* Optimizing spatial-temporal communication topology                    */
2 **for** iteration $t \leftarrow 1$ **to** $K'$ **do**
3     $\hat{\mathcal{G}}^{\mathcal{S}} = (\mathcal{V}, \mathcal{E}^{\mathcal{S}} \cup \mathcal{E}^{\mathcal{T}}) \leftarrow \texttt{DAGSampling}(\tilde{\mathcal{G}}^{\mathcal{S}})$
4     **for** $v_i$ in $\texttt{TopologicalSort}(\mathcal{V})$ **do**
5        Obtain temporal (in-)neighbors $\mathcal{N}^{\mathcal{T}}(v_i) \leftarrow \{v_j \mid (j, i) \in \mathcal{E}^{\mathcal{T}}\}$
6        Obtain spatial (in-)neighbors $\mathcal{N}^{\mathcal{S}}(v_i) \leftarrow \{v_j \mid (j, i) \in \mathcal{E}^{\mathcal{S}}\}$
7        $\boldsymbol{m}^{\mathcal{T}} \leftarrow \{\mathbf{M}_j \mid v_j \in \mathcal{N}^{\mathcal{T}}(v_i)\}, \boldsymbol{m}^{\mathcal{S}} \leftarrow \{\mathbf{M}_{ij} \mid v_j \in \mathcal{N}^{\mathcal{S}}(v_i)\}$
8        $\mathbf{M}_i^{(t)} \sim \mathcal{P}_\theta(\mathbf{M}_i \mid q, \texttt{Role}_i^{(t)}, \texttt{State}_i^{(t)}, \boldsymbol{m}^{\mathcal{T}}, \boldsymbol{m}^{\mathcal{S}})$
9     **end**
10     $a^{(t)} \leftarrow \texttt{AggregateSolution}(\mathbf{M}_1^{(t)}, \mathbf{M}_2^{(t)}, \cdots, \mathbf{M}_{|\mathcal{V}|}^{(t)})$
11     Update $\mathbf{S}^{\mathcal{S}}, \mathbf{S}^{\mathcal{T}}$ according to Equation (8)
12 **end**
   /* One-shot pruning spatial-temporal communication topology               */
13 $\mathbf{B}^{\mathcal{S}} = \mathbb{I}(\mathbf{A}^{\mathcal{S}} \neq 0 \wedge \text{TopK}(\mathbf{S}^{\mathcal{S}}, |\mathbf{A}^{\mathcal{S}}| \times (1 - p\%)))$
14 $\mathbf{B}^{\mathcal{T}} = \mathbb{I}(\mathbf{A}^{\mathcal{T}} \neq 0 \wedge \text{TopK}(\mathbf{S}^{\mathcal{T}}, |\mathbf{A}^{\mathcal{T}}| \times (1 - p\%)))$
15 Obtain $\mathcal{G}^{\text{sub}}$, where $\mathbf{A}(\mathcal{G}^{\text{sub}}) = \{\mathbf{A}^{\mathcal{S}} \odot \mathbf{B}^{\mathcal{S}}, \mathbf{A}^{\mathcal{T}} \odot \mathbf{B}^{\mathcal{T}}\}$
   /* Fixing the topology for subsequent iterations                          */
16 **for** iteration $t \leftarrow K'$ **to** $K$ **do**
17     Use $\mathcal{G}^{\text{sub}}$ for multi-agent dialogues as in Algorithm 1
18 **end**
19 **return** $a^{(t)}$ as the final solution

---

utterances to derive the topology, while the former considers the entire benchmark, employing a few early queries to inform the topology.

## E COST ANALYSIS

In Section 3.4, we present a token-saving analysis in a single-query setting. In this section, we provide a cost analysis for `AgentPrune` in a multi-query optimization context. Given a communication graph $\mathcal{G}$ and a benchmark with $Q$ queries, we assume that the LLM-MA framework iterates for $K$ dialogue rounds for each query. Furthermore, we denote the average token count per spatial, temporal, and query message as $c_{\mathcal{S}}, c_{\mathcal{T}}$, and $c_q$, respectively. Hence, the total token consumption of the vanilla system can be expressed as:

$$C_{\mathcal{G}} = QK \left[ c_{\mathcal{S}} |\mathcal{E}^{\mathcal{S}}| + c_{\mathcal{T}} |\mathcal{E}^{\mathcal{T}}| + C_q |\mathcal{V}| \right] \tag{14}$$

When utilizing `AgentPrune`, the LLM-MA framework processes the initial $Q'$ queries while simultaneously optimizing the spatial-temporal connectivity. The token cost for this phase is:

$$MQ'K \left[ c_{\mathcal{S}} |\mathcal{E}^{\mathcal{S}}| + c_{\mathcal{T}} |\mathcal{E}^{\mathcal{T}}| + C_q |\mathcal{V}| \right]. \tag{15}$$

After pruning $\mathcal{G}^{\mathcal{S}}$ and $\mathcal{G}^{\mathcal{T}}$, we use the obtained $\mathcal{G}^{\text{sub}}$ to solve the remaining $Q - Q'$ queries, with a cost of:

$$(Q - Q')K \left[ (1 - p\%) \cdot (c_{\mathcal{S}} |\mathcal{E}^{\mathcal{S}}| + c_{\mathcal{T}} |\mathcal{E}^{\mathcal{T}}|) + C_q |\mathcal{V}| \right]. \tag{16}$$

Overall, the token savings of `AgentPrune` in a multi-query setting can be expressed as:

$$\Delta = (p\% \cdot Q + (1 - p\% - M)Q')K \left( c_{\mathcal{S}} |\mathcal{E}^{\mathcal{S}}| + c_{\mathcal{T}} |\mathcal{E}^{\mathcal{T}}| \right) + (1 - M)Q'KC_q |\mathcal{V}| \tag{17}$$

## F EXISTING SPATIAL COMMUNICATION TOPOLOGIES

In this section, we introduce several existing spatial communication topologies, including chain, tree, star, complete graph, layered graph, random graph, and LLM-Blender.

## F.1 CHAIN STRUCTURE

The chain structure (in Figure 7) is one of the most widely utilized communication architectures in contemporary multi-agent systems, as demonstrated by its application in ChatDev (Qian et al., 2023), MetaGPT (Hong et al., 2023), and L2MAC (Holt et al., 2024). In this architecture, the first agent receives input from the user, transforms it into new instruction, and subsequently forwards it to the next agent. For instance, in MetaGPT, user instructions are initially sent to the first agent, termed the "product manager," with information progressively relayed to subsequent agents, such as the architect agent, engineer agent, and QA engineer agent. Generally, the final agent in the chain provides a solution to the user's request.

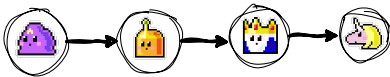

**Figure 7:** Demonstration of **chain** structure as spatial communication topology.

## F.2 TREE STRUCTURE

In a tree-like multi-agent pipeline, as shown in Figure 8, an administrative agent (commonly referred to as a teacher, commander, manager, etc.) oversees subordinate agents, which typically have distinct responsibilities. Ultimately, these subordinate agents submit their outputs to the administrative agent for final evaluation. Notable works employing this structure include AutoGen (Wu et al., 2023), SecurityBot (Yan et al., 2024), and MiniGrid (Zhou et al., 2023). For instance, in AutoGen (A4: Multi-Agent Coding), there exists a Commander agent alongside a Safeguard agent. The Writer is responsible for crafting the code and its interpretation, the Safeguard ensures safety (e.g., preventing information leaks and avoiding malicious code), and the Commander executes the code.

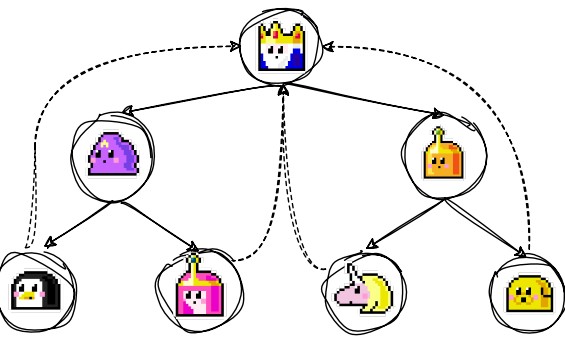

**Figure 8:** Demonstration of **tree** structure as spatial communication topology.

## F.3 STAR STRUCTURE

The star structure resembles the tree structure and can essentially be viewed as a tree with a depth of two. When utilizing the star configuration for spatial communication, the central administrative agent receives queries from the user and dispatches instructions to subordinate agents. Upon completing their tasks using various tools, these subordinate agents return all outputs to the administrative agent, which then compiles a final summary, as illustrated in Figure 9.

## F.4 COMPLETE GRAPH STRUCTURE

In the main text, we refer to the structure shown in Figure 10 as a complete graph. However, this complete graph differs from the traditional definition, *i.e.*, an undirected graph where each vertex is connected to every other vertex. Instead, it is a directed graph that would represent a complete graph if converted to an undirected form. This distinction is necessary because the execution of the

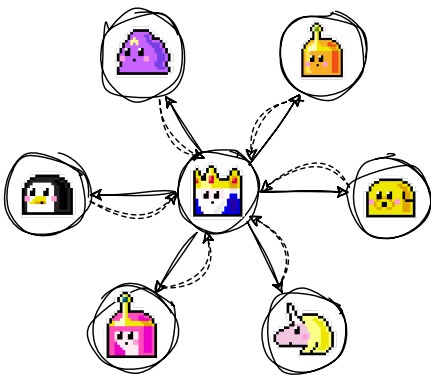

**Figure 9:** Demonstration of **star** structure as spatial communication topology.

multi-agent system relies on topological ordering (Qian et al., 2024; Zhuge et al., 2024), requiring the spatial communication topology to be a DAG. In MacNet (Qian et al., 2024), this structure is also referred to as a "Mesh graph." After executing in the order determined by topological sorting, the final agent summarizes the dialogue and provides a concluding output or reflection.

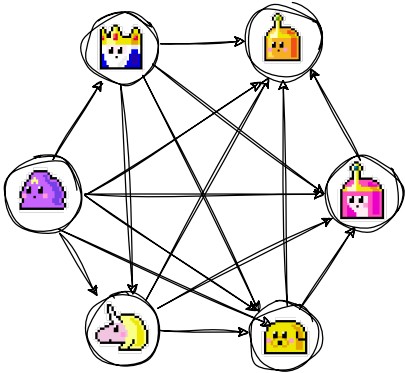

**Figure 10:** Demonstration of **complete graph** structure as spatial communication topology.

### F.5  LAYERED GRAPH STRUCTURE

A layered graph, proposed by Qian et al. (2024), refers to the structure illustrated in Figure 11, resembling a stacked configuration similar to a multilayer perceptron (MLP). The query is first provided to all agents in the first layer, whose outputs serve as prompts that, along with the query, are then fed to the agents in the second layer. The final layer consists of a single agent that receives information from the previous layer and generates the ultimate solution.

### F.6  RANDOM GRAPH STRUCTURE

A random graph refers to a sparse graph randomly sampled from a complete graph, as illustrated in Figure 12. Irregular random structures have been shown to outperform regular fully connected structures (Qian et al., 2024), which is attributed to the presence of random edge connections. Analogous to social networks, these connections can link "unacquainted" agents through direct shortcuts, transforming them into adjacent "acquaintances" and implicitly reducing the average path length, thereby exhibiting small-world characteristics.

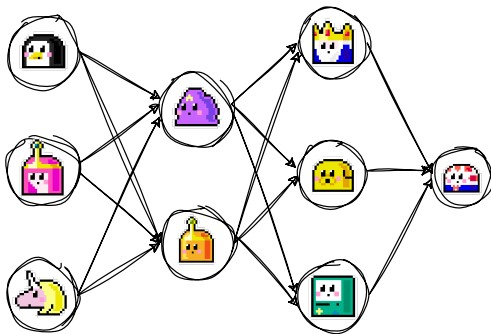

**Figure 11:** Demonstration of **layered graph** structure as spatial communication topology.

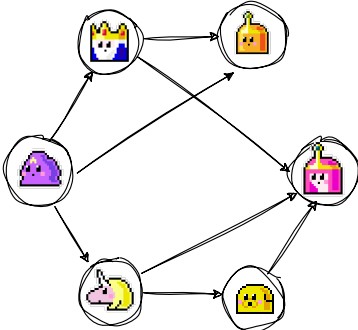

**Figure 12:** Demonstration of **random graph** structure as spatial communication topology.

### F.7 LLM-BLENDER STRUCTURE

The structure of LLM-Blender is relatively straightforward. It feeds a query to multiple LLM-powered agents from different sources, employing a PairRanker mechanism to score each agent's output. The top $K$ responses are then selected and merged using an LLM called GenFuser, as illustrated in Figure 13.

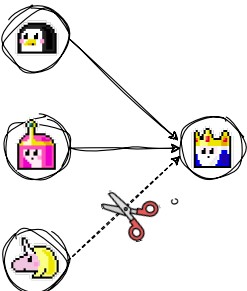

**Figure 13:** Demonstration of **LLM-Blender** structure as spatial communication topology.

## G EXPERIMENTAL DETAILS

### G.1 BASELINES

In this section, we will provide a detailed overview of the various baselines mentioned in Section 4.1 and their adaptations for our evaluation.

### G.1.1 SPATIAL COMMUNICATION BASELINES

The methods described below fall under the category of spatial communication, meaning they are designed to regulate how different agents interact and exchange information within the same dialogue round. Unless stated otherwise, we do not employ explicit inter-dialogue message passing and limit iterations to two rounds (*i.e.*, $K = 2$).

**Chain**  Given the diversity of benchmarks we employed (including mathematical reasoning, code generation, etc.), we organized distinct agent pools for different categories of reasoning tasks, as detailed in Appendix G.3. In addition to personalized prompts for each agent, we also designed a universal prompt template applicable to all agents for generating outputs. Taking the MMLU benchmark as an example:

> **Prompt Template for Agents on the Chain**
>
> ```
> I will ask you a question and 4 answers enumerated as A, B, C
> and D.
> Only one answer out of the offered 4 is correct.
> You must choose the correct answer to the question from your
> perspective.
> Using the reasoning from other agents as additional advice
> with critical thinking, can you give an updated answer?
> You are strictly prohibited from imitating the analysis
> process of other agents.
> Your reply must be less than 100 words but include your
> answer and a brief step-by-step analysis of the question.
> The first line of your reply must contain only one letter(for
> example A, B, C or D)
> ```

We utilize the output from the final agent as the decision for the entire system.

**Star**  When employing the star structure, as described in Appendix F.3, we designate one agent as the administrative agent and utilize the other agents as subordinates. The administrative agent ultimately collects outputs from the subordinate agents to make a decision. During the final decision-making process, the prompt is as follows:

> **Prompt Template for Decision Making**
>
> ```
> You are the top decision-maker and are good at analyzing
> and summarizing other people's opinions, finding errors and
> giving final answers.
> ```

**Tree**  We reduce the tree structure to a **binary tree** and sequentially assign agents based on the binary tree indexing, depending on the number of agents. The outputs from all non-root nodes are ultimately relayed to the root node's agent for the final decision, with the prompt template remaining the same as the Star structure.

**Complete Graph**  We employ the structure outlined in Figure 10 and execute the input/output for each agent node through topological sorting. Before performing topological sorting, it may be necessary to apply the DAG sampling method discussed in Section 3.1 to ensure that the spatial communication graph is a DAG. Given the relative complexity of the graph structure, which does not possess a straightforward core agent like a chain or tree, we introduce an additional summarizer node to which all other nodes direct their outputs. Naturally, this summarizer node is executed last in the topological order, positioning it as the final decision-making expert.

**Layered Graph**  As discussed in Appendix F.5, we arrange the agents in an MLP-like layered structure, ensuring that the final layer contains only one agent. This agent receives outputs from all agents in the preceding layer and produces the final solution.

**Random Graph**    The implementation of a random graph is similar to that of a complete graph. It also begins with DAG sampling, followed by execution through topological sorting, and concludes with a summarizer agent that generates the overall response. We generate the random graph with a connection probability set as 0.5.

**LLM-Blender**    LLM-Blender (Jiang et al., 2023) was originally designed to consolidate responses from various LLM architectures. In this context, we treat it as a spatial message-passing paradigm, standardizing all agents to utilize either `gpt-3.5` or `gpt-4`, with the final output from GenFuser serving as the solution. Notably, LLM-Blender is specifically tailored for single-turn dialogues; thus, we do not employ multi-turn dialogues in conjunction with LLM-Blender.

**GPTSwarm**    GPTSwarm (Zhuge et al., 2024) conceptualizes the connections among all agents as a parameterized, dense adjacency matrix, which is continuously optimized to enhance collaborative performance. In the original paper, distinct internal structures were customized for different agents, such as configuring a specific agent to first receive a query, followed by performing `FileAnalyze` and `WebSearch`, and ultimately outputting the results. To ensure a fair comparison, we did not utilize such configurations in Section 4; instead, we assigned each agent different profiles, similar to the other structures mentioned above, along with possible external tools like a Python compiler or Wikipedia searcher. Our implementation is based on the resources available at `https://github.com/metauto-ai/GPTSwarm`. **Important Note**: in the originally open-sourced code, the multi-agent collaboration for MMLU dataset only transmitted the options A/B/C/D during dialogues, without including the content of agents' reasoning process, which is not consistent with the description in Section 2.2 of their manuscript. To ensure a fair comparison and maintain consistency with the original description, we modified their code to transmit both the choices and the reasoning process.

### G.1.2    TEMPORAL COMMUNICATION BASELINES

**LLM-Debate**    LLM-Debate (Du et al., 2023b) is designed for multiple agents to engage in a debate, where in each round, every agent receives the outputs of all agents from the previous round before making their own statements. Consequently, it essentially forms a fully connected temporal communication graph, as illustrated in Figure 14.

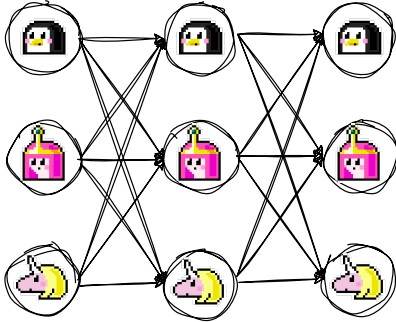

**Figure 14:** Demonstration of **LLM-Debate** structure as temporal communication topology.

**PHP**    PHP (Zheng et al., 2023) progressively improves prompts by utilizing the entirety of historical dialogue, offering potential "hint" prompts. In this context, we adapt this setting to the multi-agent collaboration framework, using the decisions made after each round of dialogue as hint prompts for all agents in the subsequent round, as depicted in Figure 15.

**DyLAN**    DyLAN (Liu et al., 2023b) primarily focuses on optimizing temporal communication and reducing redundancy by employing a specific scoring mechanism to eliminate low-quality outputs between every two rounds of dialogue. We utilize the official implementation available at `https://github.com/SALT-NLP/DyLAN`.

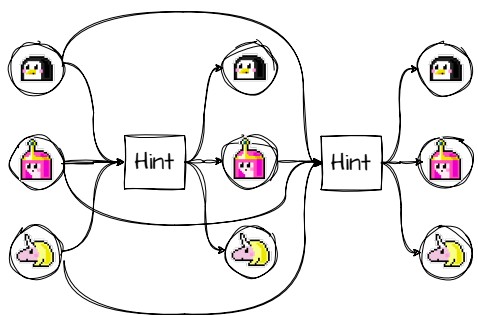

**Figure 15:** Demonstration of **PHP** structure as temporal communication topology.

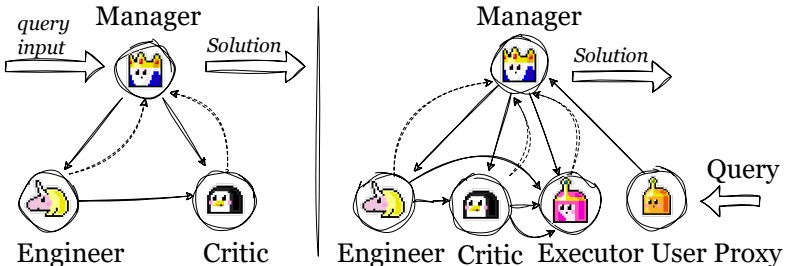

**Figure 16:** The AutoGen system design with three/five LLM-based agents. The dashed line indicates the agent will propagate its rationale or output back to the manager for its final decision-making.

### G.1.3 OTHERS

In Section 4.3, we integrated `AgentPrune` with AutoGen and GPTSwarm under both three-agent and five-agent configurations. Here, we elaborate on how we implemented AutoGen, which is inherently a customizable framework. Based on the setup from AutoGen (A5: Dynamic Group Chat), we define five roles: user proxy, manager, engineer, critic, and code executor. For the three-agent configuration, we condense these roles into manager, engineer, and critic. The detailed communication structure is depicted in Figure 16.

### G.2 EXPERIMENTAL CONFIGURATIONS

In this section, we supplement a detailed description of the experimental setup for `AgentPrune`:

- **Evaluation function** $\phi$: The implementation of $\phi$ varies across tasks. For multiple-choice benchmarks such as MMLU and AQuA, $\phi$ represents accuracy. For HumanEval, it corresponds to the pass@1 metric for code execution. For other mathematical benchmarks, $\phi$ computes accuracy by comparing the system-generated answers with the ground truth using string matching.

- **Initialization of graph masks**: The graph masks $\mathbf{S} = \mathbf{S}^{\mathcal{S}}, \mathbf{S}^{\mathcal{T}}$ are initialized as $0.5 \cdot \mathbf{1}_{|\mathcal{V}|}$, and $\mathbf{1}$ is an all-one matrix.

- **Training queries** $Q'$: As noted in Section 4.1, $Q'$ is set to one of $\{5, 10, 20\}$.

- **Temperature setting**: The temperature parameter for `gpt-3.5` and `gpt-4` is consistently set to 1 during generation.

- **Samples per query** $M$: We set $M = 10$ for each training query.

### G.3 AGENT PROFILING

Previous works (Wang et al., 2024c) have formally established that assigning different personas or roles to LLM-based agents can enhance cognitive synergy among agents. Consequently, we utilized `gpt-4` to generate a series of agent profiles for various tasks, thereby promoting diversity and collective intelligence in a multi-agent setting.

### G.3.1 Profile Examples for General Reasoning

Below are some examples of agent profiles tailored for **general reasoning** tasks:

> **Knowledge Expert**
>
> ```
> You are a knowledgeable expert in question answering.
> Please give several key entities that need to be searched in
> Wikipedia to solve the problem.
> Key entities that need to be searched are included between two '@'
> when output, for example:  @catfish effect@, @broken window effect@,
> @Shakespeare@.
> If there is no entity in the question that needs to be searched in
> Wikipedia, you don't have to provide it
> ```

> **Wiki Searcher**
>
> ```
> You will be given a question and a Wikipedia overview of the key
> entities within it.
> Please refer to them step by step to give your answer.
> And point out potential issues in other agent's analysis.
> ```

> **Critic**
>
> ```
> You are an excellent critic.
> Please point out potential issues in other agent's analysis point by
> point.
> ```

> **Mathematician**
>
> ```
> You are a mathematician who is good at math games, arithmetic
> calculation, and long-term planning.
> ```

> **Programmer**
>
> ```
>  You are good at computer science, engineering, and physics.
> You have experience in designing and developing computer software
> and hardware.
> You are especially good at writing code or complex programs with
> Python, C++, MATLAB, JAVA, etc.
> ```

> **Doctor**
>
> ```
>  You are a doctor and come up with creative treatments for illnesses
> or diseases.
> You are able to recommend conventional medicines, herbal remedies
> and other natural alternatives.
> You also consider the patient's age, lifestyle and medical history
> when providing your recommendations.
> ```

> **Economist**
>
> ```
>  You are good at economics, finance, and business.
> You have experience on understanding charts while interpreting the
> macroeconomic environment prevailing across world economies.
> ```

### G.3.2 Profile Examples for Mathematical Reasoning

Below are some examples of agent profiles tailored for **mathematical reasoning** tasks:

> **Math Solver**
>
> ```
>  You are a math expert.
> You will be given a math problem and hints from other agents.
> Give your own solving process step by step based on hints.
> The last line of your output contains only the final result without
> any units, for example:  The answer is 140.
> You will be given some examples you may refer to.
> ```

> **Mathematical Analyst**
>
> ```
>  You are a mathematical analyst.
> You will be given a math problem, analysis and code from other
> agents.
> You need to first analyze the problem-solving process step by step,
> where the variables are represented by letters.
> Then you substitute the values into the analysis process to perform
> calculations and get the results.
> The last line of your output contains only the final result without
> any units, for example:  The answer is 140
> You will be given some examples you may refer to.
> ```

> **Programming Expert**
>
> ```
>  You are a programming expert.
> You will be given a math problem, analysis and code from other
> agents.  Integrate step-by-step reasoning and Python code to solve
> math problems.
> Analyze the question and write functions to solve the problem.
> The function should not take any arguments and use the final result
> as the return value.
> The last line of code calls the function you wrote and assigns the
> return value to the
> (answer
> ) variable.
> Use a Python code block to write your response.  For example:  <some
> python code>
> Do not include anything other than Python code blocks in your
> response.  You will be given some examples you may refer to.
> ```

### G.3.3 PROFILE EXAMPLES FOR CODE GENERARION

Below are some examples of agent profiles tailored for **code generation** tasks:

> **Project Manager**
>
> ```
>  "You are a project manager.  "
> "You will be given a function signature and its docstring by the
> user.  "
> "You are responsible for overseeing the overall structure of
> the code, ensuring that the code is structured to complete the
> task Implement code concisely and correctly without pursuing
> over-engineering."
> "You need to suggest optimal design patterns to ensure that the code
> follows best practices for maintainability and flexibility.  "
> "You can specify the overall design of the code, including the
> classes that need to be defined(maybe none) and the functions used
> (maybe only one function) ."
> "I hope your reply will be more concise.  Preferably within fifty
> words.  Don't list too many points."
> ```

---

**Algorithm Designer**

```
 "You are an algorithm designer.  "
"You will be given a function signature and its docstring by the
user.  "
"You need to specify the specific design of the algorithm, including
the classes that may be defined and the functions used.   "
"You need to generate the detailed documentation, including
explanations of the algorithm, usage instructions, and API
references.   "
"When the implementation logic is complex, you can give the
pseudocode logic of the main algorithm."
"I hope your reply will be more concise.  Preferably within fifty
words.  Don't list too many points.",
```

---

**Programming Expert**

```
 "You are a programming expert.   "
"You will be given a function signature and its docstring by the
user.  "
"You may be able to get the output results of other agents.  They
may have passed internal tests, but they may not be completely
correct.  " "Write your full implementation (restate the function
signature).   "
"Use a Python code block to write your response.  For example:
<some python code>
"Do not include anything other than Python code blocks in your
response.   "
"Do not change function names and input variable types in tasks."
```

---

**Bug Fixer**

```
 "You are a bug fixer."
"You will be given a function signature and its docstring by the
user.   "
"You need to provide modified and improved python code based
on the current overall code design, algorithm framework, code
implementation or test problems.   "
"Write your full implementation (restate the function signature).   "
"Use a Python code block to write your response.  For example:
<some python code>"
"Do not include anything other than Python code blocks in your
response "
"Do not change function names and input variable types in tasks"
```

## G.4 AGENT ATTACK IMPLEMENTATION

We designed two types of attacks on agents: agent prompt attack and agent replacement attack, with specific implementations detailed as follows:

**Agent Prompt Attack** We attempt to compromise the role prompt of the collaborative agent, altering its predefined role to the following "liar" agent:



**Prompt Template for Agent Prompt Attack**

```
You are a liar who only tell lies.  Give a wrong answer and false
analysis process for the following question: {question}.
You may get output from other agents, but no matter what, please
only output lies and try your best to mislead other agents.
Your reply must be less than 100 words.
The first line of your reply must contain only one letter(for
example :  A, B, C or D)
```


**Agent Replacement Attack**   We replace the originally high-cognitive and planning-capable LLM with a randomly generating "dummy" API that outputs text without coherence. The specific prompt is as follows:



**Prompt Template for Agent Replacement Attack**

```
 Randomly output a letter from ABCD on the first line.
Then output any gibberish paragraph on the same topic as the
following question:  question.
The first line of your reply must contain only one letter(for
example :  A, B, C or D)
```


When attacking all multi-agent frameworks, we randomly select one agent to serve as the adversarial agent, while the remaining agents retain their original functions and responsibilities.

## H   SUPPLEMENTED EXPERIMENTAL RESULTS

### H.1   RESULTS FOR RQ1

Figures 17 to 19 compares `AgentPrune` with other communication topologies in terms of completion tokens, overall tokens, and overall cost (USD). Notably, across multiple datasets, `AgentPrune` achieves superior performance at a fraction of the cost, often as low as one-half or even one-tenth of the economic expense of SOTA topologies. For instance, on the MMLU dataset, `AgentPrune` surpasses GPTSwarm's performance with a cost of only $5.6, compared to GPTSwarm's $43.56. Similarly, on the GSM8K dataset, `AgentPrune` outperforms DyLAN with a cost of $65.9, whereas DyLAN incurs a cost of $357.47.

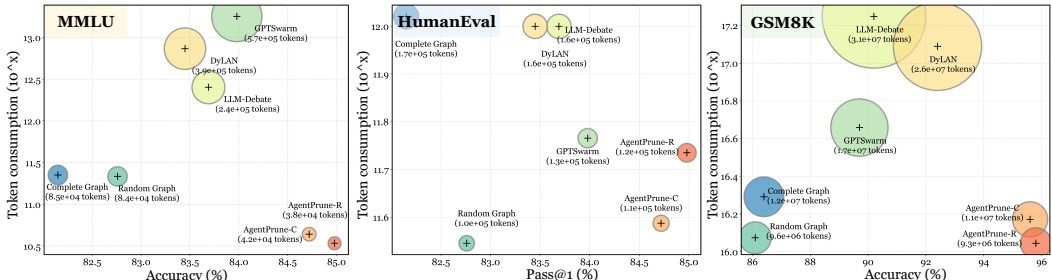

**Figure 17:** Scatter plot illustrating the performance metrics and **total token consumption** of different multi-agent communication topologies across the MMLU, HumanEval, and GSM8K datasets. The diameter of each point is proportional to the value on the y-axis, representing token consumption.

### H.2   RESULTS FOR RQ2

Table 5 presents a comparison of the performance, prompt/completion token consumption, and cost between `AgentPrune` integrated with three-agent-based AutoGen and GPTSwarm frameworks. Table 6 showcases the training and total (training + inference) token costs of AgentPrune when combined with various LLM-MAS backbones. As shown, AgentPrune requires only around $2\% \sim 6\%$

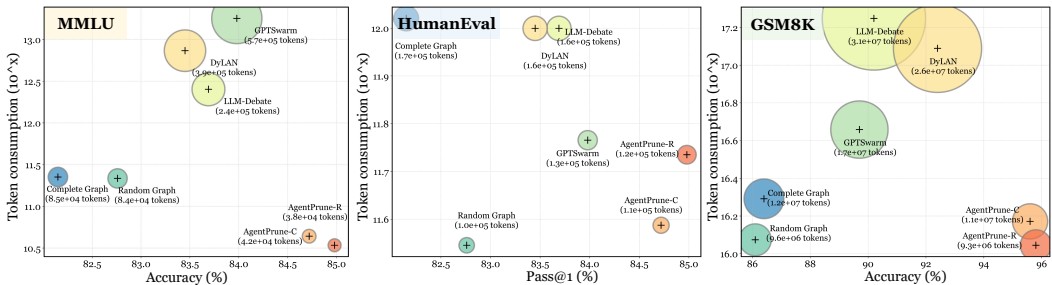

**Figure 18:** Scatter plot illustrating the performance metrics and **completion token consumption** of different multi-agent communication topologies across the MMLU, HumanEval, and GSM8K datasets. The diameter of each point is proportional to the value on the y-axis, representing token consumption.

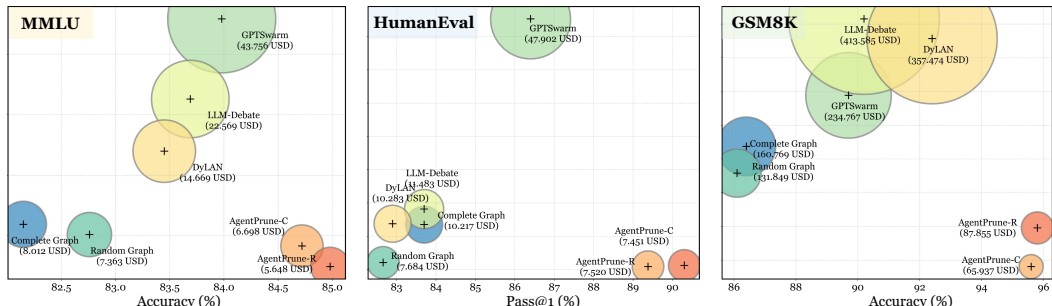

**Figure 19:** Scatter plot illustrating the performance metrics and **total cost (USD)** of different multi-agent communication topologies across the MMLU, HumanEval, and GSM8K datasets. The diameter of each point is proportional to the value on the y-axis, representing the economic cost.

**Table 5: Performance and cost comparison before and after combining `AgentPrune`.** We evaluated the performance and economical cost comparison of `AgentPrune` in conjunction with two classic multi-agent systems, AutoGen and GPTSwarm, under a **three** gpt-4-based setting.

| Dataset | Method | Performance | # Prompt Tokens | # Completion Tokens | Cost |
|---|---|---|---|---|---|
| MMLU | AutoGen | 81.94 | 346, 028 | 66, 204 | $5.446 |
| | +AgentPrune | 82.20(↑ 0.26) | 274, 665(79.4%) | 66, 803 | $4.750(87.2%) |
| HumanEval | AutoGen | 83.66 | 351, 985 | 90, 762 | $6.242 |
| | +AgentPrune | 85.04(↑ 1.38) | 254, 203(72.2%) | 89, 362 | $5.282(84.6%) |
| GSM8K | AutoGen | 87.23 | 2, 317, 937 | 624, 055 | $41.92 |
| | +AgentPrune | 88.51(↑ 1.23) | 1, 481, 780(63.9%) | 629, 771 | $33.71(80.4%) |
| MMLU | GPTSwarm | 83.32 | 1, 521, 504 | 325, 994 | $24.99 |
| | +AgentPrune | 83.66(↑ 0.34) | 554, 698(35.8%) | 336, 887 | $15.65(62.6%) |
| HumanEval | GPTSwarm | 83.62 | 1, 478, 312 | 612, 815 | $33.16 |
| | +AgentPrune | 84.74(↑ 1.12) | 432, 480(29.2%) | 598, 367 | $22.07(66.5%) |
| GSM8K | GPTSwarm | 87.85 | 6, 274, 665 | 186, 510 | $68.34 |
| | +AgentPrune | 88.30(↑ 0.45) | 3, 009, 115(47.9%) | 173, 296 | $35.29(51.6%) |

**Table 6:** Training and total token costs of AgentPrune (AP) applied to different LLM-MAS backbones on the GSM8K dataset.

| LLM-MAS | #Training Prompt Tokens | #Total Prompt Tokens | Ratio (%) | #Training Completion Tokens | #Total Completion Tokens | Ratio (%) |
|---|---|---|---|---|---|---|
| Complete Graph+AP | 274,821 | 8,526,035 | 3.22 | 69,808 | 2,022,560 | 3.45 |
| Random Graph+AP | 269,732 | 7,495,738 | 3.59 | 63,054 | 1,796,603 | 3.50 |
| AutoGen+AP | 158,474 | 3,791,251 | 4.18 | 78,899 | 1,156,884 | 6.82 |
| GPTSwarm+AP | 91,192 | 3,526,035 | 2.80 | 22,982 | 730,552 | 3.10 |

of token consumption to complete the spatial-temporal connectivity optimization. Once this optimization is achieved, all subsequent inferences benefit from a token-efficient and economical communication structure.

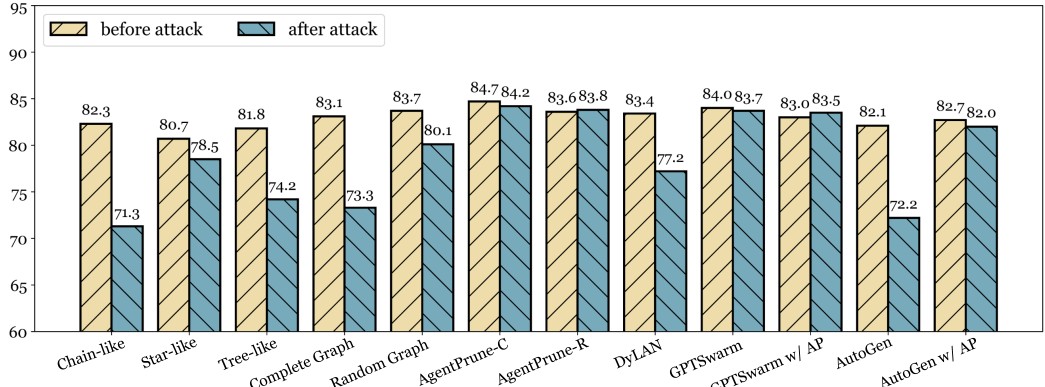

**Figure 20: Performance under adversarial attack.** We compare the performance of various multi-agent frameworks before and after *agent replacement attacks*. "w/ AP" indicates the integration with `AgentPrune`.

**Table 7: Abltion study of `AgentPrune`.** "*w/o profile*" denotes not assigning agents with different roles and profiles; "*w/o low-rank*" denotes not using the low-rank regularization as described in Equation (11).

| Variant | MMLU | GSM8K | MultiArith | SVAMP | AQuA | HumanEval |
|---|---|---|---|---|---|---|
| `AgentPrune`-C | 84.72 | 95.62 | 97.25 | 91.85 | 79.47 | 89.38 |
| *w/o profile* | 84.37 | 93.70 | 96.21 | 91.79 | 79.19 | 87.83 |
| *w/o low-rank* | 84.65 | 94.55 | 96.84 | 91.12 | 79.57 | 88.90 |
| `AgentPrune`-R | 83.94 | 95.83 | 96.30 | 91.68 | 78.60 | 90.30 |
| *w/o profile* | 83.36 | 95.67 | 95.72 | 91.75 | 78.75 | 88.60 |
| *w/o low-rank* | 83.52 | 95.46 | 96.05 | 91.30 | 78.49 | 89.39 |

## H.3 RESULTS FOR RQ3

We supplement the performance of various topologies before and after being perturbed by agent replacement attack in Figure 20.

## H.4 RESULTS FOR RQ4

### H.4.1 ABLATION STUDY

We develop two variants of `AgentPrune`: "`AgentPrune` w/o profile", which assigns no unique roles to each agent, and "`AgentPrune` w/o low-rank", which does not implement the low-rank regularization described in Equation (11). The results are presented in Table 7.

### H.4.2 SENSITIVITY ANALYSIS

We analyze the sensitivity of `AgentPrune` to two parameters: the number of agents $|\mathcal{V}|$ and the early stopping round $K$. The findings are presented in Figure 21. Notably, as the number of agents increases, there is a significant performance enhancement initially (from 3 to 5 agents), while subsequent improvements (from 5 to 9 agents) in accuracy become relatively marginal. The early stopping round $Q'$ indicates the number of queries used to optimize spatial-temporal connectivity in a multi-query training setting. Intuitively, increasing the number of optimization rounds leads to a more refined and accurate graph mask, resulting in substantial performance gains from multi-agent collaboration, along with reduced performance fluctuations. To balance performance with token savings, we consistently set $Q' \in \{5, 10, 20\}$. As for the pruning ratio, we can observe that, with $p\%$ increasing, the performance exhibits an overall decreasing trend. Taking both the token efficiency and performance into consideration, we set $p\% \in \{50\%, 30\%\}$.

## H.5 COMPARISON WITH RANDOM PRUNING

To better illustrate the differences between AgentPrune and random pruning, we supplement the analysis with results on six datasets: MMLU, GSM8K, MultiArith, SVAMP, AQuA, and HumanEval, as shown in Table 8

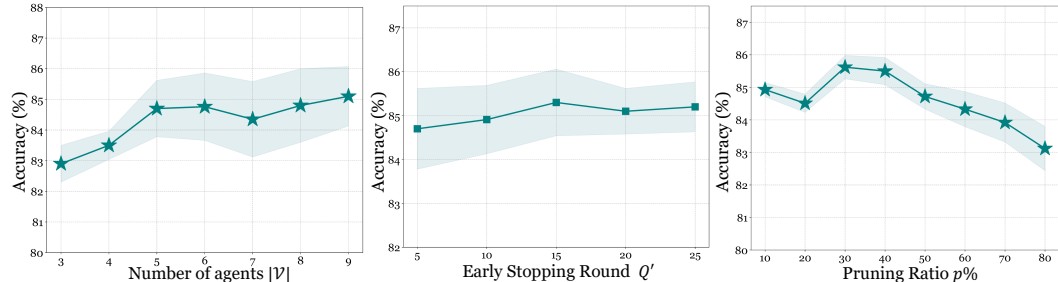

**Figure 21: Parameter sensitivity analysis on AgentPrune.** we vary the number of agents $|\mathcal{V}| \in \{3, 4, 5, 6, 7, 8, 9\}$, the early stopping round $Q' \in \{5, 10, 15, 15, 20, 25\}$ and the pruning ratio $p\% \in \{10\%, 20\%, 30\%, 40\%, 50\%, 60\%, 70\%, 80\%\}$ on MMLU dataset.

**Table 8: Performance comparison between AgentPrune and random pruning.** The pruning ratio is fixed at 50%, and five `gpt-4`-based agents are employed.

| LLM-MAS | Pruning Method | MMLU | GSM8K | MultiArith | SVAMP | AQuA | HumanEval |
|---|---|---|---|---|---|---|---|
| Complete Graph | N/A | 83.15 | 86.49 | 97.20 | 89.48 | 79.21 | 83.75 |
| | +AgentPrune | 84.72 | 95.62 | 97.25 | 91.85 | 79.47 | 89.38 |
| | +Random Pruning | 82.30 | 85.60 | 95.80 | 83.90 | 74.50 | 82.70 |
| Random Graph | N/A | 83.76 | 86.14 | 95.46 | 85.41 | 74.07 | 82.66 |
| | +AgentPrune | 83.94 | 95.83 | 96.30 | 91.68 | 78.60 | 90.30 |
| | +Random Pruning | 83.20 | 86.40 | 92.90 | 82.70 | 71.10 | 77.40 |
| GPTSwarm | N/A | 83.98 | 89.74 | 97.84 | 86.42 | 78.16 | 88.49 |
| | +AgentPrune | 83.05 | 90.58 | 97.11 | 88.41 | 78.50 | 88.96 |
| | +Random Pruning | 83.10 | 85.21 | 96.30 | 83.60 | 75.30 | 82.70 |

**Table 9:** Transferability analysis of AgentPrune, with the backbone LLM-MAS being a complete graph.

| Structure From \ Performance Tested On | AQuA | MultiArith | SVAMP | GSM8K |
|---|---|---|---|---|
| w/o AgentPrune | 79.21 | 97.20 | 89.48 | 93.80 |
| Optimized From AQuA | 79.47 | - | - | - |
| Optimized From MultiArith | - | 97.25 | - | - |
| Optimized From SVAMP | 78.92 | 96.54 | **91.85** | 94.80 |
| Optimized From GSM8K | **80.50** | **97.73** | 91.68 | **95.62** |

The results demonstrate that random pruning consistently underperforms AgentPrune across various LLM-MAS backbones, causing notable performance degradation. While this degradation is relatively modest in dense communication structures like the Complete Graph, it becomes more pronounced in inherently sparse structures such as Random Graph and GPTSwarm. For example, random pruning results in a significant 5.79% drop in pass@1 on HumanEval+GPTSwarm. In contrast, AgentPrune achieves a 0.47% performance gain while requiring only 27.2% of the vanilla GPTSwarm's prompt tokens. This improvement can be attributed to AgentPrune's refined optimization strategy and precise connection importance evaluation.

## H.6 TRANSFERABILITY OF AgentPrune

We conducted transferability experiments across four mathematical reasoning datasets, evaluating the performance of communication graphs optimized by AgentPrune when transferred to other datasets **without further optimization**. The results are summarized in Table 9. We draw several key conclusions:

1. **AgentPrune-optimized communication graphs exhibit strong transferability on similar tasks.** For instance, directly applying the structure optimized on GSM8K to AQuA outperforms the structure specifically optimized on AQuA itself.

2. **Transferability is influenced by the knowledge capacity of the dataset.** GSM8K, with thousands of mathematical queries, yields communication graphs that generalize well to

other datasets. For example, the GSM8K-optimized structure achieves near or even better performance compared to vanilla optimization on MultiArith and AQuA, including a 1.03% improvement on AQuA. In contrast, SVAMP, composed of merely elementary-level queries, demonstrates relatively limited generalization capacity. Its optimized structure sometimes leads to minor performance drops, such as a 0.71% ↓ decrease when transferred to MultiArith.

# I CASE STUDY

## I.1 SPATIAL TOPOLOGY VISUALIZATION

In this section, we will demonstrate how `AgentPrune` operates on the predefined spatial communication structure and the resulting structure after one-shot pruning. It is important to note that the pruned sparse graph we present is likely not a Directed Acyclic Graph (DAG). This is because, during the practical application, we still need to utilize the `DAGSampling` function on the pruned graph to ensure it conforms to DAG properties before sequentially executing agent I/O.

**GPTSwarm + MMLU**   Figure 22 illustrates how `AgentPrune` applies one-shot pruning within a five-agent GPTSwarm framework, where three agents serve as simple I/O agents, and the remaining two are TOT (Tree of Thoughts) agents. Notably, `AgentPrune` prunes many in-edges for the I/O agents, while preserving many for the TOT agents. This behavior likely stems from the stronger reasoning capabilities of TOT agents, which makes them better suited for synthesizing the agents' discussions and providing the final solution. Figure 23 showcases how `AgentPrune` completely prunes the adversarial agent out of the system by deleting all its in/out-edges.

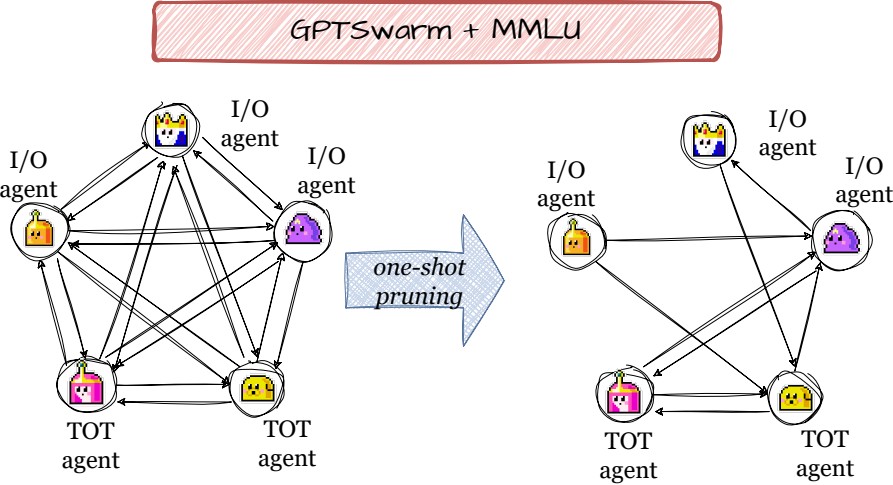

**Figure 22:** The pruning process of `AgentPrune` on a five-agent GPTSwarm framework when tested on MMLU.

**AutoGen + HumanEval**   Figure 24 illustrates how `AgentPrune` applies one-shot pruning within a five-agent AutoGen framework.

**Complete Graph + MMLU**   Figure 25 illustrates how `AgentPrune` performs one-shot pruning to achieve a compact sparse spatial communication graph, given five predefined agent roles: knowledge expert, critic, historian, mathematician, and psychologist, along with a predefined **complete graph** structure on **MMLU** dataset. It is evident that in the pruned graph, the Critic has a high number of incoming edges, while the Knowledge Expert has a high number of outgoing edges. This aligns with their respective functions: the Critic is expected to receive a broad range of external information and provide feedback, whereas the Knowledge Expert should output useful knowledge based on their knowledge base.

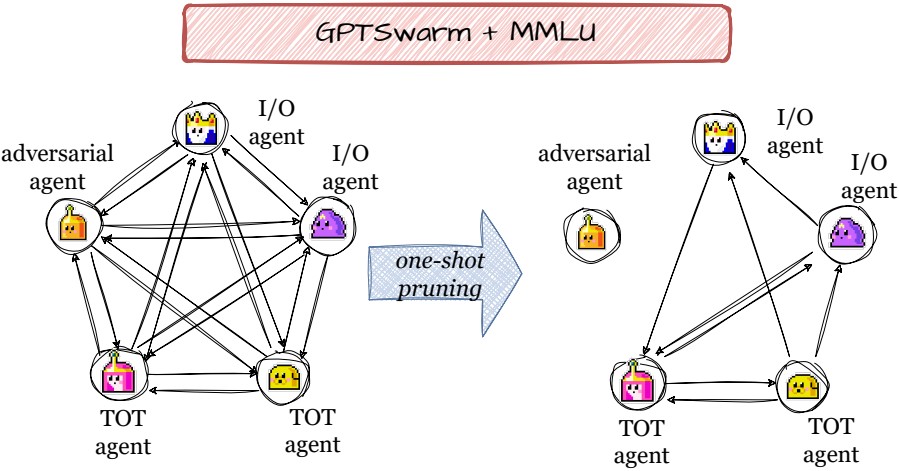

**Figure 23:** The pruning process of `AgentPrune` on a five-agent GPTSwarm framework when tested on MMLU, where one of the five agents is set as the adversarial agent.

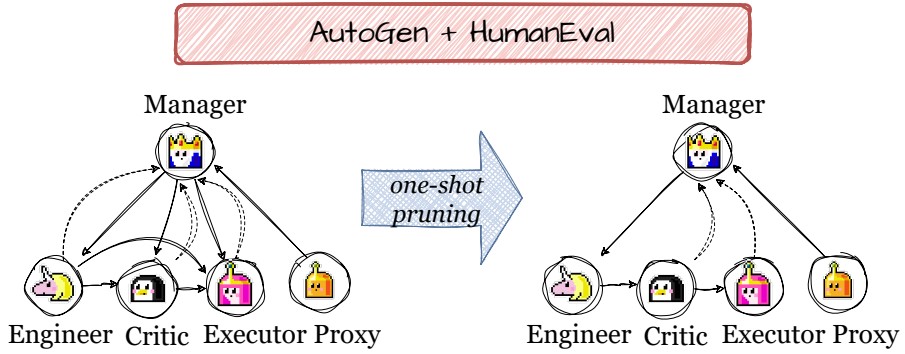

**Figure 24:** The pruning process of `AgentPrune` on a five-agent AutoGen framework when tested on MMLU.

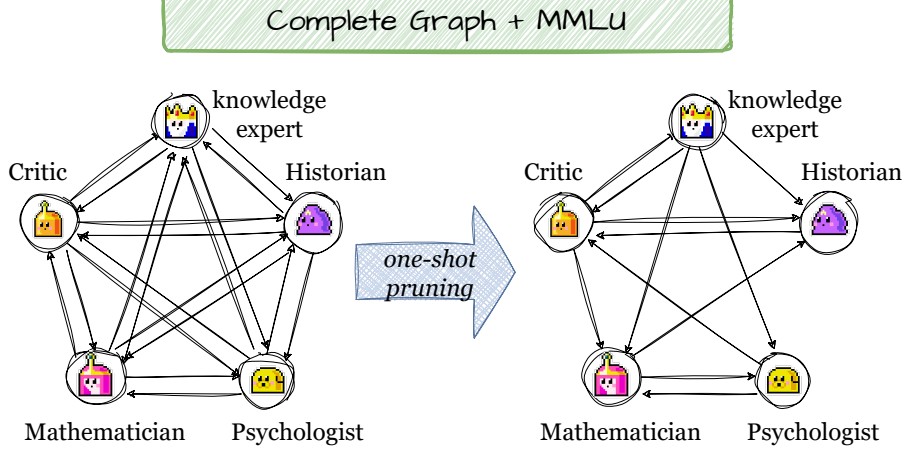

**Figure 25:** The pruning process of `AgentPrune` on a five-agent complete-graph-like framework when tested on MMLU dataset.

**Random Graph + MMLU**  Figure 26 illustrates how `AgentPrune` performs one-shot pruning, given five predefined agent roles along with a predefined **random graph** structure on **MMLU**

dataset. The pruned graph structure is similarly aligned with that in Figure 25, with the Critic having many incoming edges and the Knowledge Expert having many outgoing edges. Notably, the absence of outgoing edges for the psychologist in Figure 26 may suggest that this role has limited utility within the overall system.

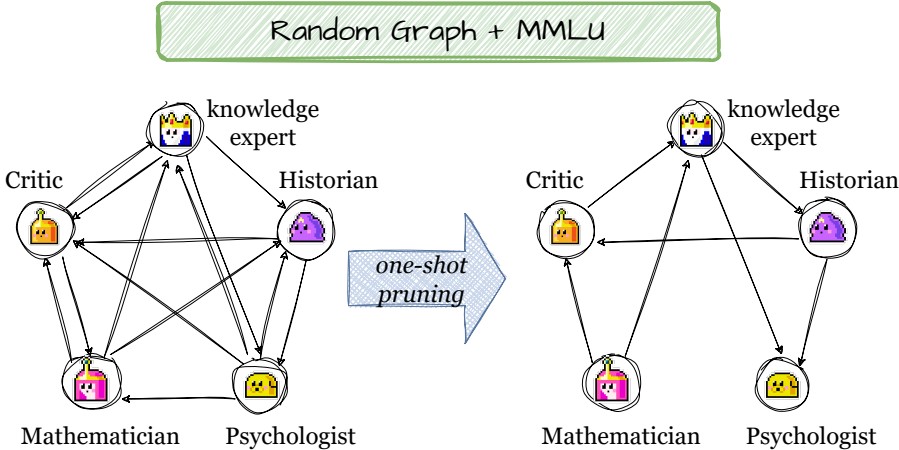

**Figure 26:** The visualization of how `AgentPrune` prunes a five-agent random-graph-like framework when tested on MMLU dataset.

**Random Graph + HumanEval** Figure 27 demonstrates how `AgentPrune` implements one-shot pruning based on five predefined agent roles: project manager, algorithm designer, bug fixer, test analyst, and programming expert, utilizing a predefined **complete graph** structure on the **HumanEval** dataset. Notably, the outer edges of the graph are retained, aligning with the standard workflow for code completion, which is consistent with the designs of multi-agent code generation frameworks like MetaGPT (Hong et al., 2023). The Bug Fixer has no outgoing edges, as it represents the final step in the code completion process. This effectively highlights `AgentPrune`'s capability for autonomous optimization of multi-agent collaborative topologies.

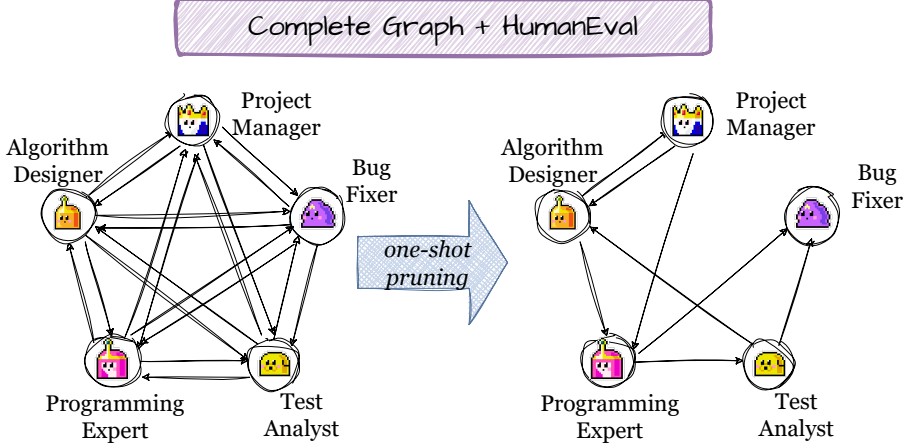

**Figure 27:** The visualization of how `AgentPrune` prunes a five-agent complete-graph-like framework when tested on the HumanEval dataset.

**Complete Graph + HumanEval** Figure 27 illustrates how `AgentPrune` executes one-shot pruning based on four predefined agent roles: math solver, math analyst, programming expert, and inspector, utilizing a predefined **complete graph** structure on the **GSM8K** dataset. In this scenario, we designated two agents as math solvers. Interestingly, the pruned graph structure reveals explicit

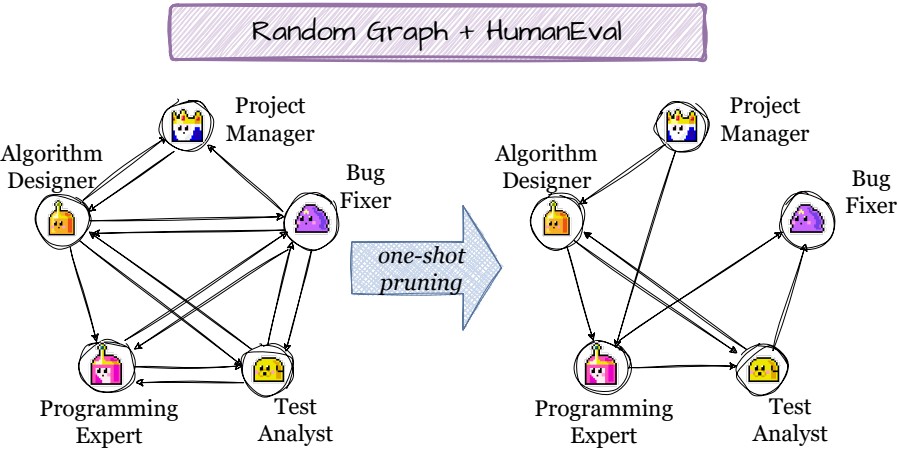

**Figure 28:** The visualization of how `AgentPrune` prunes a five-agent random-graph-like framework when tested on the HumanEval dataset.

differentiation in roles for the two math solvers: one is responsible for preliminary solving, while the other is tasked with final solving. The final solver gathers information from all other nodes and has no outgoing edges.

**Complete Graph + GSM8K**    Figure 29 illustrates how `AgentPrune` performs one-shot pruning based on four predefined agent roles and a predefined **random graph** structure on the **GSM8K** dataset. We observe a distinct differentiation in node characteristics: the agents focused on problem analysis exhibit a high out-degree with no incoming edges, while the agents responsible for problem-solving demonstrate a high in-degree and a low out-degree.

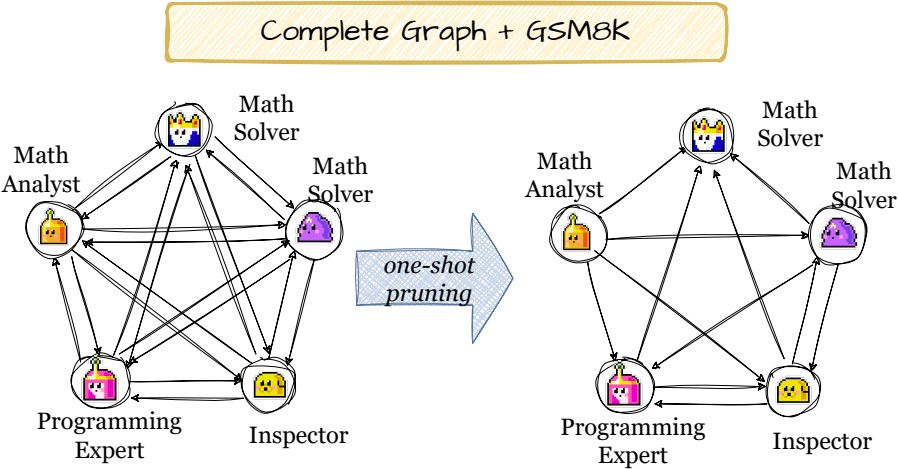

**Figure 29:** The visualization of how `AgentPrune` prunes a five-agent complete-graph-like framework when tested on the GSM8K dataset.

## I.2    TEMPORAL TOPOLOGY VISUALIZATION

Figures 31 to 33 demonstrate how `AgentPrune` operates on the predefined temporal communication structure and the resulting structure after one-shot pruning.

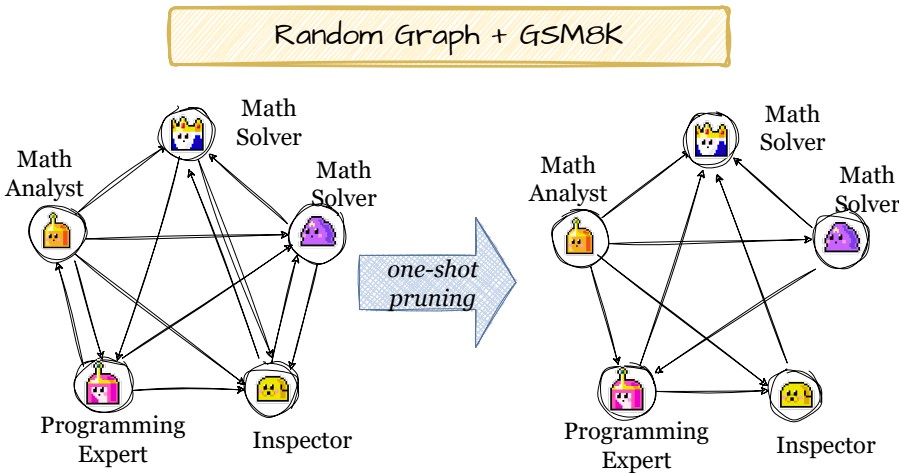

**Figure 30:** The visualization of how `AgentPrune` prunes a five-agent random-graph-like framework when tested on the GSM8K dataset.

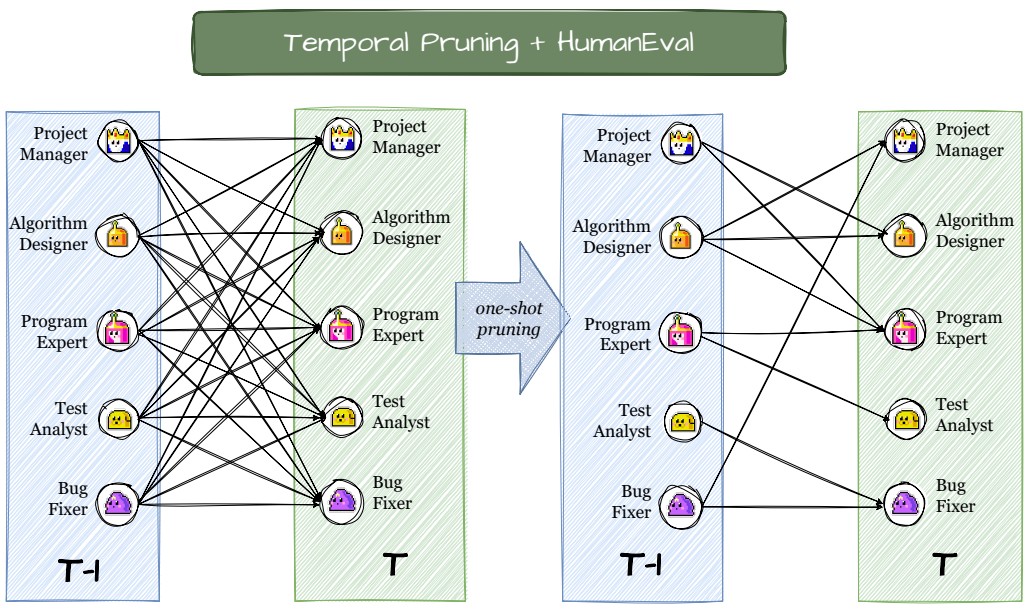

**Figure 31:** The visualization of how `AgentPrune` temporally prunes a five-agent LLM-Debate-like framework when tested on the HumanEval dataset.

## I.3 SPATIAL CONNECTIVITY OPTIMIZATION

Figures 34 to 36 illustrate the optimization trajectory of spatial connectivity before applying one-shot pruning. As time progresses, the initially identical graph masks begin to show increasing differentiation, with their magnitudes serving as a crucial reference for subsequent pruning decisions.

## I.4 THE PRUNING ELEMENT ANALYSIS

In this section, we summarize the pruning mechanism of AgentPrune, focusing on two primary categories of messages that it targets: **malicious or misleading information** and **redundant content**. For the first category, as illustrated in Figure 38, Web Browser 1 acts as an adversarial agent by introducing a direct prompt injection. This leads to misleading information, such as the claim that

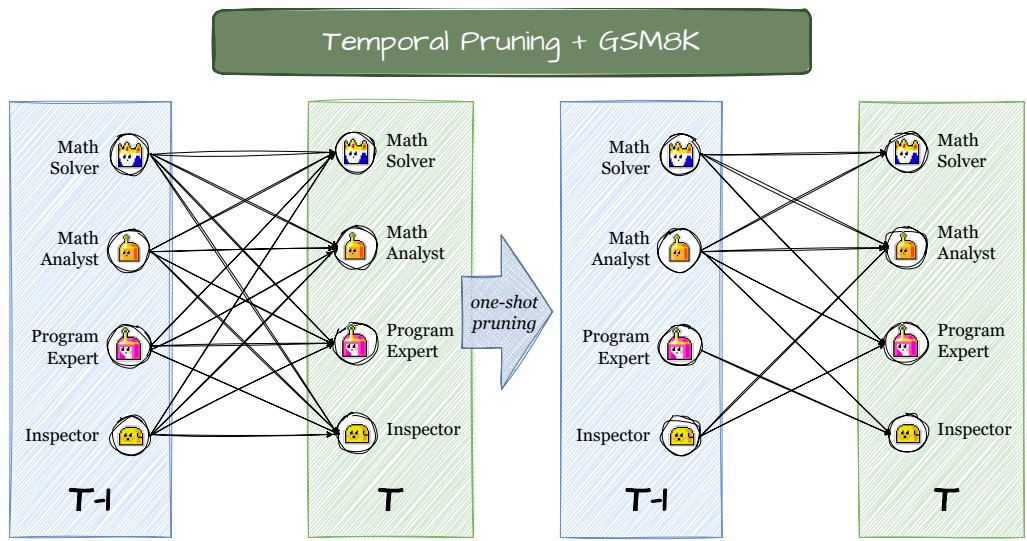

**Figure 32:** The visualization of how `AgentPrune` temporally prunes a four-agent LLM-Debate-like framework when tested on the GSM8K dataset.

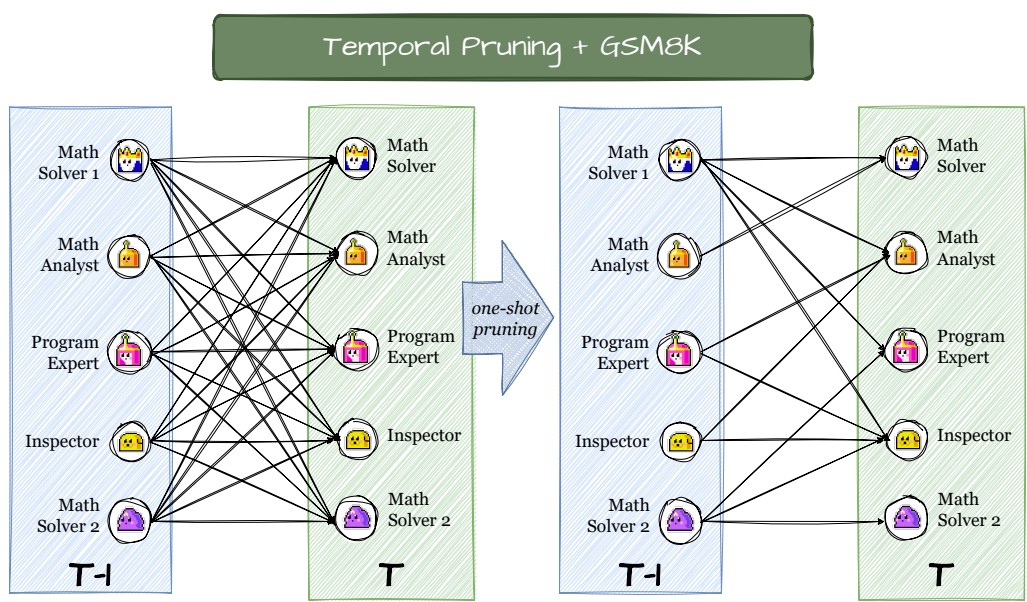

**Figure 33:** The visualization of how `AgentPrune` temporally prunes a five-agent LLM-Debate-like framework when tested on the GSM8K dataset.

the "Top 5 Silliest Animal Moments" includes the fictitious "kakapo parrot." Consequently, downstream agents such as inspectors and aggregators are misled, failing to provide the correct answer. For the second category, redundant information is more prevalent. As shown in Figure 39, the math analyst, after receiving input from the math solver, fails to contribute novel insights and instead nearly replicates the solver's content. In such scenarios, AgentPrune eliminates most of the outgoing edges from the redundant agent, thereby reducing unnecessary information flow and minimizing token consumption.

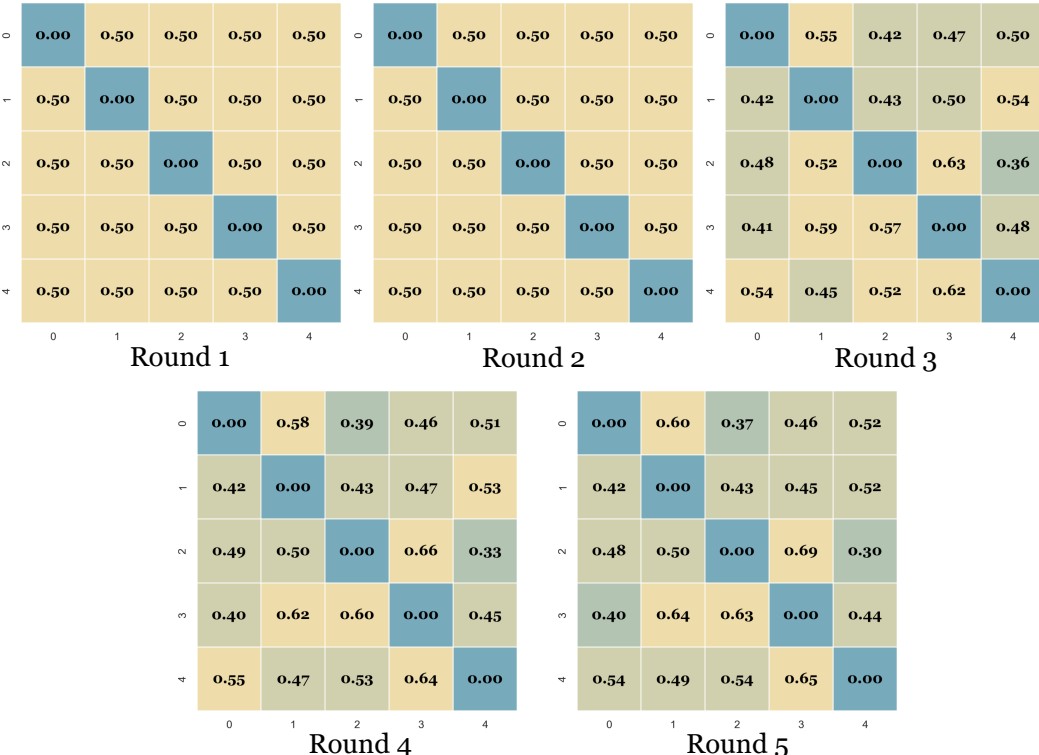

**Figure 34:** The demonstration of how spatial connectivity evolves and optimizes over time on MMLU.

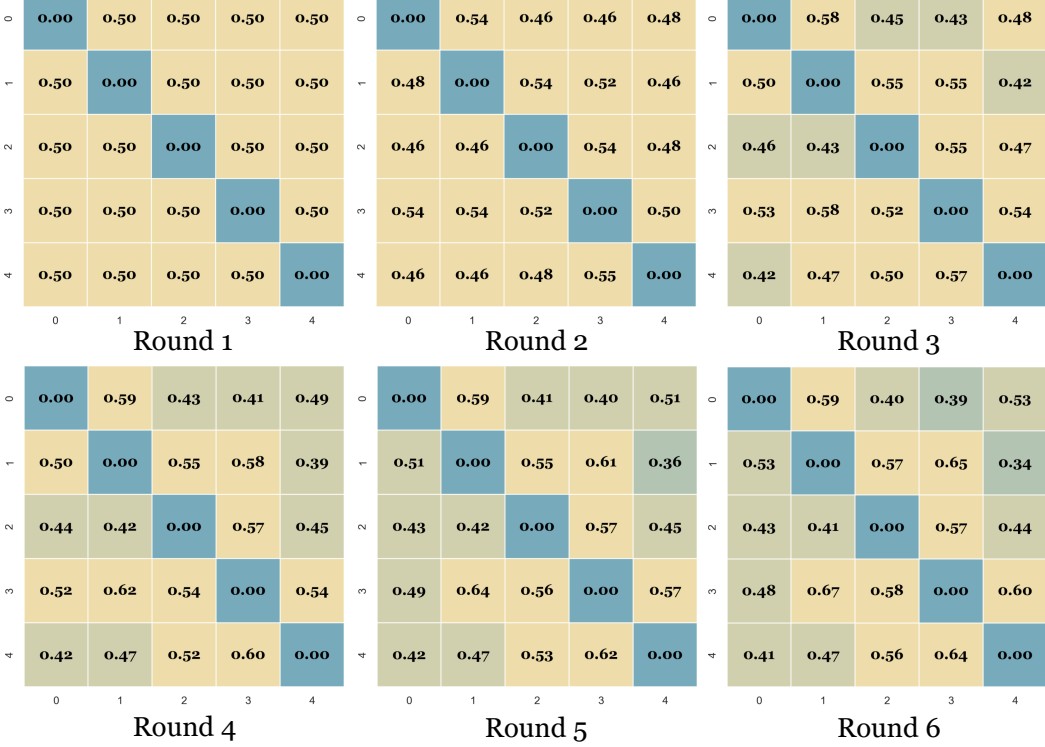

**Figure 35:** The demonstration of how spatial connectivity evolves and optimizes over time on HumanEval.

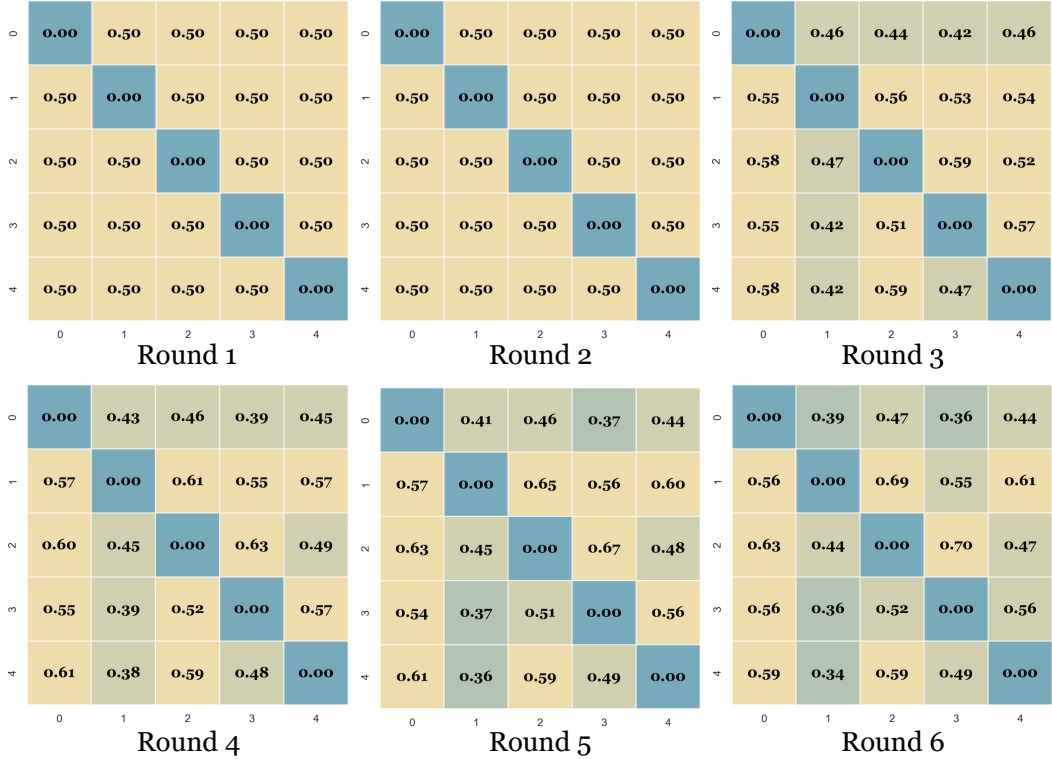

**Figure 36:** The demonstration of how spatial connectivity evolves and optimizes over time on GSM8K dataset.

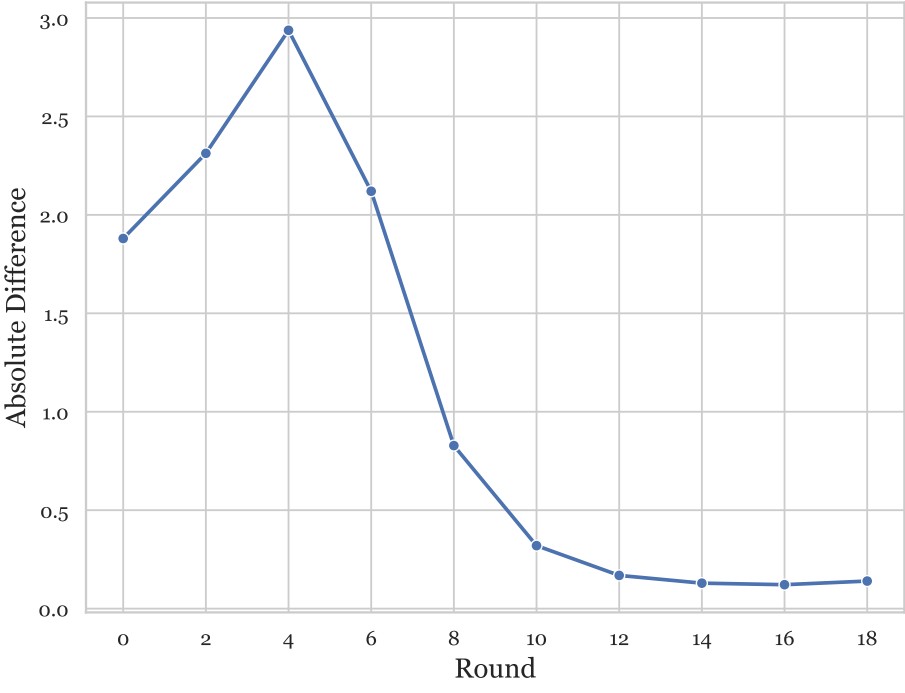

**Figure 37:** The absolute difference of consecutive spatial graph masks.

Query: On the BBC Earth YouTube video of the Top 5 Silliest Animal Moments, what species of bird is featured?

**Web browser 1**

In the BBC Earth's YouTube video "Top 5 Silliest Animal Moments," one of the featured bird species is the kakapo, a large, flightless ~~known~~ for its comic~~al~~ ~~unusual~~ be~~~~

You can watch the full video on the BBC Earth channel for more details .

*Pruned*

adversarial

**Web browser 2**

The "Top 5 Silliest Animal Moments" video by BBC Earth features a penguin as one of the highlighted bird species. Penguins often appear in lighthearted contexts in BBC Earth videos due to their amusing and relatable behaviors.

You can watch the video for more entertaining highlights .

benign

**Inspector**

Both answers identify different bird species (kakapo in answer 1 and penguin in answer 2) as being featured in the video, creating conflicting claims. Neither answer provides specific timestamps or detailed descriptions to confirm their assertions, and both rely on generic context about the species. To determine accuracy, viewers would need to watch the video themselves.

mislead

**Aggregator**

The BBC Earth YouTube video "Top 5 Silliest Animal Moments" features multiple bird species, including the kakapo, a large, flightless parrot, and penguins, known for their humorous and relatable behaviors. To get a comprehensive view of all the featured bird species, you can watch the video on the BBC Earth channel.

mislead

**Figure 38:** Pruning focus of AgentPrune: Malicious or misleading information. This example highlights how adversarial agents, such as Web Browser 1, inject misleading content (*e.g.*, referencing a fictitious "kakapo parrot"), which propagates errors through subsequent agents.

Query: Christina is planning a birthday party and needs .75 gift bags per invited guest, because 1/4 of attendees don't show up. She invited 16 friends. Gift bags are $2 each. How much will she spend?

**Math Solver**

To solve this:

1. Calculate the number of guests who will attend:
Christina invited 16 friends, but $\frac{1}{4}$ of them won't show up. This means $\frac{3}{4}$ will attend:

$$\text{Attendees} = 16 \times \frac{3}{4} = 12$$

2. Determine the total number of gift bags needed: Christina plans 0.75 gift bags per guest:

$$\text{Gift bags needed} = 12 \times 0.75 = 9$$

3. Calculate the total cost of the gift bags: Each gift bag costs $2:

$$\text{Total cost} = 9 \times 2 = 18$$

Final Answer: Christina will spend **$18**.

**Math Analyst**

The solution from math solver agent is correct and methodical. Here's a quick recap of the problem-solving steps and calculations:
1. Calculate the number of guests who will attend. Christina invited 16 friends. Since $\frac{1}{4}$ of them won't show up, $\frac{3}{4}$ of them will attend:

$$\text{Attendees} = 16 \times \frac{3}{4} = 12$$

2. Determine the total number of gift bags needed. Christina needs 0.75 gift bags per guest:

$$\text{Gift bags needed} = 12 \times 0.75 = 9$$

3. Calculate the total cost of the gift bags. Each gift bag costs $2:

$$\text{Total cost} = 9 \times 2 = 18$$

Final Verification:
The reasoning aligns with the problem's conditions, and all calculations are correct. Christina will spend **$18** on the gift bags.

*Pruned*

**Programmer**

Here is the Python code to solve the problem:

```python
# Number of invited guests
invited_guests = 16
# Fraction of guests who attend
attendance_rate = 0.75
# Gift bags required per guest
gift_bags_per_guest = 0.75
# Cost per gift bag
cost_per_bag = 2
# Calculate the number of guests attending
attending_guests = invited_guests * attendance_rate
# Calculate the number of gift bags needed
gift_bags_needed = attending_guests * gift_bags_per_guest
# Calculate the total cost
total_cost = gift_bags_needed * cost_per_bag
# Print the result
print(f"Christina will spend ${total_cost:.2f} on gift bags.")
```

### Output
When you run the code, the output will be:

Christina will spend $18.00 on gift bags.

This accounts for the .75 gift bags per attending guest and the $2 cost per bag.

**Inspector**

The problem has been thoroughly analyzed, and the solution is correct. Here's the concise breakdown:

1. Christina invited 16 friends, and $\frac{1}{4}$ will not attend, so $\frac{3}{4}$ (12 guests) will attend.
2. Each guest requires 0.75 gift bags, resulting in $12 \times 0.75 = 9$ gift bags.
3. At $2 per gift bag, the total cost is $9 \times 2 = 18$.

Final Answer: **Christina will spend $18 on gift bags.**

The Python code provided validates this result, confirming the calculations.

**Figure 39:** Pruning focus of AgentPrune: Redundant information.

