# OpenReview forum: "Cut the Crap: An Economical Communication Pipeline for LLM-based Multi-Agent Systems"
_ICLR.cc/2025/Conference — ICLR 2025 Poster_

### Official Review · Reviewer_rYs7 · 2024-11-02

**Soundness:** 2
**Presentation:** 2
**Contribution:** 2
**Rating:** 3
**Confidence:** 4

**Summary:**

The paper introduces AgentPrune, a framework aimed at optimizing communication in LLM-based multi-agent systems by reducing unnecessary message exchanges, which in turn lowers token usage and economic costs. In MAS, agents collaborate through complex communication topologies, which, while enhancing collective intelligence, can lead to significant token overhead and high operational expenses.

AgentPrune addresses this challenge by introducing a "communication pruning" mechanism, where redundant or non-essential messages within the spatial (intra-dialogue) and temporal (inter-dialogue) message-passing graph are pruned.

Experiments demonstrate AgentPrune's effectiveness across several benchmarks, including MMLU, GSM8K, and HumanEval. Experimental results indicate that AgentPrune achieves comparable performance to existing MAS topologies at a fraction of the cost, with a reported token reduction of up to 72.8% and an increased robustness to basic adversarial attacks by 3.5% to 10.8%

**Strengths:**

1. It addresses an important problem in MAS involving LLMs: the significant economic costs associated with token usage. By focusing on token efficiency, it highlights an often-overlooked aspect of MAS design, especially relevant as these systems scale up. The emphasis on reducing token consumption could encourage future MAS research to adopt similar cost-effective approaches, making it a useful reference for economical MAS deployment.

2. It borrows idea from the spatial-temporal graphs to manage agent communication provides a structured way to visualize and optimize message passing in MAS. This graph-based approach brings an analytical perspective to inter-agent communication, offering insights into which interactions can be minimized without sacrificing performance. Such structured pruning could inspire other MAS research to explore graph-based techniques for optimizing agent communication.

3. It formally define "communication redundancy" in LLM-powered MAS, bringing attention to the often excessive and unnecessary communications within these systems. By identifying this redundancy, the authors shed light on a specific inefficiency that had not been well-formalized in previous MAS research, thus contributing a valuable perspective to MAS optimization.

**Weaknesses:**

1. Inadequate Justification for Pruning Methodology: The one-shot pruning approach, though efficient, lacks theoretical grounding in the context of MAS. The decision to prune a fixed percentage of tokens and communication paths across all tasks does not account for the varying communication needs of different agents or tasks. This oversimplified pruning criterion may lead to suboptimal performance in scenarios where more dynamic and context-sensitive pruning is required.

2. The paper provides little analysis of how pruning affects the quality and depth of agent interactions or task accuracy, especially in more nuanced MAS tasks. **For instance, the pruning’s impact on collaborative tasks requiring high contextual awareness is underexplored.**

3. The number of colors and the format of text in Figure 1 is excessive, and the text is too lengthy, making the annotations unclear and distracting from the overall understanding of the figure.

**Questions:**

Regarding point 2: If you could include experiments that require high contextual awareness, demonstrating that your model can improve both quality and token efficiency in such scenarios, I would be happy to raise my evaluation score.

And I would appreciate a deeper analysis of the types of errors or ineffective communications that AgentPrune eliminates, as this would clarify the practical implications of its pruning method. Such insights would better illustrate AgentPrune’s strengths and limitations and help inform users of any potential side effects.

**Details Of Ethics Concerns:**

The paper's open-sourced repository (https://anonymous.4open.science/r/AgentPrune-F6A5) was last updated on October 19th, which is after the ICLR submission deadline. I am uncertain whether this warrants a desk rejection.

---

> ### Author Response · Authors · 2024-11-19
> **[Part 1/4] Response to Reviewer rYs7**
>
> We sincerely thank you for your careful comments and thorough understanding of our paper! Here we give point-by-point responses to your comments and describe the revisions we made to address them.
>
> ---
> > `Weakness 1.1: Inadequate Justification for Pruning Methodology` The one-shot pruning approach, though efficient, lacks theoretical grounding in the context of MAS.
>
>
> We sincerely appreciate your thoughtful feedback! Please allow us to introduce **how AgentPrune is motivated and grounded**. The idea behind AgentPrune is quite intuitive: data structures such as graphs or networks are often highly redundant. This concept has been explored, defined, and addressed in thousands of studies, covering areas such as neural network pruning [1,2], graph sparsification [3,4], graph-parameter co-pruning [5], neural architecture search (NAS) [6] and traditional multi-agent reinforcement learning (MARL) [7]. These studies have demonstrated that whether dealing with neural networks composed of neurons or graphs made up of nodes and edges, many components can be pruned without significantly impacting the system's utility.
>
> As formalized in Section 2, LLM-MAS can be naturally defined as a spatial-temporal graph. Consequently, examining its redundancy and removing it becomes a natural and meaningful task. One-shot pruning, as a classic pruning method, has shown significant success in network pruning [2], graph pruning [4], NAS [6] and MARL [7]. Thus, we selected one-shot pruning as the initial approach for achieving efficient LLM-MAS. We further hope that AgentPrune can pave the way for the community to explore more advanced techniques for enhancing MAS efficiency, with our well-formalized problem definition.
>
> ---
> > `Weakness 1.2: Fixed structure across all tasks` The decision to prune a fixed percentage of tokens and communication paths across all tasks does not account for the varying communication needs of different agents or tasks. This oversimplified pruning criterion may lead to suboptimal performance in scenarios where more dynamic and context-sensitive pruning is required.
>
> To address your concerns, we have undertaken the following: (1) an analysis of why AgentPrune works, (2) an empirical examination of AgentPrune's transferability, and (3) the introduction of how AgentPrune can achieve task adaptivity.
>
> **(1) Why AgentPrune Works** We respectfully revisit the underlying rationale of AgentPrune, which utilizes a small subset of samples from a dataset to train and subsequently prune a communication structure that generalizes across the entire dataset. This approach is effective because queries within the same dataset typically belong to the same task category, such as mathematical reasoning or code generation, which often share similar reasoning processes and division of labor among agents. At the same time, we respectfully note that leveraging a fixed communication structure for tasks of the same domain is not an uncommon practice, which is also seen in several well-established multi-agent frameworks, including LLM-Debate, MetaGPT, GPTSwarm, CAMEL, AutoGen and Exchange-of-Thought.
>
> **(2) AgentPrune Exhibits Cross-Dataset Generalization**
> While the workflow discussed earlier operates within a single dataset, we wish to highlight that AgentPrune demonstrates remarkable generalization across similar task domains. Specifically, we evaluated the communication structures optimized on GSM8K and SVAMP by directly transferring them to other datasets without any further optimization. The results, shown in Table A, reveal strong transferability:
>
>
> _Table A. The transferability analysis of AgentPrune, with the backbone LLM-MAS being complete graph._
> |structure from\performance tested on|AQuA|MultiArith|SVAMP|GSM8K|
> |-|-|-|-|-|
> |w/o AgentPrune|79.21|97.20|89.48|93.80|
> |optimized from AQuA|79.47|-|-|-|
> |optimized from MultiArith|-|97.25|-|-|
> |optimized from SVAMP|78.92|96.54|**91.85**|94.80|
> |optimized from GSM8K|**80.50**|**97.73**|91.68|**95.62**|
>
>
> Notably, directly applying the structure optimized on GSM8K to AQuA yields better performance than the structure specifically optimized on AQuA itself. This finding underscores AgentPrune's impressive cross-task generalization capabilities.

---

> ### Author Response · Authors · 2024-11-19
> **[Part 2/4] Response to Reviewer rYs7**
>
> **(3) Achieving Task Adaptivity with AgentPrune**  We sincerely appreciate your insightful observation regarding the limitations of applying a fixed percentage of communication paths across diverse tasks, especially in dynamic scenarios. To address this concern, we propose a minimal enhancement to AgentPrune, requiring **fewer than 40 lines of code changes**. This enhancement enables task-adaptive allocation of communication structures through the integration of a lightweight task evaluator $\psi$:
>
> - **During training**, the evaluator takes the task description $q$ as input and outputs a task-adaptive threshold $thres$, calculated as:
>   $$ \psi(q) = \operatorname{Sigmoid}(\operatorname{FFN}(\operatorname{Embedding}(q))),$$
>   where $\operatorname{Embedding}(\cdot)$ can be implemented via any text embedding model like SentenceBert or MiniLM. The subgraphs $\\{\\mathcal{G}^\mathcal{S}\_k, \mathcal{G}^\mathcal{T}\_k\\}_{k=1}^M$ extracted in Eq. (9) and (10) are constraint by $thres$, ensuring:
>   $$ \mathbf{A}(\mathcal{G}^\mathcal{X}\_k) \subseteq \mathbb{1}[\mathbf{A}(\mathcal{G}^\mathcal{X}) < thres] \odot \mathbf{A}(\mathcal{G}^\mathcal{X}), \mathcal{X}\in\\{\mathcal{S},\mathcal{T}\\}$$
>   which allows $\psi$ to be updated jointly with the graph masks via backpropagation.
>
> - **During inference**, given a query/task $q$, the task evaluator $\psi$ computes the task-specific sparsity. Using this threshold, the optimized graph masks $\mathbf{S} = \\{\mathbf{S}^{\mathcal{S}}, \mathbf{S}^\mathcal{T}\\}$ are pruned to obtain a tailored subgraph $\mathcal{G}^{sub}_q$ for each task.
>
> This straightforward yet impactful component facilitates dynamic configuration of the communication graph. We have made its implementation publicly available in https://anonymous.4open.science/r/agentprune_rebuttal-FE38, where the implementation of $\psi$ can be found at https://anonymous.4open.science/r/agentprune_rebuttal-FE38/AgentPrune/llm/llm_embedding.py.  **The overall modification involves fewer than 40 lines of code modifications compared to the original codebase.**
>
> In the response to `Weakness 2.2`, we will experimentally evaluate whether this enables AgentPrune to further satisfy the requirements for dynamic scenarios.
>
>
>
> > `Weakness 2.1: How pruning affects quality and depth` The paper provides little analysis of how pruning affects the quality and depth of agent interactions or task accuracy, especially in more nuanced MAS tasks.
>
>
> Thank you for your insightful comment! Here, we address your concerns regarding AgentPrune's impact on interaction quality and depth separately:
>
> **Quality** To evaluate how AgentPrune impacts the quality of agent interactions, we measured the accuracy of multi-turn dialogues before and after incorporating AgentPrune across three MMLU subsets. The results are summarized in Table B:
>
> _Table B. Accuracy of agent interactions with and without AgentPrune on LLM-Debate. We use four gpt-3.5-based agents._
> |Subset|Method|Round 1|Round 2|Round 3|Round 4|
> |-|-|-|-|-|-|
> |Humanities|LLM-Debate|57.3|59.8|60.4|60.4|
> ||+AgentPrune|56.4|60.4|62.1|62.1|
> |Social Science|LLM-Debate|70.3|74.8|74.8|72.7|
> ||+AgentPrune|72.7|74.8|77.4|77.4|
> |STEM|LLM-Debate|57.0|65.9|69.7|69.7|
> ||+AgentPrune|58.1|69.0|73.3|73.3|
>
> The results reveal two main observations. First, **with AgentPrune, the system achieves peak performance more quickly**. For instance, in the Social Science subset, AgentPrune enables the system to reach an accuracy of 74.8% in Round 2, comparable to the Round 3 accuracy of vanilla LLM-Debate. Second, **AgentPrune improves the ultimate accuracy of the system.** In the STEM subset, for example, it boosts final accuracy by 3.6%, surpassing the peak performance of the vanilla LLM-Debate. These findings demonstrate that AgentPrune enhances the quality of interactions, fostering more effective and redundancy-free agent discussions.
>
> **Depth** In our experiments, the number of interaction rounds was fixed for all LLM-MAS setups to ensure fair comparisons. Therefore, AgentPrune does not explicitly alter the depth of interactions. However, as shown in Table B, it enables the system to converge more quickly to accurate consensus, optimizing the trade-off between depth and quality within the same interaction constraints.

---

> ### Author Response · Authors · 2024-11-19
> **[Part 3/4] Response to Reviewer rYs7**
>
> > `Weakness 2.2: Pruning's impact on collaborative tasks` For instance, the pruning’s impact on collaborative tasks requiring high contextual awareness is underexplored.
>
>
> To address your concerns, we will: (1) clarify that the benchmarks currently used already require high contextual awareness, and (2) provide additional experimental results for AgentPrune on more dynamic tasks.
>
> **Firstly, we humbly emphasize that some of the baselines we selected inherently demand high contextual awareness.** For example, the code generation benchmark *HumanEval* assigns distinct roles to different agents (e.g., code reviewer, programmer, product manager), each receiving unique context and handling specific tasks. This process often requires iterative refinement of code to achieve the desired outcome. Thus, we respectfully assert that the current experimental results already reflect AgentPrune's effectiveness in tasks requiring high contextual awareness.
>
> **Secondly, we supplement our experiments in more dynamic and open-ended environments.** Specifically, we evaluated AgentPrune's performance and token efficiency on two subtasks from the GAIA [8] benchmark: web browsing (355 queries) and diverse filetype reading (129 queries), when combined with GPTSwarm. These subtasks involve either highly dynamic open web environments or complex toolchains and are widely recognized as agentic benchmarks that necessitate high contextual awareness. The results are summarized in Table C.
>
> _Table C. Performance and token efficiency of AgentPrune combined with GPTSwarm. Both web browsing and file reading used Level 1 difficulty tasks. GPT-4 and AutoGPT results are taken from the GAIA benchmark._
> |Subtask|Web browsing||File reading|||
> |-|-|-|-|-|-|
> |**Metric**|**Perf**|**#Token consumption**|**Performance**|**#Token consumption**|
> |single GPT4|18.1|-|5.3|-|
> |AutoGPT|20.5|-|15.4|-|
> |GPTSwarm|28.32|694,354|19.70|234,665|
> |GPTSwarm+AgentPrune|27.18|470,228 (67.7%)|22.18|150,589 (64.1%)|
> |GPTSwarm+AgentPrune (w/ $\psi$)|30.12|402,681 (60.5%)|21.59|92,620 (39.4%)|
>
> Our observations are as follows:
> 1. **AgentPrune remains effective in highly dynamic environments.**
>    The web browsing task includes complex and dynamic scenarios such as website widgets access and Google Street View, while file reading involves intricate tool usage like Excel summarization and PowerPoint viewer. Despite the challenges posed by such dynamic and complex benchmarks, AgentPrune achieves significant token savings, up to **60%**, when integrated with GPTSwarm.
>
> 2. **Task-adaptive AgentPrune provides additional benefits.**
>    As discussed in our response to `Weakness 1.1`, AgentPrune w/ $\psi$ dynamically adjusts the sparsity of the communication structure based on task requirements. This adaptation delivers two key improvements:
>    - **Performance Gains:** AgentPrune w/ $\psi$ achieves a **1.8%** improvement in web browsing and a **1.89%** improvement in file reading.
>    - **Greater Token Efficiency:** In file reading, AgentPrune w/ $\psi$ delivers **24.7%** more token savings compared to the vanilla AgentPrune, thanks to its task-adaptive complexity adjustment.
>
> These results further validate that AgentPrune are well-suited for tasks requiring high contextual awareness and excel in dynamic and complex environments.
>
> **Finally, we present a case study demonstrating the task-adaptiveness of AgentPrune.** As shown in Figure 39 of our updated manuscript, we observe that for elementary-level queries, AgentPrune effectively adjusts to a higher sparsity, resulting in a highly simplified communication structure. In contrast, for more complex queries, a more intricate structure is allocated. We believe these discussions sufficiently validate that AgentPrune excels in addressing dynamic and open-world challenges.
>
> ---
> > `Weakness 3: Formate issue of Figure 1`
>
> Thank you for your valuable suggestions! Based on your feedback, we have carefully revised Figures 1 and 4 to enhance their readability:
>
> - **For Figure 1 (introduction figure),** we
>   - Standardized all fonts to Georgia for consistency.
>   - Reorganized the legend to improve clarity and ease of understanding.
>
> - **For Figure 4 (overall framework),** we
>   - Simplified the color scheme, unifying red for the left side and blue for the right side.
>   - Limited all fonts to Times New Roman and Georgia to avoid excessive formatting.
>   - Removed overly lengthy text, making the framework more concise and intuitive.
>   - Eliminated unnecessary unannotated symbols or elements, placing greater emphasis on AgentPrune’s pruning workflow.
>
> These updates are reflected in the revised manuscript available on OpenReview. We hope these changes address your concerns effectively.

---

> ### Author Response · Authors · 2024-11-19
> **[Part 4/4] Response to Reviewer rYs7**
>
> > `Question 1: Experiments that require high contextual awareness` If you could include experiments that require high contextual awareness, demonstrating that your model can improve both quality and token efficiency in such scenarios, I would be happy to raise my evaluation score.
>
> We sincerely appreciate your insightful feedback and generosity! In our response to `Weakness 2.2`, we supplemented experiments demonstrating AgentPrune's performance in more dynamic and context-intensive tasks, specifically web browsing and file reading. These results confirm that AgentPrune effectively maintains solution quality while significantly reducing token consumption. We hope this addresses your concerns and would be delighted to clarify any further questions!
>
>
>
>
> ---
> > `Question 2: Deeper analysis` And I would appreciate a deeper analysis of the types of errors or ineffective communications that AgentPrune eliminates, as this would clarify the practical implications of its pruning method.
>
> Thank you once again for your thoughtful question! To address your concerns, we have categorized two primary pruning focuses of AgentPrune in practical implementations. Detailed illustrations can be found in the revised manuscript (Figures 37 and 38). Specifically:
>
> - **Malicious or misleading information**: As demonstrated in Figure 37, Web Browser 1 incorrectly claims that the "Top 5 Silliest Animal Moments" includes the fictitious "kakapo parrot." This misinformation propagates to downstream agents, such as inspectors and aggregators, leading to incorrect conclusions. After applying AgentPrune, 3 out of 5 outgoing edges from Web Browser 1 are pruned, allowing the system to generate the correct answer.
>
> - **Redundant information**: In Figure 38, the math analyst provides no novel insights after receiving input from the math solver, merely reiterating the solver’s output. In such cases, AgentPrune significantly reduces the redundant agent's outgoing edges, effectively limiting unnecessary information flow and optimizing token usage.
>
> We hope these analyses clarify the mechanisms and success of AgentPrune in enhancing system efficiency and performance.
>
>
>
> ---
> > `Ethics Concerns`
>
> Regarding the ethics concerns you raised, we respectfully clarify that after thoroughly reviewing all relevant ICLR conference policies, we found no rule prohibiting authors from updating their anonymous code repository after the submission deadline. Unlike ICML, which explicitly addresses this in its guidelines (https://icml.cc/Conferences/2024/AuthorInstructions), **ICLR does not impose such restrictions**. In fact, such updates align with ICLR's encouragement of open-source research and enhancing code usability.
>
> Additionally, we would like to emphasize that the last update to our anonymous repository (on October 19, 2024) involved **fewer than five lines of code changes and minor updates to the README file**. These modifications were solely intended to promote the open-source spirit and ensure that our code remains accessible and easy to use for the research community.
>
> ---
> [1] Only train once: A one-shot neural network training and pruning framework, NeurIPS 2021
>
> [2] Opq: Compressing deep neural networks with one-shot pruning-quantization, AAAI 2021
>
> [3] Demystifying graph sparsification algorithms in graph properties preservation, VLDB 2023
>
> [4] One-shot neural network pruning via spectral graph sparsification, TAG-ML 2023
>
> [5] A unified lottery ticket hypothesis for graph neural networks, ICML 2021
>
> [6] Single path one-shot neural architecture search with uniform sampling, ECCV 2020
>
> [7] Multi-Agent Game Abstraction via Graph Attention Neural Network, AAAI 2020
>
> [8] GAIA: A Benchmark for General AI Assistants

---

> > ### Author Response · Authors · 2024-11-24
> > **Additional Explanations on the Rebuttal**
> >
> > Dear Reviewer rYs7,
> >
> > We would like to express our heartfelt gratitude for your time and effort in reviewing our manuscript. Recognizing the potential differences in our understanding of the work, we deeply value your insights and have provided additional clarifications below to address your concerns and facilitate a better understanding of our work:
> >
> > 1. **Regarding `Weakness 1`: Inadequate Justification for Pruning Methodology**
> >    We respectfully highlight that AgentPrune represents the first formal exploration of pruning and sparsification methods within LLM-based multi-agent systems, building on the enduring successes of these techniques across network pruning, graph sparsification, neural architecture search (NAS), and multi-agent reinforcement learning (MARL). The one-shot pruning employed by AgentPrune is a well-established and robust methodology, validated by extensive literature, and it enables significant token savings due to its simplicity and efficiency.
> > 2. **Regarding `Weakness 2`: Pruning’s Impact on Collaborative Tasks Requiring High Contextual Awareness**
> >    In [**`[Part 3/4] Response to Reviewer rYs7`**](https://openreview.net/forum?id=LkzuPorQ5L&noteId=iwkIhBM1Xp), we have included additional results demonstrating AgentPrune’s **performance** and **token efficiency** on tasks requiring high contextual awareness, such as **web browsing** and **file reading**. These results show that AgentPrune maintains strong performance even in context-intensive scenarios. Additionally, we have made the code implementation available here (https://anonymous.4open.science/r/agentprune_rebuttal-FE38/).
> > 3. **Regarding `Weakness 3`: Formatting Issues in Figures**
> >    Following your suggestions, we have revised Figure 1 (Introduction) and Figure 4 (Overall Framework) to improve their clarity and visual presentation.
> >
> > Your profound expertise and insightful comments inspire our utmost respect. **Addressing your feedback appropriately has become our sincerest commitment.** Therefore, we humbly inquire whether our rebuttal has satisfactorily resolved your concerns and if it might possibly warrant a slight reconsideration of the evaluation. Such a possibility would fill us with immense gratitude and pleasure.
> >
> > Thank you again for your invaluable guidance and thoughtful review.
> >
> > Warm regards,
> >
> > Authors

---

> ### Author Response · Authors · 2024-11-23
> **Thank you & Looking forward to further discussion**
>
> Dear Reviewer rYs7,
>
> We sincerely appreciate your generous commendation and thorough feedback on our work. Your brilliant insights deeply inspired us, and **we are genuinely committed to addressing them with the utmost care.** To aid in better understanding our rebuttal and revisions, we have summarized your key concerns and our responses below:
>
> - **Inadequate Justification for Pruning Methodology** **`Weakness 1`**  We have carefully clarified (1) how AgentPrune is conceptually grounded and motivated, (2) the strong cross-dataset generalization capabilities of AgentPrune's fixed structure, and (3) that **AgentPrune can achieve one structure per task with fewer than 40 lines of code modification**.
> - **How pruning affects quality and depth** **`Weakness 2 & Question 2`**
>   We provided additional analysis of how AgentPrune influences the quality and depth of MAS interactions.
> - **Pruning's impact on collaborative tasks** **`Weakness 2 & Question 1`**  We validated AgentPrune’s adaptability and performance in dynamic collaborative tasks such as web browsing and file reading.
> - **Formatting issues**  **`Weakness 3`**  We have revised Figures 1 and 4 to enhance their clarity and presentation quality.
>
> Thank you once again for your dedication throughout the review process. We fully and humbly understand that you might have a tight schedule, and we would be sincerely grateful if you could let us know **whether our rebuttal has sufficiently addressed your concerns** and **whether it might warrant a reconsideration of the score**. Thank you immensely!
>
> Warm regards,
>
> Authors

---

> ### Author Response · Authors · 2024-11-26
> **Manuscript Revision Deadline Approaching; We Are Sincerely At Your Service**
>
> Dear Reviewer rYs7,
>
> We would like to thank you for your insightful review and dedication to the review process. With your help, the quality of the manuscript has been significantly improved. Due to the approaching deadline for submitting the revised manuscript, we are now submitting a revised version with highlighted changes.
>
> **Also, we would like to respectfully convey our willingness to address any further concerns you may have.** We aim to make the most of this opportunity for revision, as we deeply value your invaluable insights.
>
> Thank you once again for your support.
>
> Warm regards,
>
> Authors

---

> ### Author Response · Authors · 2024-11-27
> **Heartfelt and Humble Invitation to Discussion Before PDF Revision Deadline**
>
> Dear Reviewer rYs7,
>
> We sincerely apologize for reaching out again and fully understand that your time is extremely valuable. If this message causes any inconvenience, we deeply regret and apologize for the interruption.
>
> The reason for our follow-up is that **your review has been really insightful and constructive**, and therefore we are eager to know whether our responses have adequately addressed your concerns.
>
> To further address your expectations regarding **collaborative tasks requiring high contextual awareness**, we extend our experiments beyond the previously included web browsing and file reading tasks by incorporating results on the ALFWorld benchmark. ALFWorld, a diverse suite of synthetic, language-based, interactive decision-making tasks in household environments, serves as a robust evaluation platform for assessing AgentPrune’s effectiveness in applications involving interactive or online decision-making. The results are summarized below:
>
> _Table A. Results on ALFWorld benchmark.
> We employ a four-agent system (instantiated by GPT-3.5), where the roles include task suggestor, executor, grounding agent, and household expert. Results of methods marked with * are borrowed from original papers. For simplicity, in-context examples and parts of the exploration trajectory have been omitted, focusing only on components that contribute to success or failure across three attempts._
> | **Metric**          | **Success Ratio (%)** | **#Prompt Token Consumption** | **#Completion Token Consumption**
> |-|-|-|-|
> | Single GPT-3.5*     | 47%                  | -                     |-|
> | ReAct*             | 54%                  | -                     |-|
> | CAMEL*             | 52%                  | -                     |-|
> | AutoGen            | 70%                  | 334,464                     | 108,736
> | AutoGen + AgentPrune | 73%              | 213,274 (**reducing 36.2%**) | 98,551
>
> We observe that in open-world text-based scenarios—including tasks such as finding hidden objects (e.g., locating a spatula in a drawer), moving objects (e.g., transferring a knife to a cutting board), and manipulating objects with other objects (e.g., chilling a tomato in the fridge)—**AgentPrune is still capable of achieving performance improvement with 36% reduction in prompt token consumption**.  We humbly believe this further corroborates AgentPrune’s utility in highly collaborative, context-sensitive tasks.
>
>
>
> After all these efforts, we would greatly appreciate your guidance on the following points:
> - `For Weakness 1`, whether our **justification of the one-shot pruning methodology** is sufficiently clear.
> - `For Weakness 2`, whether the additional experiments in **collaborative scenarios**, which involve more dynamic and context-sensitive pruning, meet your expectations.
> - `For Question 2`, whether the **deeper analysis** we provided for AgentPrune is sufficient.
>
> We have done our utmost to provide more experimental results and analyses in response to your invaluable feedback. However, **with the PDF revision deadline fast approaching**, we are uncertain if these efforts fully address your insightful comments. We would greatly appreciate your reply and any further suggestions you may have.
>
> Thank you once again for your time and invaluable insights!
>
> Best regards,
>
> All Authors!

---

> ### Author Response · Authors · 2024-12-01
> **Follow-up on Weakness 1: Justification for Pruning Methodology**
>
> Dear Reviewer rYs7,
>
> Warm greetings as we step into December!
>
> With the author-reviewer discussion deadline approaching in two days, we have provided additional explanations regarding your observation on `Weakness 1: Inadequate Justification for Pruning Methodology` to facilitate further communication.
>
> You mentioned that AgentPrune employs an "oversimplified pruning criterion." However, we find this critique could similarly apply to a substantial number of existing pruning methods. AgentPrune leverages magnitude-based pruning with trainable masks, a paradigm widely adopted in the literature. For instance, representative examples include:
>
> - **LTH [1]** (ICLR 2019 Oral; 4000+ citations; magnitude pruning with trainable masks)
> - **GLT [2]** (ICML 2021; 200+ citations; magnitude pruning with trainable masks)
> - **RigL [3]** (ICML 2022; 600+ citations; magnitude pruning with weights)
> - **LLM-Pruner [4]** (NeurIPS 2023; 370+ citations; gradient magnitude pruning)
> - **Wanda [5]** (ICLR 2024; 340+ citations; magnitude pruning with weights and activations)
>
> These examples demonstrate that magnitude-based pruning, whether in network or graph pruning contexts, is not only common but also well-established and grounded in prior work.
>
> Finally, we respectfully reiterate that we have conducted comprehensive analyses and experiments to address all your concerns in detail. With the discussion deadline so close, we are eager to know if our responses have alleviated your concerns and if there is room for you to revisit your evaluation score, as you generously mentioned earlier.
>
> Thank you once again for your valuable feedback and kind consideration.
>
> Best regards,
> Authors
>
> ---
>
> [1] The Lottery Ticket Hypothesis: Finding Sparse, Trainable Neural Networks, ICLR 2019
> [2] A Unified Lottery Ticket Hypothesis for Graph Neural Networks, ICML 2021
> [3] Rigging the Lottery: Making All Tickets Winners, ICML 2022
> [4] LLM-Pruner: On the Structural Pruning of Large Language Models, NeurIPS 2023
> [5] A Simple and Effective Pruning Approach for Large Language Models, ICLR 2024

---

> ### Author Response · Authors · 2024-12-02
> **[ 24 hours remaining ] Invitation to Final Discussion Before Deadline**
>
> Dear Reviewer rYs7,
>
> Thank you for your evaluation of our work and the valuable feedback you have provided. As the deadline for the author-reviewer discussion is fast approaching, we sincerely hope you can join us for the final discussions to further enhance the quality of our manuscript.
>
> We have tried our best to give more experimental results and corresponding analysis according to your requirements. However, we do not know whether this can be a good answer to your comments. We look forward to your reply and corresponding suggestions!
>
> Thank you immensely for your time and insights!
>
> Best regards,
>
> All authors!

---

### Official Review · Reviewer_sHy9 · 2024-11-03

**Soundness:** 3
**Presentation:** 3
**Contribution:** 2
**Rating:** 8
**Confidence:** 4

**Summary:**

The paper studies the communication redundancy problem, i.e. the fact that when multiagent networks solve a task, several exchanged messages are redundant. To address this problem, they model communication as a spatiotemporal graph (i.e. a graph that models both which agents interact with each other and what context should be included) and introduce AgentPrune, an optimization technique that reduces the level of redundancy while preserving performance. This technique involves three steps:
* In the "training" phase ($K^\prime$ rounds), the spatiotemporal graph is optimized according to a metric that rewards assigning higher weights to important connections while at the same time increasing sparsity;
* The connections are then pruned according to a threshold;
* From that point onwards ($K - K^\prime$ rounds), only the non-pruned connections are used.

The authors then study the performance of AgentPrune both in its standalone form and in conjunction with other multiagent frameworks, finding a reduction in overall cost; they also find that AgentPrune leads to improvements in robustness against agent-targeted adversarial attacks.

Overall, the paper studies an important problem and proposes a reasonable (though far from perfect) approach to reduce communication redundancy. In particular, the technique requires an expensive training step; however, for large enough datasets, this leads to long-term efficiency gains. Aside from some issues that raised a few eyebrows, the experimental analysis is well done and the model also offers some robustness gains as a bonus. ~For this reason, my recommendation is 6 (Marginally above acceptance threshold).~

~I would be willing to raise my score should the authors take the following steps (see the Weaknesses section):~
* ~Presenting the results in Table 2 and 3 in a more balanced light;~
* ~Adding the missing experimental info;~
* ~Answering the questions reported in the Questions section (assuming that the results showcase the transferability of AgentPrune).~

Update: the authors' rebuttal has addressed essentially all of my concerns, which is why I'm raising my score to 8 (Accept).

**Strengths:**

* The paper does a good job of formalizing the problem, which I agree is an important one;
* Modelling the task as a spatiotemporal graph is also a pretty solid idea, and the proposed technique can be seen as an intuitive extension of this approach;
* The cost reduction is significant;
* I appreciated the parameter sensitivity analysis and the ablation study, as these types of studies, while expensive, are very important for proper science.

**Weaknesses:**

Note: Most of these weaknesses have been addressed in the rebuttal.

It is unclear to me how applicable AgentPrune is in a real-world context. AP makes sense in situations where the deployer already has a training set for a given task and is willing to invest resources into optimizing the communication graph for future queries. In other words, using AP is reasonable from an economics point of view only at large scales, where there are enough training examples and enough expected future queries to warrant such an optimization step. To be clear, this is not a dealbreaker: several real-world applications fit this description, and my critique of AP can also be applied to any other network optimization technique (e.g. GPTSwarm). Still, it’s hard to make a case for widely applicable cost savings, especially since each graph appears to be task-specific and thus not transferable. Comparing with optimization-free baselines (e.g. CoT) is also a bit unfair, since they do not require this initial training step.

Moreover, emphasizing the reduction in prompt tokens instead of the cost itself (e.g. in Table 3) is slightly misleading, since a) deployers care about the cost, not the prompt tokens b) the cost reductions, while still good, are often less impressive than the prompt token reduction (e.g. -17% cost vs -36% tokens for AG+HumanEval, -55% cost vs -78% tokens for GPTSwarm+MMLU, -52% cost vs -73% tokens for GPTSwarm+HumanEval). In the same vein, reporting in Table 2 the performance gain w.r.t. Vanilla sort of hides the fact that AgentPrune has slightly worse performance compared to Reflexion prompting (which is not a dealbreaker, but it should still be noted).

Finally, there are some missing experimental details in the paper, such as:
* What function is used as $\phi$ in the experiments?
* What connection probability is used to generate the random graph?
* What’s the diversity penalty for the OpenAI models?

While I found the answers to these questions in the code, given the non-static nature of repositories these values should also be reported in the experimental setup appendix of the code.

Minor notes:
* In Table 6 there's a typo related to the word "ablation";
* Again in Table 6, the original scores have 4 significant digits, but the ones after ablation only have 3;
* The title of Appendix F contains a misspelled "Existing";
* Figure 30 has a typo in the word HumanEval;
* In G.3 “liar" is misspelled as “lier”.

**Questions:**

* What’s the minimum scale (in terms of e.g. dataset size and number of queries) where AgentPrune is more cost-effective than other techniques?
* Do networks trained on a task transfer to similar tasks?

---

> ### Author Response · Authors · 2024-11-19
> **[Part 1/3] Response to Reviewer sHy9**
>
> We would like to express our deepest respect for your meticulous review! We can genuinely sense that you have dedicated considerable time to thoroughly reviewing our manuscript (even including the appendices) and providing very specific insights and feedback. We must acknowledge that **this is one of the most enlightening and helpful reviews we have received in recent years**! In response to your efforts, we have carefully prepared a point-by-point reply:
>
> ---
> > `Weakness 1.1: wide application of AgentPrune` In other words, using AP is reasonable from an economics point of view only at large scales, where there are enough training examples and enough expected future queries to warrant such an optimization step. Still, it's hard to make a case for widely applicable cost savings, especially since each graph appears to be task-specific and thus not transferable.
>
> We acknowledge your insight that both AgentPrune and other multi-agent network optimization techniques, such as GPTSwarm and AFlow [1] (a recent multi-agent search framework open-sourced by the MetaGPT team), rely on the availability of a training set, which limits their applicability in broader scenarios. However, we respectfully present the unique characteristic of AgentPrune: **while it does depend to some extent on training resources, this dependency is minimal.**
>
> From an experimental perspective, we provide a comparison of the training costs of AgentPrune and GPTSwarm in **Table A**. As shown, the training token consumption of AgentPrune is significantly lower than that of GPTSwarm. It requires only 3%-4% of the total token usage in the entire experiment to complete the spatial-temporal connectivity optimization.
>
> _Table A. Training and total token costs of AgentPrune-C and GPTSwarm on the GSM8K dataset._
> | Optimization technique| Training Prompt Tokens | Total Prompt Tokens | Ratio | Training Completion Tokens | Total Completion Tokens | Ratio |
> |-|-|-|--|-|-|-|
> | GPTSwarm |  5,570,320 | 14,005,945 | 39.7% | 1,314,780 | 3,156,916 | 41.6%|
> | AgentPrune-C | 274,821| 8,526,035| 3.2% | 69,808 | 2,022,560 | 3.4% |
>
> In terms of the required number of training queries, AgentPrune only needs $\\{5, 10, 20\\}$ queries for optimization, as stated on Line 402, which contributes to its token-saving efficiency. Given this context, we believe that in scenarios with limited data, developers may be unable to use methods like GPTSwarm for initial optimization (since this requires hundreds of queries). However, AgentPrune could be a viable choice.
>
> ---
> > `Weakness 1.2: unfair comparison` Comparing with optimization-free baselines (e.g. CoT) is also a bit unfair, since they do not require this initial training step.
>
> Thank you for your comment! We acknowledge that comparing our approach with optimization-free baselines, such as CoT and ComplexCoT, may not be fair, as they utilize only a single LLM-based agent and do not involve a training process. However, following the methodology of several well-known multi-agent studies, such as EoT[2], AgentVerse[3], GPTSwarm, and DyLAN, which also compare CoT with multi-agent approaches, we chose to follow their setting.
>
> ---
> > `Weakness 2: misleading result` emphasizing the reduction in prompt tokens instead of the cost itself (e.g. in Table 3) is slightly misleading; In the same vein, reporting in Table 2 the performance gain w.r.t. Vanilla sort of hides the fact that AgentPrune has slightly worse performance compared to Reflexion prompting (which is not a dealbreaker, but it should still be noted).
>
> Thank you for pointing that out, and we apologize for any potential confusion caused! Following your suggestion, we have updated **Tables 3 and 5** in the revised manuscript to more comprehensively present savings in both token consumption and cost. Additionally, we have restructured **Table 2** to highlight the ranking order of the different methods. We hope this revision helps to present the results in a more balanced light.

---

> ### Author Response · Authors · 2024-11-19
> **[Part 2/3] Response to Reviewer sHy9**
>
> > `Weakness 3: missing experimental details` Finally, there are some missing experimental details in the paper, such as: What function $\phi$ is used as in the experiments? What connection probability is used to generate the random graph? What’s the diversity penalty for the OpenAI models? While I found the answers to these questions in the code, given the non-static nature of repositories these values should also be reported in the experimental setup appendix of the code.
>
> Thank you for your valuable feedback! Following your suggestion, we have added more detailed experimental settings in `Appendix G.2` and `Appendix G.1.1` of the revised manuscript, including but not limited to:
>
> - The implementation details of function $\phi$
> - The generation process of the random graph communication structure
> - The initialization of the graph masks $\mathbf{S}^\mathcal{S}$ and $\mathbf{S}^\mathcal{T}$
> - The temperature setting for GPT models
>
> We hope these additions enhance the completeness and reproducibility of the paper.
>
> ---
> > `Minor notes: typos`
>
> We greatly appreciate your thorough review! Based on your feedback, we have carefully re-examined the manuscript and made corrections to the typos you pointed out, including spelling errors and issues with numerical precision. All these revisions have been presented in the update manuscript, available on OpenReview.
>
> ---
> > `Question 1: applicable dataset scale` What’s the minimum scale (in terms of e.g. dataset size and number of queries) where AgentPrune is more cost-effective than other techniques?
>
> Your question is highly insightful! To address it, we provide a detailed cost analysis based on **Section 3.4** and **Appendix E** of the manuscript, demonstrating that AgentPrune achieves significant cost savings even on small dataset scales.
>
>
> **Notations**  Consider a dataset comprising $Q$ queries, where $Q'$ queries are used for training. Let the communication graph be $\mathcal{G}$, with $K$ dialogue rounds. Assume the average token consumption per spatial, temporal, and query message is $c_\mathcal{S}$, $c_\mathcal{T}$, and $c_q$, respectively.
>
> **Cost Analysis** For a vanilla multi-agent system, the total token consumption can be approximated as:
> $$
> C_\mathcal{G} = QK\left[ c_\mathcal{S} |\mathcal{E}^\mathcal{S}| + c_\mathcal{T} |\mathcal{E}^\mathcal{T}| + c_q|\mathcal{V}| \right].
> $$
>
> With AgentPrune, the token cost, as given in Eq. (16), becomes:
> $$
> C\_{\mathcal{G}^{sub}} = (Q - Q')K[ (1-p\\%)\cdot\left(c\_\mathcal{S} |\mathcal{E}^\mathcal{S}| + c\_\mathcal{T} |\mathcal{E}^\mathcal{T}|\right) + c\_q|\mathcal{V}| ].
> $$
> The cost saving introduced by AgentPrune can then be quantified as:
> $$
> \Delta = (p\\%\cdot Q + (1-p\\%-M)Q')K(c_\mathcal{S} |\mathcal{E}^\mathcal{S}| + c\_\mathcal{T} |\mathcal{E}^\mathcal{T}|) + (1-M)Q'Kc\_q|\mathcal{V}|.
> $$
>
> When $\Delta > 0$, AgentPrune reduces overall system costs. From $\Delta > 0$, we derive the following constraint:
> $$
> \frac{Q}{Q'} > \frac{(M-1)c_q|\mathcal{V}|}{p\\%(c_\mathcal{S} |\mathcal{E}^\mathcal{S}| + c_\mathcal{T} |\mathcal{E}^\mathcal{T}|)} - (1-p\\%-M).
> $$
>
> This inequality defines the range of dataset sizes $Q$ and training set sizes $Q'$ where AgentPrune is effective.
>
> **Illustrative Example**  While the above constraint is rigorous, it may lack intuitive clarity. To provide a concrete example, we assign realistic parameter values:
> - Vanilla multi-agent system: random graph for spatial communication; LLM-Debate for temporal communication; five LLM-based agents.
> - Parameters: $C_q = 30, C_\mathcal{S}=100, C_\mathcal{T}=100, |\mathcal{E}^{\mathcal{S}}|=10， |\mathcal{E}^{\mathcal{T}}|=25$. The other parameters are the same with the original manuscript, like  $M=10, p\\%=50\\%, Q'=10$.
>
> Substituting these into the equation yields $Q > 102$. This indicates that AgentPrune achieves cost savings on datasets with more than 102 queries. Notably, this threshold is far below the size of datasets we use, such as HumanEval (164 queries) and GSM8K (8.5k queries), as well as popular alternatives like MBPP (1000 queries) [4] and GAIA (450 queries) [5], all of which exceed this requirement.
>
> **Conclusion** This analysis underscores AgentPrune’s broad applicability. It demonstrates that even datasets as small as ~100 queries can benefit from significant cost reductions (and potentially improved performance) through AgentPrune.

---

> ### Author Response · Authors · 2024-11-19
> **[Part 3/3] Response to Reviewer sHy9**
>
> > `Question 2: Transferability of AgentPrune` Do networks trained on a task transfer to similar tasks?
>
> Your question is crucial for enhancing the practical applicability of AgentPrune! To address it, we conducted transferability experiments across four mathematical reasoning datasets, evaluating the performance of communication graphs optimized by AgentPrune when transferred to other datasets **without further optimization**. The results are summarized as follows:
>
> _Table A. The transferability analysis of AgentPrune, with the backbone LLM-MAS being complete graph._
> |structure from\performance tested on|AQuA|MultiArith|SVAMP|GSM8K|
> |-|-|-|-|-|
> |w/o AgentPrune|79.21|97.20|89.48|93.80|
> |optimized from AQuA|79.47|-|-|-|
> |optimized from MultiArith|-|97.25|-|-|
> |optimized from SVAMP|78.92|96.54|**91.85**|94.80|
> |optimized from GSM8K|**80.50**|**97.73**|91.68|**95.62**|
>
> We draw several key conclusions:
> **(1) AgentPrune-optimized communication graphs exhibit strong transferability on similar tasks.** For instance, directly applying the structure optimized on GSM8K to AQuA outperforms the structure specifically optimized on AQuA itself.
> **(2) Transferability is influenced by the knowledge capacity of the dataset.**
> GSM8K, with thousands of mathematical queries, yields communication graphs that generalize well to other datasets. For example, the GSM8K-optimized structure achieves near or even better performance compared to vanilla optimization on MultiArith and AQuA, including a $1.03\%$ improvement on AQuA.
> In contrast, SVAMP, composed of merely elementary-level queries, demonstrates relatively limited generalization capacity. Its optimized structure sometimes leads to minor performance drops, such as a $0.71\%\downarrow$ decrease when transferred to MultiArith.
>
> Overall, we respectfully submit that **AgentPrune demonstrates strong generalization within similar task domains**. Hopefully this can address your concerns, and we are more glad to respond to any further questions you may have!
>
> ---
> [1] AFlow: Automating Agentic Workflow Generation
>
> [2] Exchange-of-Thought: Enhancing Large Language Model Capabilities through Cross-Model Communication, EMNLP2023
>
> [3] AgentVerse: Facilitating Multi-Agent Collaboration and Exploring Emergent Behaviors
>
> [4] Program Synthesis with Large Language Models
>
> [5] GAIA: A Benchmark for General AI Assistants

---

> ### Author Response · Authors · 2024-11-23
> **Thank you & Looking forward to further discussion**
>
> Dear Reviewer sHy9,
>
> We extend our highest respect for the time and effort you devoted to reviewing our manuscript—we can truly sense the considerable attention you have given to every detail. To facilitate better communication and express our gratitude, we sincerely summarize our rebuttal as follows:
>
> 1. **In what scenarios is AgentPrune useful?** **`Weakness 1`** We deeply value your insightful observations! At the same time, we respectfully demonstrate that while both AgentPrune and existing multi-agent optimization methods (e.g., GPTSwarm) require certain optimization resources, AgentPrune's dependence on training resources is significantly lower than that of GPTSwarm, making it applicable to a wider range of scenarios.
> 2. **Does AgentPrune exhibit transferability?** **`Question 2`** We validated the strong transferability of AgentPrune across four datasets: AQuA, MultiArith, SVAMP, and GSM8K.
> 3. **What is the minimum dataset size for AgentPrune to remain cost-effective?** **`Question 1`**  We provided the lower bound for the applicable dataset size of AgentPrune, along with an illustrative example to clarify its effectiveness.
> 4. **Misleading results** **`Weakness 2`** Following your recommendation, we reorganized Tables 2, 3, and 5 to ensure a more balanced presentation of the results.
> 5. **Missing experimental details** **`Weakness 3`** We have included additional experimental parameter details in Appendix G.2 and Appendix G.1.1.
>
> For any other issues not mentioned here, please refer to our detailed rebuttal response. We sincerely and humbly believe we have responded to your concerns properly, and we once again thank you for your meticulous and generous review!
>
> Warm regards,
>
> Authors

---

> > ### Comment · Reviewer_sHy9 · 2024-11-23
> >
> > Apologies for my late response.
> >
> > I have looked at the improvements you have made to the paper and I'd say you have pretty much answered all of my questions. Your transferability results in particular are pretty promising, and I acknowledge the fact that gathering these results during the relatively short rebuttal period must have been quite difficult, which is why I'm even more glad you managed to answer my questions.
> >
> > I also looked at other reviewers' observations and your rebuttals and I did not find any particularly strong flaws.
> >
> > I'm raising my score to 8 (Accept) and my confidence to 4.
> >
> > Thank you again for taking the time to address my concerns, and I wish you the best.
> >
> > Reviewer sHy9

---

> > > ### Author Response · Authors · 2024-11-23
> > > **Thank you immensely!**
> > >
> > > Dear Reviewer sHy9,
> > >
> > > Thank you for your kind comment and stronger support of our work. We deeply appreciate your reviews and feedbacks on the applicability and transferability of AgentPrune, which have significantly elevated the quality of our work. It is our honor to address your concerns, and we sincerely thank you once again.
> > >
> > > Warm regards,
> > >
> > > Authors

---

### Official Review · Reviewer_CryL · 2024-11-04

**Soundness:** 3
**Presentation:** 3
**Contribution:** 3
**Rating:** 5
**Confidence:** 3

**Summary:**

This work focuses reducing communication between multi-agent LLM systems. The goal is to reduce spatial and temporal communication required between multiple LLM agents while keeping high accuracy. The authors propose learning a graph mask using distribution approximation and low-rank sparsity. The proposed method is evaluated on a wide variety of benchmark in terms of communication and cost required and the accuracy achieved.

**Strengths:**

- This work looks into an important problem of reducing cost when using multi-agent LLM system.
- The paper is well written and relatively easy to follow.

**Weaknesses:**

- The paper start with showing random pruning can provide performance improvement, but the proposed method is not compared to the random pruning, so it is hard to understand how much improvement compared to random pruning we have.
- It is not unclear how much training and query/tokens is need to train the system, and how different number of training (one-shot) is needed and how they affect the performance.
- Missing ablation study about the benefit of each of the propose methods (i.e. distribution approximation, low-rank sparsity)

**Questions:**

- What's the training time for this proposed method. Does it overfit to the training data? How does the K' affects the performance.
- Does the mask changes depending on the input data?
- How does the proposed method perform compared to random pruning?
- Can distribution approximation and low-rank sparsity work in isolation? what's their effect on the accuracy and communication reducing?

---

> ### Author Response · Authors · 2024-11-19
> **[Part 1/2] Response to Reviewer CryL**
>
> We sincerely thank you for the thoughtful and constructive reviews of our manuscript! Based on your questions and recommendations, we give point-by-point responses to your comments and describe the revisions we made to address them.
>
> ---
> > `Weakness 1: Comparison with random pruning` The paper start with showing random pruning can provide performance improvement, but the proposed method is not compared to the random pruning, so it is hard to understand how much improvement compared to random pruning we have.
>
> Thank you for your insightful comment! To better illustrate the differences between AgentPrune and random pruning, we supplement the analysis with results on six datasets: MMLU, GSM8K, MultiArith, SVAMP, AQuA, and HumanEval, as shown below:
>
> _Table A. Performance comparison between AgentPrune (AP) and random pruning (RP). The pruning ratio is fixed at 50%._
> | LLM-MAS        | Pruning Method   | MMLU  | GSM8K  | MultiArith | SVAMP  | AQuA   | HumanEval |
> |-----------------|------------------|-------|--------|------------|--------|--------|-----------|
> | Complete Graph | N/A              | 83.15 | 86.49  | 97.20      | 89.48  | 79.21  | 83.75     |
> |                 | +AgentPrune      | 84.72 | 95.62  | 97.25      | 91.85  | 79.47  | 89.38     |
> |                 | +Random Pruning  | 82.30 | 85.60  | 95.80      | 83.90  | 74.50  | 82.70     |
> | Random Graph   | N/A              | 83.76 | 86.14  | 95.46      | 85.41  | 74.07  | 82.66     |
> |                 | +AgentPrune      | 83.94 | 95.83  | 96.30      | 91.68  | 78.60  | 90.30     |
> |                 | +Random Pruning  | 83.20 | 86.40  | 92.90      | 82.70  | 71.10  | 77.40     |
> | GPTSwarm       | N/A              | 83.98 | 89.74  | 97.84      | 86.42  | 78.16  | 88.49     |
> |                 | +AgentPrune      | 83.05 | 90.58  | 97.11      | 88.41  | 78.50  | 88.96     |
> |                 | +Random Pruning  | 83.10 |  85.21   | 96.30      | 83.60  | 75.30  | 82.70     |
>
> The results demonstrate that random pruning consistently underperforms AgentPrune across various LLM-MAS backbones, causing notable performance degradation. While this degradation is relatively modest in dense communication structures like the Complete Graph, it becomes more pronounced in inherently sparse structures such as Random Graph and GPTSwarm. For example, random pruning results in a significant 5.79% drop in pass@1 on HumanEval+GPTSwarm. In contrast, AgentPrune achieves a 0.47% performance gain while **requiring only 27.2% of the vanilla GPTSwarm's prompt tokens (as shown in Table 3)**. This improvement can be attributed to AgentPrune’s refined optimization strategy and precise connection importance evaluation.
>
> ---
> > `Weakness 2: Training cost` It is not unclear how much training and query/tokens is need to train the system, and how different number of training (one-shot) is needed and how they affect the performance.
>
> Thank you for highlighting this issue! To address your concerns, we provide additional analyses on: (1) the training token cost of AgentPrune and (2) the impact of the number of training queries on its performance.
>
> As shown in Table B, we report the training token cost and the total token cost of AgentPrune across different datasets. The results reveal that AgentPrune requires only a minimal proportion of training tokens relative to the total inference token cost, ranging from 2.80% to 6.82%. This demonstrates the high cost-efficiency of AgentPrune.
>
> _Table B. Training and total token costs of AgentPrune applied to different LLM-MAS backbones on the GSM8K dataset._
> | LLM-MAS Backbone | Training Prompt Tokens | Total Prompt Tokens | Ratio | Training Completion Tokens | Total Completion Tokens | Ratio |
> |-|-|-|--|-|-|-|
> | Complete Graph + AP | 274,821| 8,526,035| 3.22% | 69,808 | 2,022,560 | 3.45% |
> | Random Graph + AP | 269,732 | 7,495,738 | 3.59% | 63,054 | 1,796,603| 3.50% |
> | AutoGen + AP | 158,474 | 3,791,251 | 4.18% | 78,899 | 1,156,884 | 6.82% |
> | GPTSwarm + AP| 91,192 | 3,526,035| 2.80% | 22,982 | 730,552 | 3.10% |
>
> In Table C, we conduct a sensitivity analysis of the number of training queries used by AgentPrune on MMLU and HumanEval datasets. The results indicate that the performance of AgentPrune saturates with approximately 20 training queries, which highlights that AgentPrune can effectively learn a generalizable communication structure early in the training process using only a small number of samples.
>
> _Table C. Sensitivity analysis of the number of training queries for AgentPrune-C on MMLU and HumanEval datasets._
>
> | Number of Training Queries | 5|     10   | 20   | 30   | 40  |
> |-|-|-|-|-|-|
> | MMLU                      | 84.30| 84.72 | 84.72|  83.94|  84.94|
> | HumanEval                 | 82.66 |  86.27| 89.38| 90.30| 90.30|

---

> ### Author Response · Authors · 2024-11-19
> **[Part 2/2] Response to Reviewer CryL**
>
> > `Weakness 3: Missing ablation study`: Missing ablation study about the benefit of each of the proposed methods (i.e. distribution approximation, low-rank sparsity)
>
>
> Thank you for highlighting this critical point! However, we respectfully point out that **we have conducted ablation experiments on low-rank sparsity and agent profiling in Section 4.5 and Appendix H.4.1**, with the results detailed in Table 6. It is important to note that we did not present ablation results for distribution approximation, as removing it would render AgentPrune non-functional, i.e., the same as random pruning.
>
> Please allow us to elaborate further on this: during training, the graph masks are updated according to Eq. (8), where the first term corresponds to distribution approximation and the second term corresponds to low-rank sparsity. If the first term is removed, the second term reduces to a rank minimization problem with a closed-form solution, which can be computed without gradient backpropagation. (This is analogous to training a neural network using only L0 regularization on the parameters without providing downstream supervision signals.) In this case, the trainable masks would not receive any meaningful supervision signal, and the training process would become entirely random. Therefore, we did not present results regarding distribution approximation. We hope this clarifies your concerns.
>
>
> ---
> > `Question 1`: What's the training time for this proposed method. Does it overfit to the training data? How does the K' affects the performance.
>
>
> Thank you again for your thoughtful comments! Below, we address each of your questions individually:
>
> **Regarding training time**, we have supplemented the manuscript with training time details, as shown in Table D:
>
> _Table D. The training and overall time consumption of AgentPrune._
> |Dataset|MMLU|||GSM8K|||HumanEval|||
> |-|-|-|-|-|-|-|-|-|-|
> |Time (min)|Training|Overall|Training/Overall|Training|Overall|Training/Overall|Training|Overall|Training/Overall|
> |Complete Graph|-|125|-|-|317|-|-|173|-|
> |+AgentPrune|8|108|7.0%|47|279|18%|14|123|11%|
>
> As shown, AgentPrune requires minimal training time, constituting only 7–18% of the overall wall-clock time. More importantly, it reduces overall execution time. For example, on GSM8K, AgentPrune reduces total time consumption by 38 minutes compared to the vanilla system, which stems from AgentPrune’s ability to prune unnecessary communication paths, significantly reducing redundant LLM API calls.
>
>
> **Regarding data overfitting**, as demonstrated in Table C, AgentPrune requires only a small number of training samples to perform effective one-shot pruning. Its generalization ability has been thoroughly validated across diverse datasets, as detailed in Section 4.2. We respectfully believe this evidence strongly indicates that AgentPrune does not suffer from overfitting issues.
>
>
> **Regarding how $K'$ affects performance**, our response to `Weakness 2` includes a corresponding analysis and experiments on this topic. We hope this clarification adequately resolves your concerns.
>
>
>
> ---
> > `Question 2`: Does the mask change depending on the input data?
>
> We address this question in two phases. **During the training phase**, the spatial-temporal graph masks dynamically adjust based on the MAS outputs for the input data, as detailed in Line 933 of Algorithm 3. **In the inference phase**, AgentPrune utilizes the sparse yet informative communication graph $\mathcal{G}^{sub}$ obtained through one-shot pruning to resolve the remaining queries. Since $\mathcal{G}^{sub}$ has already captured the critical knowledge within the current task domain, the multi-agent system can effectively complete tasks without further updates or iterations, at a significantly reduced computational cost, as demonstrated in Section 4.2.
>
> ---
> > `Question 3`: How does the proposed method perform compared to random pruning?
>
> Thank you for your feedback! We have addressed this in our response to `Weakness 1`, where we provide a detailed comparison between AgentPrune and random pruning.
>
> ---
> > `Question 4`: Can distribution approximation and low-rank sparsity work in isolation? what's their effect on the accuracy and communication reducing?
>
> We appreciate your insightful comment, which strengthens our work! In our response to `Weakness 3`, we have supplemented the discussion on the ablation study of AgentPrune.

---

> ### Author Response · Authors · 2024-11-23
> **Thank you & Looking forward to further discussion**
>
> Dear Reviewer CryL,
>
> We deeply appreciate your dedication to engaging in author-reviewer discussions. Recognizing the existing discrepancies in the understanding of our manuscript, we have outlined your key concerns and our responses for enhanced communication:
>
> 1. **Is AgentPrune more effective than random pruning?** **`Weakness 1, Question 3`** We empirically validated that AgentPrune significantly outperforms random pruning across six datasets.
> 2. **What is the training cost of AgentPrune?** **`Weakness 2, Question 1`** We have comprehensively reported the training token cost, the number of queries required, and the training time for AgentPrune.
> 3. **Missing ablation study** **`Weakness 3, Question 4`** We respectfully clarify that the relevant ablation studies are presented in Section 4.5 and Appendix H.4.1.
>
> For other issues not mentioned here, please refer to our detailed rebuttal response. We sincerely hope this addresses your concerns! We humbly look forward to further discussion with you.
>
> Warm regards,
>
> Authors

---

> ### Author Response · Authors · 2024-11-26
> **Manuscript Revision Deadline Approaching; We Are Sincerely At Your Service**
>
> Dear Reviewer CryL,
>
> We would like to thank you for your insightful review and dedication to the review process. With your help, the quality of the manuscript has been significantly improved. Due to the approaching deadline for submitting the revised manuscript, we are now submitting a revised version with highlighted changes.
>
> **Also, we would like to respectfully convey our willingness to address any further concerns you may have.** We aim to make the most of this opportunity for revision, as we deeply value your invaluable insights.
>
> Thank you once again for your support.
>
> Warm regards,
>
> Authors

---

> ### Author Response · Authors · 2024-11-29
> **Kind reminder to Reviewer CryL**
>
> Dear Reviewer CryL,
>
> This is a gentle reminder that **the discussion phase will end in less than 4 days**, but we have not yet received your feedback on our rebuttal. We respectfully understand that due to your other important commitments, your time is precious and the demand is high. But we are eager to collaborate with you to improve this paper, and we have made extensive contributions and efforts to this end. We sincerely hope that you find our response convincing and kindly consider revisiting your rating.
>
> We would like to express our gratitude once again to the reviewer for their time and all constructive feedback provided!
>
> Thanks and Regrads,
>
> Submission 2208 authors

---

> ### Author Response · Authors · 2024-12-02
> **[ Only 1 Day Remaining ] A Gentle Reminder of Feedbacks**
>
> Dear Reviewer CryL,
>
> We would like to sincerely you for your serious comments and time invested in our work. We have revised our paper and added relevant discussions and experiments.
>
> At present, all your concerns are responded in the rebuttal and revised version of the paper. However, as the revision deadline approaches (**only 1 day remaining**), we kindly request your feedback to confirm that our response and revision effectively address your concerns. If there are any remaining issues, we would greatly appreciate the opportunity to address them to ensure the quality of our work. **We sincerely hope that you find our response convincing and kindly consider revisiting your rating**.
>
> Thanks and Regrads,
>
> Authors

---

> ### Author Response · Authors · 2024-12-03
> **[ Only 8 Hours Remaining ! ] Humble Invitation to Final Discussion**
>
> Dear Reviewer CryL,
>
> Thank you for your evaluation of our work and the valuable feedback you have provided. As **the deadline for the author-reviewer discussion is in only 8 hours**, we sincerely hope you can join us for the final discussions to further enhance the quality of our manuscript.
>
> We have tried our best to give more experimental results and corresponding analysis according to your requirements. However, we do not know whether this can be a good answer to your comments. We look forward to your reply and corresponding suggestions!
>
> Thank you immensely for your time and insights!
>
> Best regards,
>
> All authors!

---

### Official Review · Reviewer_FgBo · 2024-11-04

**Soundness:** 3
**Presentation:** 2
**Contribution:** 3
**Rating:** 8
**Confidence:** 3

**Summary:**

The paper introduces AgentPrune, a communication framework designed for LLM-MA systems. AgentPrune addresses inefficiencies in traditional multi-agent communication by reducing redundant messaging, which can lead to excessive token usage and increased costs. The framework formalizes the communication between agents as a spatial-temporal message-passing graph and optimizes its token usage without compromising task performance. Key findings include a significant reduction in token overhead and improved robustness against adversarial attacks. Through extensive testing on various benchmarks, AgentPrune demonstrated cost-effective, high-performance capabilities, achieving similar or better results compared to conventional models but at a fraction of the token cost.

**Strengths:**

1. The connectivity optimization methods (distribution approximation and low-rank sparsity) appear highly effective. By applying a straightforward policy gradient approach and sparsity optimization on the graph masks, the framework efficiently enhances the performance and robustness across various multi-agent frameworks and tasks while reducing the token cost.

**Weaknesses:**

1. It appears necessary to fix the number and roles of agents before implementing AgentPrune.
2. To effectively train each version of the framework, a substantial amount of data is required to optimize the connectivity parameters through sufficient rollouts.

**Questions:**

I would like to know the required quantity of data to be generated as well as the number and types of mask configurations necessary for training in Sections 3.2 and 3.3.

---

> ### Author Response · Authors · 2024-11-19
> **[Part 1/1] Response to Reviewer FgBo**
>
> Thank you immensely for your time and efforts, as well as the helpful and constructive feedback! Here, we give point-by-point responses to your comments.
>
> ---
> > `Weakness 1`: It appears necessary to fix the number and roles of agents before implementing AgentPrune.
>
> Your observations are truly insightful! We address your questions regarding the number of agents and the roles of agents separately below:
>
> **On the number of agents**, we would like to respectfully clarify that it is more appropriate to say AgentPrune necessitates an upper bound on the number of agents, rather than the specific number of agents. In certain scenarios, AgentPrune can completely prune all the agents of a specific agent (nodes), effectively removing them from the system. This can occur when the number of agents is disproportionately high relative to the task complexity or when adversarial agents are present in the system. We have supplemented the revised manuscript with relevant case studies in `Appendix I.1 and I.2` to illustrate this behavior. This flexibility in handling agent numbers distinguishes AgentPrune from many existing methods, such as DyLAN, LLM-Debate, AutoGen, and MetaGPT, which require the agent count to be fixed and immutable.
>
>
> **On the roles of agents**, we acknowledge that the roles of agents shoule be predefined prior to the optimization process. However, this requirement is consistent with the majority of mainstream multi-agent systems, including GPTSwarm, DyLAN, LLM-Debate, and MetaGPT.
>
> ---
> > `Weakness 2`: To effectively train each version of the framework, a substantial amount of data is required to optimize the connectivity parameters through sufficient rollouts.
>
> We would like to respectfully point out that **AgentPrune does not require a substantial amount of data to optimize the spatial-temporal graph masks**. On the contrary, the training cost is negligible compared to the inference cost. To demonstrate this, we present in Table B the training and total (training + inference) token costs of AgentPrune when combined with various LLM-MAS backbones. As shown, AgentPrune requires only around 2%–6% of token consumption to complete the spatial-temporal connectivity optimization. Once this optimization is achieved, all subsequent inferences benefit from a token-efficient and economical communication structure.
>
> _Table B. Training and total token costs of AgentPrune (AP) applied to different LLM-MAS backbones on the GSM8K dataset._
> | LLM-MAS Backbone | Training Prompt Tokens | Total Prompt Tokens | Ratio | Training Completion Tokens | Total Completion Tokens | Ratio |
> |-|-|-|-|-|-|-|
> | Complete Graph + AP | 274,821| 8,526,035| 3.22% | 69,808 | 2,022,560 | 3.45% |
> | Random Graph + AP | 269,732 | 7,495,738 | 3.59% | 63,054 | 1,796,603| 3.50% |
> | AutoGen + AP | 158,474 | 3,791,251 | 4.18% | 78,899 | 1,156,884 | 6.82% |
> | GPTSwarm + AP| 91,192 | 3,526,035| 2.80% | 22,982 | 730,552 | 3.10% |
>
>
> ---
> > `Question 1`: I would like to know the required quantity of data to be generated as well as the number and types of mask configurations necessary for training in Sections 3.2 and 3.3
>
> In Table B, we present the token consumption during the connectivity optimization stage of AgentPrune. In practice, optimization is performed using only $\\{5, 10, 20\\}$ data queries, as stated in Line 402, which demonstrates the exceptionally low data requirements for training of AgentPrune.  The relevant parameter sensitivity analysis can be found in `Appendix H.4.2`.
>
> Regarding the specific mask configuration, we provide a brief summary: for any multi-agent system and its corresponding spatial-temporal communication graph $\mathcal{G}$, AgentPrune assigns trainable spatial and temporal graph masks, denoted as $\mathbf{S}^\mathcal{S}$ and $\mathbf{S}^\mathcal{T}$. Both masks are initially set to $0.5 \cdot \mathbf{I}\_{|\mathcal{V}|}$, where $\mathbf{I}\_{|\mathcal{V}|} \in \mathbb{R}^{|\mathcal{V}|\times |\mathcal{V}|}$ is a matrix of ones. After training and optimization over $Q' \in \\{5, 10, 20\\}$ queries, guided by Eq. (8), AgentPrune performs one-shot pruning of $\mathbf{S}^\mathcal{S}$ and $\mathbf{S}^\mathcal{T}$ based on a specified sparsity ratio $p\% \in \\{50\%, 30\%\\}$. This results in the pruned graph $\mathcal{G}^{sub}$, as defined in Line 310.
>
> As evident, the mask optimization process in AgentPrune is straightforward, with only two key parameters, $Q'$ and $p\%$. Should you have further questions regarding the mask configuration, we are more than happy to provide additional clarification.

---

> ### Author Response · Authors · 2024-11-23
> **Thank you & Looking forward to further discussion**
>
> Dear Reviewer FgBo,
>
> We would like to extend heartfelt thanks to you for your time and efforts in the engagement of author-reviewer discussion. To facilitate better understanding our rebuttal and revision, we hereby summarize your key concerns and our responses as follows:
>
> 1. **Does AgentPrune need to fix the number and roles of agents?** We respectfully clarify that (1) AgentPrune can adaptively regulate the number of agents, and (2) similar to many mainstream multi-agent systems, AgentPrune requires fixed agent roles.
> 2. **Does AgentPrune require extensive training data?** Our experiments demonstrate that AgentPrune requires only 2–6% of the overall token consumption for training.
>
> For other issues not mentioned here, please refer to our detalied rebuttal response. We sincerely hope this addresses your concerns! We respectfully look forward to further discussion with you.
>
> Warm regards,
>
> Authors

---

> ### Comment · Reviewer_FgBo · 2024-11-25
>
> Apologies for the late response.
>
> Overall, the authors' responses effectively addressed my concerns.
>
> Regarding the token consumption in connectivity optimization, my previous question primarily focused on the number of structural samples required for the policy gradient (i.e., M). In your responses to other reviewers, I noticed that M = 10 and Q' = 10 were mentioned. Does this imply that the policy gradient process identified the optimal structure using approximately 100 samples?

---

> > ### Author Response · Authors · 2024-11-25
> >
> > Thank you for your acknowledgment of our rebuttal! Regarding your question on *“the number of structural samples required for the policy gradient (i.e., M),”* we respectfully confirm that your understanding is correct. Specifically, in our experiments, we commonly set $M=10$ and $Q'=10$, meaning that the policy gradient leverages 10 training queries (with 10 samples per query, resulting in a total of 100 samples) to optimize the spatial-temporal connectivity. We have now included additional clarifications on the choice of $M$ in Section 4.1 and Appendix G.2. Nevertheless, we would like to respectfully clarify that this training cost is relatively modest compared to the entire dataset, as observed in `Table B` of `[Part 1/1] Response to Reviewer FgBo`, and it does not compromise the token-saving essence of AgentPrune.
> >
> > Once again, thank you for your meticulous review, and we are more than happy to address any further questions you may have.

---

> > > ### Comment · Reviewer_FgBo · 2024-11-25
> > >
> > > Yes, I now understand that the training cost is controllable. My concern has shifted to another issue: whether using 100 samples is too few to effectively train a policy with good generalization ability.

---

> > > > ### Author Response · Authors · 2024-11-25
> > > >
> > > > Thank you once again for your insightful comments! We would like to address your questions from two distinct perspectives:
> > > >
> > > > 1. **From the policy gradient perspective**
> > > >    We respectfully argue that, the relatively small number of training parameters required by AgentPrune (<100) justifies its modest sample requirements.
> > > >
> > > >    For example, in a 4-agent setting (using random graph+LLM-Debate), the trainable parameters in AgentPrune (i.e., $\mathbf{S}^{\mathcal{S}}$ and $\mathbf{S}^{\mathcal{T}}$) amount to only 22, representing 22 communication paths. In traditional reinforcement learning, even the simplest settings, such as the REINFORCE algorithm in Pac-Man game, often require hundreds to thousands of samples to achieve convergence. However, the parameter count for their policy networks, even implemented by the simplest MLP, typically exceeds 1000. Given that the graph structure in AgentPrune involves only tens of parameters, its optimization process is considerably less complex and thus converges more rapidly.
> > > >
> > > >    To illustrate this, we provide specific examples on GSM8K datasets:
> > > >    - How the graph masks evolve during training queries ([masks.pdf](https://anonymous.4open.science/r/agentprune_rebuttal-FE38/rebuttal/masks.pdf)).
> > > >    - How the differences between graph masks vary ([diff.pdf](https://anonymous.4open.science/r/agentprune_rebuttal-FE38/rebuttal/diff.pdf)).
> > > >
> > > >    From `diff.pdf`, we can observe that the graph masks stabilize after approximately 15 queries, showing minimal variation after gradient backpropagation. We attribute this to:
> > > >    1. The small parameter space of the graph masks, leading to lower optimization complexity.
> > > >    2. The similar connectivity patterns required by tasks within the same dataset, promoting rapid convergence.
> > > >
> > > >    We also respectfully note that such early convergence is not unique to AgentPrune. During our tests with vanilla GPTSwarm, which merely optimizes spatial communication graphs, similar trends of early convergence were also observed, with considerably good performance.
> > > >
> > > > 2. **From the graph modeling perspective**
> > > >    We respectfully argue that, **early convergence with strong generalization is a common phenomenon in graph machine learning**. In traditional graph learning, researchers have observed that even for larger graphs (with thousands or tens of thousands of nodes), optimizable graphs tend to stabilize after a few training epochs. For instance:
> > > >    - In [1], optimizable graphs converge within 10-15 epochs.
> > > >    - [2] early stopped the trainable graph masks within 100 epochs.
> > > >
> > > >    Moreover, these early-trained graphs exhibit desirable properties. In [1], such graphs were used for one-shot pruning and demonstrated surprising cross-architecture generalization. In [2], these graphs served as anchor graphs to guide subsequent training. Given that the graphs in these studies are significantly larger than the multi-agent graphs in AgentPrune, we believe that 100 samples are sufficient to discover high-performing and generalizable communication structures.
> > > >
> > > > Finally, regarding the generalization ability of AgentPrune, we would like to respectfully highlight that our experiments have comprehensively demonstrated its strong generalization within and across datasets (as shown in `Tables 1 & 9`), serving as empirical evidence supporting the aforementioned analyses.
> > > >
> > > > Your profound expertise commands our utmost admiration! We hope our carefully crafted explanations address your concerns satisfactorily. Once again, we sincerely thank you for your thorough review. If you now find our approach more convincing, we would be deeply grateful if you might possibly kindly consider revisiting the evaluation of our manuscript. Thank you immensely!
> > > >
> > > >
> > > >
> > > >
> > > >
> > > > ---
> > > > [1] Early-Bird GCNs: Graph-Network Co-Optimization Towards More Efficient GCN Training and Inference via Drawing Early-Bird Lottery Tickets. AAAI 2022
> > > > [2] Two Heads Are Better Than One: Boosting Graph Sparse Training via Semantic and Topological Awareness. ICML 2024

---

> > > > > ### Comment · Reviewer_FgBo · 2024-11-26
> > > > >
> > > > > This addressed my concern. I would like to raise the score.

---

> ### Author Response · Authors · 2024-11-26
> **Thank you immensely!**
>
> We would like to sincerely thank you for your inspiring discussion and stronger support of our work. We are more than happy to see that our rebuttal has properly addressed your concerns!

---

### Author Response · Authors · 2024-11-19
**Summary of Manuscript Revision**

Dear Reviewers,

We extend our sincere gratitude for your dedication to the review process. We are truly encouraged by the reviewers' recognition of several positive aspects of our paper, including **an important and well-formalized problem** (Reviewers `CryL`, `sHy9`, `rYs7`), **high effectiveness** (Reviewers `FgBo`, `sHy9`), **well-organized presentation** (Reviewers `CryL`, `rYs7`), and **comprehensive experiments** (Reviewers `sHy9`).

In addition to addressing your thoughtful comments point-by-point on the OpenReview forum, we have implemented the following updates in the newly uploaded version (all revisions are highlighted in blue):

1. **Training Cost of AgentPrune**: We have provided a detailed discussion of AgentPrune's training cost in `Appendix H.2`.
2. **Comparison with Random Pruning**: We present a comparison between AgentPrune and random pruning in `Appendix H.5`.
3. **Transferability of AgentPrune**: We have evaluated the transferability of the structures optimized by AgentPrune in `Appendix H.6`.
4. **AgentPrune with Task Adaptiveness**: We showcased that AgentPrune can well handle dynamic scenarios with a task evaluator $\psi$, and included performance results and a case study in `Appendix J`.
5. **Other Revisions**: These include typo corrections, format adjusting for `Tables 2, 3 and 5` and an analysis of the pruning analysis in `Appendix I.4`.

We have made diligent efforts to address the key concerns raised. We also look forward to addressing any further inquiries you may have!

Sincerely,

Authors

---

### Author Response · Authors · 2024-11-22
**General Response**

Dear Reviewers,

Thank you for your thorough and insightful reviews. We sincerely appreciate your feedback, which has significantly enhanced our paper! Below, we summarize the key concerns raised and our corresponding responses:

1. **Training Cost of AgentPrune** `Reviewers FgBo, CryL`
   We have provided additional details regarding the training cost of AgentPrune, including prompt/completion token usage, training queries, and runtime.
2. **Comparison with Random Pruning** `Reviewer CryL`
   We have conducted experiments to validate the effectiveness of AgentPrune compared to random pruning.
3. **Applicability of AgentPrune**  `Reviewers sHy9, rYs7`
   We have elaborated on the applicability of AgentPrune, specifying the applicable dataset size and presenting results in more dynamic scenarios (e.g., web browsing and file reading).
4. **Transferability of AgentPrune** `Reviewer sHy9`
   We have tested the transferability of AgentPrune on AQuA, MultiArith, SVAMP, and GSM8K.
5. **Formatting Issues** `Reviewers sHy9, rYs7`
   We have reorganized Tables 2, 3, and 5, as well as Figures 1 and 4, and corrected typos to improve clarity and presentation.

Thank you again for your valuable feedback. We remain eager to address any further questions or concerns you may have.

Sincerely,

Authors

---

### Author Response · Authors · 2024-12-01
**Rebuttal Progress Summary & Sincere Invitation to Discussion**

**Dear Reviewers,**

Warm greetings to you in December! We sincerely appreciate your invaluable support throughout the review process over the past few weeks. Your insightful feedback has significantly improved our manuscript.

**As the author-reviewer discussion period draws to a close in just two days**, we would like to take this opportunity to summarize the progress thus far. We are deeply grateful to `Reviewer FgBo` and `Reviewer sHy9` for their continuous engagement, thorough evaluation, and strong endorsement of our work, both of whom have graciously rated our AgentPrune with an acceptance score of 8. Thank you again for your thoughtful support!

Additionally, we extend our heartfelt thanks to `Reviewer CryL` and `Reviewer rYs7` for their valuable insights regarding the training cost/time of AgentPrune, ablation studies, and high-context-awareness scenarios. We have taken your feedback seriously and made extensive efforts to address your concerns in our responses. However, as we have yet to receive further feedback, we are uncertain whether our replies have satisfactorily resolved your concerns. **With the utmost humility, we kindly extend an invitation for further discussion** before the deadline. We would be honored and grateful to hear your thoughts.

Finally, we sincerely thank all reviewers for their dedication and thoughtful evaluations, and we are also deeply appreciative of the Area Chair’s guidance and support. Thank you!

Warm regards,
*All authors!*

---

### Meta-Review · Area_Chair_mniG · 2024-12-23

**Metareview:**

This paper looks at optimizing the communication paradigms in multi-agent systems — roughly, “can we use fewer tokens over the course of a multi-agent task solve to achieve similar or better end results than a baseline without optimization.”  Their focus is on pruning communication paths between agents - spatial paths, which are agent-to-agent interactions within a single time tick, and temporal paths, where an agent’s output at tick t may or may not be used at tick t+1.  Their general goal here is to maintain high connectivity (spatiotemporal) across the agents while minimizing comms paths and tokens flowing across the comms paths.  This AC appreciates the formalization of the multi-agent communication dynamics, sees the optimizations done as relatively straightforward, but mainly was impressed by the applied part of this project - it does look like this can be a drop-in optimization into many of the current MAS frameworks, as supported by the extensively experimental results in the latter half of the paper and the rebuttal.

**Additional Comments On Reviewer Discussion:**

We commend the authors for their extensive rebuttal.  Two reviewers were interactive with the rebuttal, and were pleased by the rebuttal's addressing of their concern.  Those reviewers voted strongly for accept.  Two reviewers, CryL and rYs7, were not responsive.  This AC looked at those reviews in greater depth as well as the authors' response to them and believes concerns have largely been addressed there - as such, this AC is relatively discounting those inactive reviewers lower scores.  Support acceptance.

---

### Decision · Program_Chairs · 2025-01-22

Accept (Poster)